# No phenotypic or genotypic evidence for a link between sleep duration and brain atrophy

Anders M. Fjell [1,2 ✉], Øystein Sørensen [1], Yunpeng Wang [1],
Inge K. Amlien [1], William F. C. Baaré [3], David Bartrés-Faz [4,5,6],
Lars Bertram[1,7], Carl-Johan Boraxbekk [3,8,9,10], Andreas M. Brandmaier [11,12],
Ilja Demuth [13,14], Christian A. Drevon[15,16], Klaus P. Ebmeier [17],
Paolo Ghisletta [18,19,20], Rogier Kievit [21], Simone Kühn [11,22],
Kathrine Skak Madsen [3,23], Athanasia M. Mowinckel[1], Lars Nyberg [1,8],
Claire E. Sexton [17,24,25,26], Cristina Solé-Padullés[4,5,6], Didac Vidal-Piñeiro [1],
Gerd Wagner[27], Leiv Otto Watne [28,29] & Kristine B. Walhovd [1,2]

Short sleep is held to cause poorer brain health, but is short sleep associated with higher rates of brain structural decline? Analysing 8,153 longitudinal MRIs from 3,893 healthy adults, we found no evidence for an association between sleep duration and brain atrophy. In contrast, cross-sectional analyses (51,295 observations) showed inverse U-shaped relationships, where a duration of 6.5 (95% confidence interval, (5.7, 7.3)) hours was associated with the thickest cortex and largest volumes relative to intracranial volume. This fits converging evidence from research on mortality, health and cognition that points to roughly seven hours being associated with good health. Genome-wide association analyses suggested that genes associated with longer sleep for below-average sleepers were linked to shorter sleep for above-average sleepers. Mendelian randomization did not yield evidence for causal impacts of sleep on brain structure. The combined results challenge the notion that habitual short sleep causes brain atrophy, suggesting that normal brains promote adequate sleep duration—which is shorter than current recommendations.

Adults are advised to sleep at least seven to eight hours each night[1–4], and it is widely perceived that shorter sleep could be a pervasive negative factor for physical, mental and cognitive health[5–8], yielding increased risk of Alzheimer's disease (AD) and other dementias[9–15]. However, we still do not know what amount of sleep is associated with good brain health and whether a causal relationship between variations in habitual sleep duration and brain health exists. Here we address these questions, analysing MRIs of the brain and genetic data in a combined longitudinal and cross-sectional design.

Brain health encompasses multiple features[16]. Important aspects can be indexed by rate of atrophy, which increases in normal ageing[17],

in cognitive decline[18], in AD[19] and with, for example, cardiovascular risk factors[20]. Lower rates of atrophy are related to healthy lifestyle[21] and better maintained cognitive function[22]. Hence, if insufficient habitual sleep has detrimental effects on the brain, it is likely that short sleep will be associated with higher rates of atrophy. Still, the evidence for a role of sleep in neurodegeneration was not considered sufficiently strong to include sleep among the 12 potentially modifiable risk factors by the Lancet Commission on dementia prevention[23], the World Health Organization guidelines on the risk reduction of cognitive decline and dementia[24] do not mention sleep, and only a few studies have tested the relationship between sleep duration and brain atrophy

using longitudinal MRIs. One study reported durations shorter and longer than seven hours to be associated with more frontotemporal grey matter loss[25], while three others found no relationships[26–28]. The paucity of relationships reported may be due to small effect sizes with insufficient statistical power and scarce sampling of very short and very long sleep durations. In a longitudinal study of 28,000 participants, faster cognitive decline was observed in individuals sleeping four hours or less or ten hours or more, compared with a reference group sleeping seven hours, with no relationship between these extreme intervals[29]. In a cross-sectional study of 21,000 participants from the UK Biobank (UKB), we found that variations within the range of five to nine hours of sleep were not related to smaller hippocampal volume, whereas shorter and longer durations were[26].

The question of how much sleep is associated with good brain health can also be addressed using cross-sectional data. There seems to be an inverted U-shaped relationship between sleep duration and brain health, since both long and short sleep are associated with increased risk of cognitive decline[30,31] and smaller regional brain volumes[26]. This pattern falls into a broader line of converging evidence from multiple sources of research. A meta-analysis of 35 studies of sleep duration and mortality found that seven hours was associated with the lowest risk[32]. Two recent very large studies found seven hours of sleep to be associated with the highest cognitive performance[33,34] and lowest dementia risk[35]. Seven hours is close to the average reported sleep duration in epidemiological studies[36], suggesting that average and 'optimal' sleep duration converge. Hence, we would expect similar estimates in cross-sectional analyses of brain morphometry. Importantly, such results cannot be used to make inferences about atrophy and brain change, as inter-individual brain volumetric differences even in adults mainly reflect early developmental processes[17,37]. Accordingly, larger brain volumes are positively and stably related to lifelong higher cognitive function and demographic variables such as education[17,38,39]. Cross-sectional sleep–volume relationships[26,40–50] therefore represent mostly stable factors, not brain changes[26] (for an overview of previous studies, see Supplementary Information, 'Reviewed studies').

Cross-sectional relationships can be further investigated using genetic information. Twin and genome-wide association studies (GWAS) have demonstrated heritability of and polygenic influences on sleep duration, although GWAS heritability is modest[51–57]. Single nucleotide polymorphism heritability (SNP-$h^2$) for sleep duration is typically below 10% (ref. [58]). To date, up to 78 independent genetic loci have been associated with sleep duration[51], among which the thyroid-specific transcription factor gene (*PAX8*) and *Vaccinia related kinase 2* (*VRK2*) have been considered as the most robust findings. Besides gene discovery, genetic overlaps between sleep duration and other conditions have been studied[51,53,59], suggesting pleiotropy between sleep duration, somatic disorders and neuropsychiatric health. However, no studies have investigated whether genes affect sleep duration uniformly for below-average versus above-average sleepers. Sleep duration tends to be positively related to health in below-average sleepers and negatively related to health in above-average sleepers. If the same is true for brain characteristics, it will be interesting to investigate genetic differences between these participants and to use Mendelian randomization (MR) analyses to examine the possible relationships between sleep duration and brain health as indexed by MRI.

Here we tested the relationship between sleep duration and rates of brain atrophy. Sleep duration was chosen as the sleep metric of focus because it is the most widely used, represents an aspect of sleep that for many people is under voluntary control and constitutes the basis for most recommendations about sleep. A higher rate of atrophy was regarded as a marker of declining brain health[17–20]. Longitudinal data from the Lifebrain consortium[60] were combined with legacy data, yielding a sample of 8,153 longitudinal MRI brain scans from 3,893 participants (20–89 years), with two to seven examinations covering up to 11.2 years (mean, 2.51; s.d., 1.45; see Table 1). Possible influences

## Table 1 | Origins of the total sample

| Study | Observations Cross-sectional/ longitudinal | Participants Cross-sectional/ longitudinal | Age (mean) | Age range |
|---|---|---|---|---|
| HCP | 974 | 974 | 28.8 | 22–37 |
| BASE-II | 675/568 | 391/284 | 63.2 | 24–83 |
| Barcelona | 113/112 | 39/38 | 70.9 | 64–81 |
| Cam-CAN | 884/504 | 632/252 | 55.1 | 20–88 |
| LCBC | 1,474/1,011 | 803/340 | 49.4 | 20–89 |
| UKB | 45,983/5,692 | 43,137/2,846 | 64.5 | 45–83 |
| Betula | 423/266 | 284/133 | 62.3 | 25–85 |
| Whitehall-II | 769 | 769 | 69.8 | 60–85 |
| Total | 51,295/8,153 | 47,029/3,893 | 63.4 | 20–89 |

HCP, Human Connectome Project; BASE-II, Berlin Aging Study II; Barcelona, University of Barcelona brain studies; Cam-CAN, Cambridge Centre for Ageing and Neuroscience; LCBC, Center for Lifespan Changes in Brain and Cognition, University of Oslo.

of relevant somatic, psychiatric and societal variables were assessed. Additional analyses were conducted using 51,295 MRIs from 47,029 participants to estimate the amount of sleep associated with the overall thickest cortex and largest regional brain volumes. Genetic analyses were undertaken to further investigate the sleep–brain relationships. We took advantage of measured variation in genes for each trait of interest and used MR[61] to explore the associations between sleep duration and brain structure.

## Results

Associations were tested by using generalized additive mixed models (GAMMs) in R[62], a nonlinear statistical approach that does not require a priori specification of a polynomial functional form[63]. Because the relationships between sleep duration and a range of health-related measures typically form an inverted U-shape, this approach allows us to accurately estimate the number of hours of sleep associated with the largest regional brain volumes and thickest cortex[64]. The code, detailed model statistics, complementary results and exact sample size for each sub-analysis are presented in the Supplementary Information. All statistical tests were two-tailed, and P values were adjusted according to the Benjamini–Hochberg procedure[65].

### Self-reported sleep across ages

Mean self-reported sleep duration per night as a function of age is shown in Fig. 1a, imposed on the US National Sleep Foundation recommendations[66]. The average sleep duration was relatively stable around seven hours across the lifespan. While it was significantly related to age ($F = 33.1$, $P < 2 \times 10^{-16}$, $N = 47,034$), age explained a very small part of the variance ($R^2 = 0.006$). The average reported sleep durations were at or below the lower recommended limits at most ages. The distributions of sleep durations as functions of different covariates are shown in the Supplementary Information, 'Subcortical cross-sectional'.

### Longitudinal sleep–brain atrophy associations

We analysed 19 volumetric brain variables and 32 cortical regions[67], summed across the hemispheres. For each measure $y$, we ran the following model for the $i$th observation of the $j$th participant:

$$y_{ij} = f\left(age_{bl,j}\right) + \beta_1\left(age_{bl,j}\right) \times sleep_j + \beta_2\left(age_{bl,j}\right)$$
$$\times time_{ij} + \beta_3\left(sleep_j\right) \times time_{ij} + covariates_{ij} + b_j + \varepsilon_{ij}$$

Here $f(age_{bl,j})$ is a smooth function of age at baseline, $age_{bl,j}$. Next, $\beta_1(age_{bl,j})$, $\beta_2(age_{bl,j})$ and $\beta_3(sleep_j)$ are varying-coefficient terms that depend smoothly on their arguments[68]. All smooth terms were

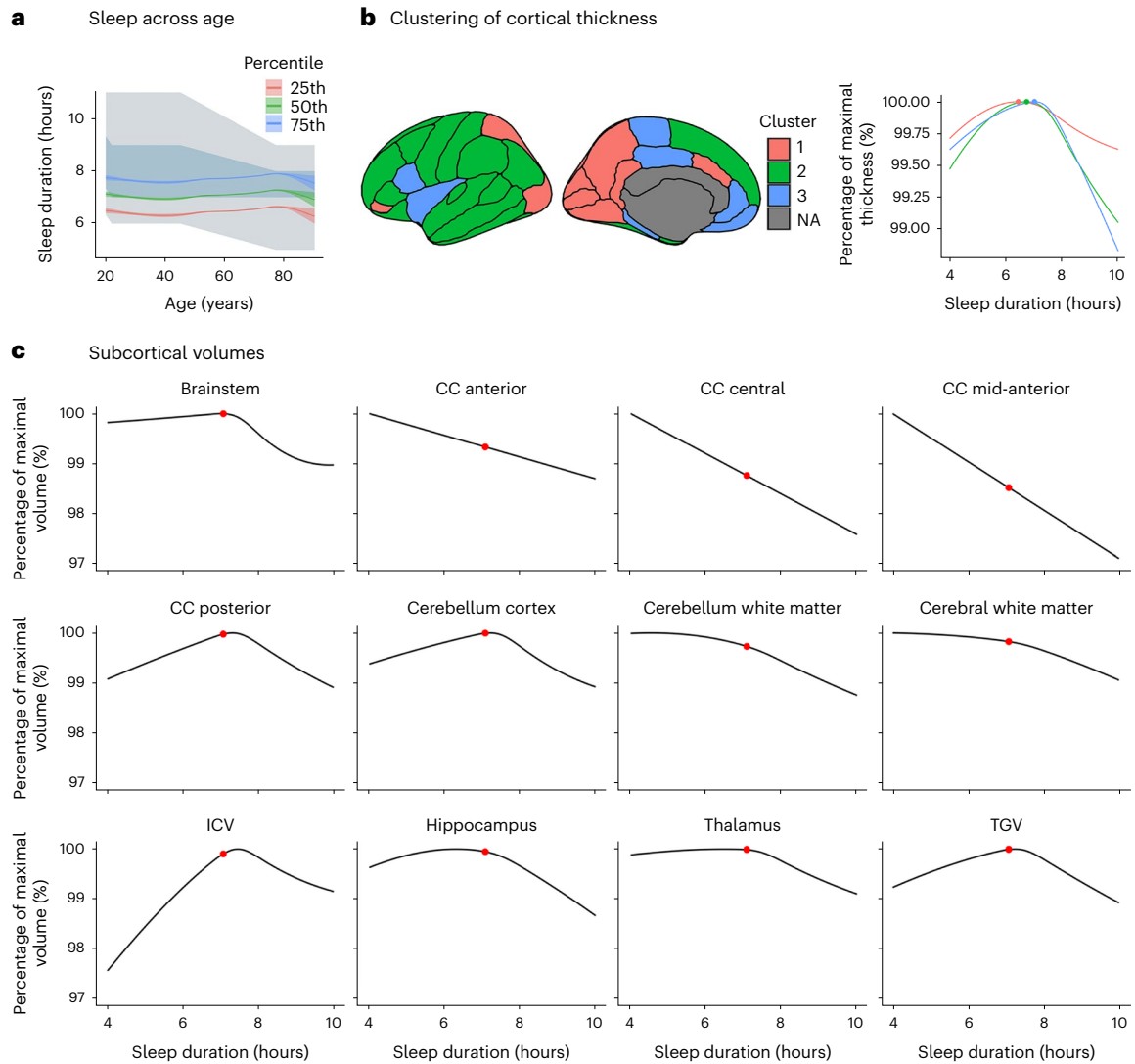

**Fig. 1 | Cross-sectional relationships. a**, Self-reported sleep duration superimposed on the recommended sleep intervals from the National Sleep Foundation. The blue/grey area depicts the recommended sleep interval (blue indicates 'recommended'; grey indicates 'may be appropriate'). The green line shows average self-reported sleep in this study; the blue and red lines show the 75th and 25th percentiles, respectively. The shaded area around each curve shows the 95% CI. **b**, Clusters of regions showing similar relationships between thickness and sleep duration. The graph shows thickness in each cluster as a function of sleep duration. The maximum thickness is 100%, illustrated by the coloured dots. NA, non-cortical region. **c**, Subcortical and global volumes as a function of sleep duration. The maximum volume is 100%. The red dots show the average reported sleep duration. Only regions significantly related to sleep duration are shown. The plots are corrected for baseline age, sex, site, follow-up time and ICV (except for the ICV plot). CC, corpus callosum.

constructed with cubic regression splines and penalized on the basis of their squared second derivatives. The term $time_{ij}$ denotes the time since baseline at the $i$th time point of the $j$th participant, and $sleep_j$ denotes the sleep duration of the $j$th participant. The first three smooth terms serve to control for the effect of age on the brain measure, the cross-sectional (between-participant) effect of sleep on the brain measure and how the effect of time depends on age, respectively. The fourth term, $\beta_3(sleep_j) \times time_{ij}$, is of primary interest, since it describes how the effect of time depends on sleep duration. Baseline age, self-reported sex, site and follow-up time were used as covariates. Intracranial volume (ICV) was included as a covariate in the volumetric analyses. Finally, $b_j$ is a random intercept term for participant $j$, and $\varepsilon_{ij}$ is a residual, both assumed normally distributed. The model was estimated using maximum marginal likelihood.

As sleep duration was available for one time point only for most of the participants, we used the average value across time points for the small number of participants for whom more than one observation was available. The cortical analyses focused on thickness, which changes considerably with age[69–71], but the results for area and volume are reported for completeness. Post hoc analyses were run controlling for socio-economic status (SES: income and education), body mass index (BMI), depression symptoms and a measure of global sleep quality in turn as covariates, as these variables may affect sleep duration, brain structure and possibly the relationship between them[72–74].

For the 32 cortical regions, no significant relationships between sleep duration and cortical thinning, volume loss or area changes were found. As can be seen in Fig. 2 (right), for no region or metric was the $P$ value smaller than 0.05. The same was seen for volume and area. In the main analyses, sex was included as a regressor. We also ran separate analyses for males and females, still yielding no evidence for significant relationships between sleep and thickness change for any cortical region (Supplementary Information, 'Cortical longitudinal').

The results for the 19 volumetric structures are shown in Table 2. Longer sleep was linearly related to greater volume loss for the caudate

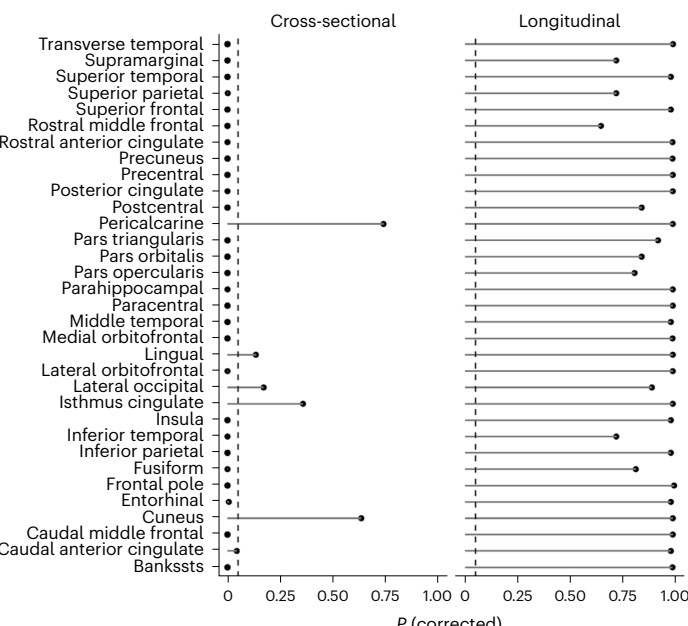

**Fig. 2 | Sleep duration, cortical thickness and thickness change.** *P* values corrected for multiple comparisons by FDR are shown for each cortical region. The left panel shows the cross-sectional results (thickness). The right panel shows the longitudinal results (thickness change). GAMMs were used for testing. The *P* values are two-sided and adjusted using the Benjamini–Hochberg procedure. The dashed lines show *P* < 0.05. The results for cortical area and volume are shown in the Supplementary Information, 'Cortical cross-sectional' and 'Cortical longitudinal'. Bankssts, banks of the superior temporal sulcus.

(*P* = 0.02), while shorter sleep was linearly related to greater volume loss for cerebellum white matter (*P* = 0.006) and thalamus (*P* = 0.006) and greater expansion of the ventricles (*P* = 0.02) (for the details, see the Supplementary Information, 'Subcortical longitudinal'). When we restricted the analyses to sleep duration at least five and no more than nine hours, the *P* value for the association between sleep duration and caudate atrophy increased to 0.07, while the other relationships were still significant. However, none of the sleep–atrophy relationships survived controlling for SES, and the relationship with the ventricles also did not survive controlling for BMI or depression symptoms, despite low correlations between sleep duration and the different covariates (for education, *r* = 0.02; for income, *r* = −0.02; for BMI, *r* = −0.04; for height (UKB only, controlling for age and sex), *r* = 0.032; for depression, *r* = −0.06; all *P* < 0.05). Controlling for the global sleep quality score did not weaken the duration–brain change relationships, but the relationships for brainstem and putamen became significant when controlling for the global score.

### Cross-sectional sleep–brain morphometry associations
We ran the model

$$y_{ij} = f(\text{age}_{ij}, \text{sleep}_j) + \text{covariates}_{ij} + b_j + \varepsilon_{ij}$$

for each brain variable, where $y_{ij}$ denotes the volume or thickness for participant $j$ at time point $i$; $f(\text{age}_{ij}, \text{sleep}_j)$ is a tensor interaction term constructed with cubic regression splines according to ref. 75; the covariates are sex, site and (for volumetric analyses) ICV; $b_j$ are random intercepts; and $\varepsilon_{ij}$ are residuals. Note that although the estimated effects are cross-sectional, all available data were used, and hence random intercepts were included. The full model was compared to two reduced models: a model in which the tensor interaction term was replaced by two additive terms, $f_1(\text{age}_{ij})$ and $f_2(\text{sleep}_j)$, and another model in which

sleep was completely removed. As these models are nested, comparison in terms of likelihood ratio tests is valid. We hence based model selection on a likelihood ratio test with a 5% significance level. This allowed us to estimate the sleep duration associated with the maximum subcortical volume and cortical thickness and the smallest ventricles. The cortical results were visualized using ggseg[76] (the vertex-wise results are shown in the Supplementary Information, 'Cortical vertex analyses').

For 27 of the 32 cortical regions, a significant relationship between sleep duration and thickness was found (Fig. 2, left)—that is, the model without sleep terms was rejected in the likelihood ratio test. When we split the analyses by sex, five regions showed significant sleep–thickness relationships in males only (cuneus, lateral orbitofrontal, lateral occipital, fusiform and entorhinal). A formal sex-interaction analysis of these regions showed a significant effect of sex only for lateral orbitofrontal cortex, where very short and very long sleep were both more associated with thinner cortex in males than females.

The results for the total sample were entered into a *K*-means cluster analysis to reduce the dimensionality of the cortical data. This yielded three clusters (Fig. 1b and Supplementary Information, 'Cortical cross-sectional'). One cluster (Cluster 1) covered the posterior medial cortices, superior parietal and caudal anterior cingulate cortex. The second and largest cluster (Cluster 2) included most of the lateral surface and superior frontal cortex. The third cluster (Cluster 3) included the rest of the lateral cortex (that is, insula and pars opercularis) and medial regions such as medial orbitofrontal cortex, rostral anterior cingulate, posterior cingulate and parahippocampal gyrus. All clusters showed inverted U-shaped relationships to sleep duration, but Cluster 1 showed the weakest effect. The vertex-wise analyses confirmed this general finding, showing only positive relationships between sleep duration and thickness in the below-average sleepers, and only negative relationships in the above-average sleepers (Supplementary Information, 'Cortical vertex analyses'). To exclude the possibility that the use of different scanners influenced the results, we also used a two-stage approach. We first ran the thickness–sleep duration GAMMs separately in each sample and then performed meta-analysis of the different cohort results. This yielded very similar estimates, demonstrating that the use of different scanners did not bias the results (Supplementary Information, 'Mega vs meta-analytic approach').

The volumetric results are shown in Fig. 1c, Fig. 3 and Table 3. Most structures showed significant inverse U-shaped relationships to sleep duration. The sleep durations associated with the maximum subcortical volume, the thickest cortex and the smallest ventricles in Table 3 were entered into a meta-analysis. Weights were applied such that cortex and subcortex contributed equally to the meta-analytic fit. We excluded total grey matter volume (TGV) since this variable is a sum of other included variables. Corpus callosum structures were excluded because, with one exception, their estimated sleep duration at maximum volume could not be defined (monotonous sleep–volume relationship). Random-effects meta-analysis was used, to allow regions to have different sleep durations associated with maximum volume or thickness. The estimates and standard errors were computed by 5,000 Monte Carlo samples from the empirical Bayes posterior distribution of the model for each region, constraining the number of hours of sleep to be between four and ten. The detailed results are presented in the Supplementary Information, 'Meta-analysis'. A sleep duration of 6.5 hours was associated with the maximum subcortical volume, the smallest ventricles and the thickest cortex. The critical values, as defined by the 95% confidence interval (CI), were 5.7 and 7.3 hours. Variability across age was small, while variability across regions was considerable. Controlling for the effects of SES, BMI and depression symptoms as covariates had no notable effects on the results, and no significant interaction effects with these variables were found (Supplementary Information, 'Subcortical cross-sectional'). The analyses were also run controlling for the global sleep quality score (Fig. 3). Except for the thalamus, where sleep duration at maximum volume

**Table 2 | Associations between sleep duration and brain volumetric change**

| | Range of sleep duration | | Controlling for additional covariates | | | |
|---|---|---|---|---|---|---|
| | Full range | Restricted (5–9 h) | SES | BMI | Depression | Sleep quality |
| **Brain region** | **_F_, d.f., _N_, _P_** | **_F_, d.f., _N_, _P_** | _F_, d.f., _N_, _P_ | **_F_, d.f., _N_, _P_** | **_F_, d.f., _N_, _P_** | **_F_, d.f., _N_, _P_** |
| Accumbens | 0.07, 1, 8,153, 0.88 | 0.56, 1, 7,966, 0.69 | 2.73, 1, 4,654, 0.59 | 1.01, 2.09, 5,416, 0.52 | 1.94, 2.42, 5,683, 0.45 | 3.18, 1, 4,287, 0.16 |
| Amygdala | 4.15, 1, 8,151, 0.13 | 3.75, 1, 7,964, 0.15 | 1.84, 2.66, 4,652, 0.59 | 4.5, 3.08, 5,414, **0.03** | 3.83, 1, 5,681, 0.24 | 0.83, 1, 4,287, 0.44 |
| Brainstem | 4.22, 2.27, 8,137, 0.05 | 4.53, 2.35, 7,950, 0.03 | 2.25, 1, 4,636, 0.63 | 1.08, 1, 5,398, 0.52 | 1.42, 1.69, 5,665, 0.45 | 7.83, 1, 4,279, **0.02** |
| Caudate | 8.48, 1, 8,146, **0.02** ↓ | 4.26, 1.3, 7,959, 0.07 | 1, 1, 4,650, 0.66 | 10.39, 1, 5,412, **0.02** | 18.51, 1, 5,679, **0.00** | 8.54, 1, 4,286, **0.02** |
| CC anterior | 0.01, 1, 8,140, 0.98 | 0, 1, 7,953, 0.98 | 0.44, 1, 4,640, 0.69 | 0.16, 1, 5,402, 0.83 | 0.05, 1, 5,669, 0.88 | 0, 1, 4,280, 0.99 |
| CC central | 0.41, 1, 8,144, 0.7 | 0, 1, 7,957, 0.98 | 0.02, 1, 4,646, 0.9 | 1.09, 1, 5,408, 0.52 | 0.55, 1, 5,675, 0.67 | 1.85, 1, 4,284, 0.3 |
| CC mid-anterior | 1.4, 1, 8,132, 0.43 | 1.78, 1, 7,947, 0.36 | 0.04, 1, 4,638, 0.9 | 1.09, 1, 5,398, 0.52 | 0.73, 1, 5,665, 0.6 | 5.16, 1, 4,272, 0.07 |
| CC mid-posterior | 0.03, 1, 8,114, 0.94 | 0.15, 1, 7,927, 0.8 | 1.32, 1, 4,626, 0.66 | 0.22, 1, 5,388, 0.83 | 0.36, 1, 5,655, 0.74 | 4.23, 1, 4,278, 0.1 |
| CC posterior | 2.72, 1, 8,100, 0.22 | 2.95, 1.81, 7,913, 0.15 | 0.71, 1, 4,618, 0.66 | 2.39, 1, 5,372, 0.42 | 1.26, 1, 5,637, 0.46 | 2.01, 1, 4,258, 0.28 |
| Cerebellum cortex | 0.39, 1, 8,129, 0.7 | 0.82, 1, 7,944, 0.58 | 0, 1, 4,636, 0.99 | 0.02, 1, 5,390, 0.91 | 0.01, 1, 5,657, 0.93 | 0.34, 1, 4,271, 0.67 |
| Cerebellum white matter | 13.46, 1, 8,116, **0.01** ↑ | 12.63, 1, 7,929, **0.01** ↑ | 0.85, 1, 4,630, 0.66 | 12.66, 1, 5,384, **0.01** | 8.76, 1, 5,651, **0.04** | 27.32, 1, 4,272, **0.00** |
| Cerebral white matter | 1.84, 3.39, 8,150, 0.22 | 2.63, 2.35, 7,963, 0.15 | 1.08, 3.5, 4,652, 0.66 | 0.25, 1, 5,412, 0.83 | 0.22, 1, 5,679, 0.76 | 0.91, 1, 4,286, 0.44 |
| ICV | 0.34, 1, 8,160, 0.7 | 1.06, 1, 7,973, 0.52 | 0.88, 1.71, 4,654, 0.66 | 0.28, 1, 5,416, 0.83 | 0.07, 1, 5,683, 0.85 | 0, 1, 4,290, 0.99 |
| Hippocampus | 2, 1.8, 8,150, 0.43 | 0.26, 1, 7,963, 0.72 | 1.88, 2.33, 4,646, 0.64 | 1.33, 1.86, 5,408, 0.52 | 1.75, 1.97, 5,675, 0.45 | 0.83, 1, 4,288, 0.44 |
| Pallidum | 5.46, 1, 8,157, 0.07 | 3.47, 1, 7,970, 0.15 | 3.41, 2.24, 4,654, 0.59 | 5.83, 1, 5,416, 0.08 | 4.3, 1, 5,683, 0.21 | 4.01, 1, 4,289, 0.1 |
| Putamen | 0.87, 1, 8,138, 0.58 | 0.45, 1, 7,953, 0.7 | 0.57, 1, 4,642, 0.66 | 0.01, 1, 5,404, 0.91 | 1.49, 1, 5,671, 0.46 | 6.28, 1, 4,278, **0.04** |
| Thalamus | 12.39, 1, 8,150, **0.01** ↑ | 10.23, 1, 7,963, **0.01** ↑ | 3.28, 1, 4,650, 0.59 | 8.97, 1, 5,412, **0.03** | 4.61, 1, 5,679, 0.2 | 14.05, 1, 4,284, **0.00** |
| TGV | 0.32, 1, 8,152, 0.7 | 0.32, 1, 7,965, 0.7 | 1.54, 3.29, 4,648, 0.59 | 2.81, 3.2, 5,410, 0.42 | 2.67, 3.22, 5,677, 0.46 | 0.07, 1, 4,286, 0.89 |
| Ventricles | 9.24, 1, 8,139, **0.02** ↓ | 4.56, 2.86, 7,952, **0.02** ↓ | 0.08, 1, 4,650, 0.9 | 1.84, 1, 5,409, 0.44 | 2.21, 1, 5,678, 0.45 | 12.71, 1, 4,275, **0.00** |

The _P_ values are two-sided and adjusted using the Benjamini–Hochberg procedure (bold indicates _P_ < 0.05), and all models were controlled for baseline age, sex, site and follow-up time. Downwards arrows (↓) indicate that longer sleep is associated with greater volume loss; upwards arrows (↑) indicate that shorter sleep is associated with greater volume loss. As the _P_ values are corrected for multiple testing, they are not identical to the _P_ values that can be computed from the d.f., _F_ value and _N_. For GAMMs, the d.f. reported are estimated degrees of freedom, representing the amount of nonlinearity in the fit. This means that conventional _F_-tests are not used, and _P_ values for smooth terms are computed using an algorithm accounting for the fact that the degrees of freedom are estimated and not fixed.

was reduced to the lower limit (four hours) when controlling for global sleep quality, most peak estimates were similar for the default model versus the model including global sleep quality as a covariate.

Since ICV showed a relationship with sleep duration, we reran the cross-sectional meta-analysis without controlling for ICV. As expected, this affected the results, yielding 7.0 hours (95% CI, (6.3, 7.7)) as the duration associated with maximum volume and thickness.

**GWAS, polygenic scores and MR**

To explore the possible associations between brain structure and sleep duration, we performed a series of genetic analyses using cross-sectional data from UKB. The hippocampus, TGV and ICV were chosen as the regions of interest (ROIs) as they showed the typical inverted U-shaped relationship to sleep duration. For details about the selection of participants, quality control procedures and genetic analyses, see the Supplementary Information, 'Genetic analyses', 'Genetics notes' and 'Genetics tables'.

The sample was stratified into shorter-than-average (≤7 hours) and longer-than-average (>7 hours) sleepers. Since an inverse U-shaped relationship between sleep duration and health—including brain health—is established, both short and long sleep are associated with poorer health. Importantly, the genetic contributions to sleep duration and brain health may be different in short sleepers compared with long sleepers[51], and hence different relationships were expected in these two groups. Two independent samples were used for GWAS: (1) participants sleeping ≤7 hours without MRI (_N_ = 197,137) and (2) participants sleeping >7 hours without MRI (_N_ = 112,839). GWAS was performed independently for each trait in the corresponding sample. We further performed GWAS for hippocampal volume, TGV and ICV using the 29,155 UKB participants who were not included for the

sleep duration GWAS. The GWAS results for these brain features were used for the polygenic score (PGS) and MR analysis bellow. Further details, Manhattan plots and QQ plots showing the GWAS results are presented in the Supplementary Information, 'Genetic analyses'. We did not observe noticeable inflation in the association statistics ($\lambda = 1.03$ and 1.02 for shorter- and longer-than-average sleepers, respectively).

We discovered three genomic loci significantly associated with sleep duration for participants sleeping ≤7 hours and one for those sleeping >7 hours, with minor allele frequency (MAF) >0.001. The three loci for short sleep included a region on chromosome 3 (hg19, chr3:52978418–53171555), a region on chromosome 11 (chr11:116631186–117072176) and a region on chromosome 15 (chr15:54586505–54622690). Genes mapped to these regions include _APOA1/4/5_, _APOC3_, _ZNF256_, _BUD13_, _UNC13C_, _SIDT2_, _TAGLN_, _SIK3_, _PCSK7_, _RFT1_, _SFMBT1_ and _PAFAH1B2_. The only locus for longer-than-average sleepers was mapped to chromosome 3 (chr3:70671137–70843060) and included two pseudo-genes, _COX6CP6_ and _RNU6-281P_, neither of which yet has known functions.

SNP-$h^2$ was estimated by the linkage disequilibrium (LD) regression models[77] for sleep duration in the shorter-than-average sleepers ($h^2 = 0.045$, s.e. = 0.0035) and longer-than-average sleepers ($h^2 = 0.021$, s.e. = 0.0047), hippocampal volume ($h^2 = 0.29$, s.e. = 0.03), TGV ($h^2 = 0.22$, s.e. = 0.03) and ICV ($h^2 = 0.35$, s.e. = 0.03). The genetic correlation for sleep duration was negative for the shorter- versus longer-than-average sleepers ($r_g = -0.40$, s.e. = 0.10, $P = 9.65 \times 10^{-5}$), showing that the genes related to longer sleep in the below-average-sleep-duration group are related to shorter sleep in the above-average-sleep group.

Corresponding PGSs were calculated for each variable in each sleep duration group separately. The PGSs for ICV (PGS-ICV: t = 8.47;

$P$ corrected by false discovery rate (FDR), $2.4 \times 10^{-15}$) and TGV (PGS-TGV: t = 4.65, $P_{FDR} = 3.28 \times 10^{-5}$) were significantly associated with sleep duration in the shorter-than-average sleepers (Fig. 4). The PGS for sleep duration in the shorter-than-average sleepers was significantly related to ICV (t = 6.99, $P_{FDR} = 3.03 \times 10^{-11}$, Fig. 4b) and to a lesser extent with TGV (t = 2.69, $P_{FDR} = 6.42 \times 10^{-2}$). No significant associations were identified for other pairs of traits.

We performed bidirectional MR analysis for each brain volumetric trait to sleep duration (see also Supplementary Information, 'STROBE-MR-checklist', reporting according to best practice for MR studies). Among the 12 pairs, ICV showed an effect (34 instrumental SNPs; minimal $F$ statistics, >24; inverse-variance weighted $\beta$; 0.060; s.e. = 0.017; $P = 5.36 \times 10^{-4}$) on sleep duration for the shorter-than-average sleepers (Fig. 4c and Supplementary Information, 'Instrumental variables'), with no evidence of effects of sleep on ICV. TGV showed a trend-level effect for the shorter-than-average sleepers ($P = 0.12$). The low heritability for sleep in the study resulted in a weaker genetic instrument. While we were powered (>80%) to detect a true causal effect for hippocampal volume and ICV of 0.3 or larger, the low heritability for sleep in the study required a much larger sample size, on the basis of the Freeman model for power calculations in MR studies[78]. We therefore performed a robust MR analysis using the robust adjusted profile score[79] for the direction from sleep to brain traits, but we did not detect significant relationships for the directionality of effects by this even more liberal threshold. The only significant relation was robust when we performed the analysis using both a stringent and a weaker instrument selection protocol. This means that we did not detect evidence for strong effects of sleep duration on brain morphometry. See the Supplementary Information, 'Genetic analyses' and 'Genetic notes', for the full statistical results.

## Discussion

The current results give no indication that shorter or longer habitual sleep duration is associated with higher rates of brain atrophy measured longitudinally. Across a range of cortical and subcortical regions and metrics, no statistically significant relationships were observed when controlling for BMI, SES or depression symptoms. The absence of significant relationships was observed both when using the full range of self-reported sleep durations and when restricting the sample to those sleeping between 5 and 9 hours. Cross-sectionally, 6.5 hours of sleep was associated with the maximum relative regional brain volume and cortical thickness and the smallest ventricular volumes when controlling for ICV, with the critical lower limit being 5.7 and the higher limit 7.3 hours. This was true also when controlling for a measure of global sleep quality. A duration of 6.5 hours is below the lower limit of the current international recommendations, and 7.3 is lower than the upper limit suggested by the US National Sleep Foundation[1–4]. ICV was positively related to sleep duration, so not controlling for ICV yielded a cross-sectional association with maximum volume and thickness of 7.4 hours. Aligning with the longitudinal results, the MR analyses did not reveal evidence for an impact of short sleep on brain structure. Taken together, the longitudinal, cross-sectional and genetic results suggest that short habitual sleep duration is weakly related to poorer brain health in healthy adults as indexed by structural brain measures, and that somewhat less than 7 hours of sleep is associated with the most favourable features, in line with converging evidence from research on mortality, health and cognition.

### Sleep duration and the brain

Sleep duration is the most widely studied, best supported and most straightforward sleep measure to address in relation to health[7]. It is also an aspect of sleep that may partly be modified by lifestyle. Our longitudinal results did not yield evidence for any relationship between sleep duration and brain atrophy. We therefore used the full sample of cross-sectional data to calculate the amount of sleep associated

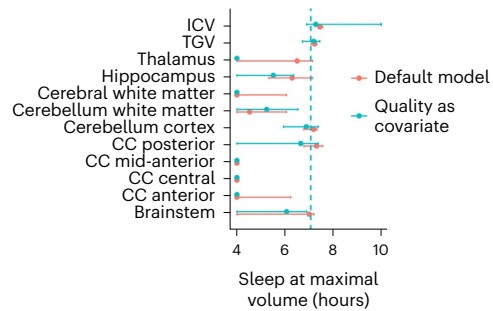

**Fig. 3 | Sleep at maximum subcortical volume.** The sleep durations associated with the maximum subcortical volume are indicated by the dots. Only regions significantly related to sleep duration are shown. The error bars indicate 95% CIs ($N = 47{,}029$; 51,295 observations). The default model is shown in red, and the model including global sleep quality as a covariate is shown in turquoise.

with maximum relative regional brain volume and cortical thickness. It is noteworthy that the resulting 6.5 hours is relatively well aligned with the average reported sleep duration of 7 hours and similar to the results of a recent meta-analysis of more than one million participants[36]. Furthermore, this corresponds with the conclusion of a meta-analysis of 35 studies of sleep duration and mortality, where 7 hours of sleep was associated with lowest risk of premature death among adults[32]. A study of more than 700,000 participants found 7 hours of sleep to be associated with the highest performance on a spatial navigation task[34], and a prospective study of more than 400,000 UKB participants found the lowest dementia risk in those reporting to sleep 7 hours[35]. Converging evidence thus suggests that somatic and brain health are associated with about 7 hours of sleep.

Results across studies suggest that substantially shorter sleep than the recommended duration does not need to be associated with worse outcomes. However, longer sleep may be. Previous research has established associations between long sleep and poorer brain[48], cognitive[25,26,29,49,80] and somatic health[81]. For instance, less than 5 hours and 8 hours of sleep were associated with similar increases in risk of premature death[32]. This mirrors the present results: 4 and 8 hours of sleep were associated with the same deviations from maximum cortical thickness. The American Academy of Sleep Medicine and the Sleep Research Society proposed no upper limit[7] on sleep duration, whereas the US National Sleep Foundation recommended a maximum of 9 hours through most of adulthood and 8 hours in older adults[66]. From the present analyses, the critical values for short and long sleep were 5.7 and 7.3 hours, demonstrating that sleep durations longer than average but well within the recommended range may still be associated with less favourable volumetric brain outcomes. Associations between longer sleep duration and worse outcomes are often ascribed to underlying comorbidities[7,66,81]. We addressed this by controlling for somatic (BMI), mental (symptoms of depression) and social (SES) factors. Importantly, the longitudinal results showed that neither long nor short sleep was associated with higher rates of brain atrophy. We therefore believe that the combined longitudinal and cross-sectional results make a strong case that short habitual sleep is not a prevalent cause of poorer brain health as indicated by structural brain measures and rates of atrophy in the samples studied here.

In this regard, the association between ICV and sleep duration is interesting. ICV was the MRI-derived measure most positively associated with sleep duration, and the MR analysis suggested an effect of ICV on sleep duration in the shorter-than-average sleepers but not the inverse. As sleep has no causal effect on ICV in adults, this relationship must reflect other factors and demonstrates that associations between sleep duration and MRI-derived volumes may reflect non-causal and stable relationships that do not emerge as a function of variations in sleep duration. The partly common genetic underpinning of ICV and

**Table 3 | Estimated sleep duration in hours associated with maximum (minimum for ventricles) volume or thickness for the variables used in the meta-analysis**

| Region | Sleep at max volume (s.e.) | 95% CI (low, high) | P (corrected) | d.f. | N | F |
|---|---|---|---|---|---|---|
| Accumbens | 7.1 (3.0) | 4.0, 10.0 | 0.98 | 1 | 51,284 | 0 |
| Amygdala | 4.1 (0.9) | 4.0, 4.0 | 0.07 | 1 | 51,283 | 3.63 |
| Brainstem | 6.3 (1.2) | 4.0, 7.2 | **5×10⁻⁷** | **3.58** | **51,266** | **9.92** |
| Caudate | 9.9 (0.9) | 10.0, 10.0 | 0.06 | 1 | 51,276 | 4.03 |
| Cerebellum cortex | 7.2 (0.3) | 6.9, 7.4 | **3.3×10⁻⁵** | **3.42** | **51,260** | **7.64** |
| Cerebellum white matter | 4.7 (0.8) | 4.0, 6.3 | **8.5×10⁻⁵** | **2.06** | **51,255** | **9.97** |
| Cerebral white matter | 4.6 (0.8) | 4.0, 6.3 | **1.2×10⁻⁵** | **2.29** | **51,280** | **11.25** |
| Hippocampus | 6.3 (0.5) | 4.7, 7.1 | **0.00** | **2.96** | **51,279** | **13.72** |
| Pallidum | 9.0 (2.2) | 4.0, 10.0 | 0.33 | 1 | 51,286 | 1.03 |
| Putamen | 4.5 (1.6) | 4.0, 10.0 | 0.20 | 1 | 51,267 | 1.91 |
| Thalamus | 6.0 (1.1) | 4.0, 7.2 | **1.0×10⁻⁵** | **3.02** | **51,279** | **9.21** |
| Ventricles | 5.6 (1.3) | 4.0, 7.3 | **0.002** | **2.62** | **51,268** | **5.61** |
| Cortex Cluster 1 | 6.4 (0.4) | 5.7, 7.1 | **0.00** | **2.86** | **51,460** | **7.10** |
| Cortex Cluster 2 | 6.7 (0.2) | 6.3, 7.1 | **0.00** | **3.52** | **51,460** | **25.3** |
| Cortex Cluster 3 | 7.0 (0.2) | 6.4, 7.2 | **0.00** | **3.53** | **51,460** | **25.6** |

Clusters 1, 2 and 3 refer to the cortical thickness clusters in Fig. 1. ICV was used as a covariate in the volumetric analyses. GAMMs were used for testing. The P values are two-sided and adjusted using the Benjamini–Hochberg procedure (bold indicates P<0.05). For the full details, see the Supplementary Information, 'Subcortical measures cross-sectional' and 'Cortical measures cross-sectional'.

sleep duration suggests that there may be a mechanistic association, but this is not caused by sleep. Controlling for ICV removes the effect of global scaling—that is, that regional brain volumes scale with head size. Since ICV is sometimes regarded as a proxy for maximal brain size, controlling for ICV yields regional volumes representing deviations from the expected based on head size. Controlling for ICV also controls to some extent for body size, as head size and body size are normally related, although height and BMI were weakly related to sleep duration in the present data. Not controlling for ICV naturally led to a higher sleep duration estimate of 7.4 hours associated with maximal brain volumes and thickness, as ICV and sleep duration were positively related.

Underlying neurobiology cannot be directly inferred from MRIs, but the number of neurons has been shown to correlate with regional[82] and global[83] brain volumes cross-sectionally. Longitudinally, volumetric reductions and cortical thinning occurring during adulthood may be associated with the shrinkage of neurons, dendrites and axonal arborizations[84,85], reduced spine numbers and density[86], loss of synapses and dendritic branches[87], and, in degenerative conditions such as AD, also neuronal loss[84], although neuronal[85,88,89] or glial[90] loss probably plays a limited role in the volumetric reductions seen in normal ageing. Sleep duration–brain correlations were seen in the cross-sectional analyses only, so it is unlikely that these are caused by neurobiological events underlying morphometric changes observable with MRI during adulthood. Events ongoing during earlier life stages, in development, may thus be more relevant. Processes such as synaptogenesis and synapse elimination/pruning[91], dendritic and axonal growth[92,93], and intracortical myelinization[94] can be involved in morphometric changes in childhood development. Some of these, however, such as synaptic density, will have minute effects on volumetric measures because their total volume is very small[93,95]. In any case, it must be stressed that the volumetric analyses in the present study are corrected for ICV, which means that relative and not absolute volumes are used in the calculations. It is thus not clear which or any of the processes above can contribute to explaining the observed cross-sectional relationship. Research into the neurobiology of sleep has focused more on electrophysiological processes and neurotransmitter systems than on macrostructural differences. In addition, the association between neurodegeneration and sleep problems may be due to any disturbance of normal brain function and structure probably affecting how we sleep, and the neurobiological foundation will then vary depending on the underlying condition. Hence, a neurobiological interpretation of the present findings will be speculative and must be based on general knowledge about the relationship between brain features and different human traits. For example, we have previously shown that sleep disturbances are associated with spatial expression patterns of oligodendrocytes and S1 pyramidal cell genes[71], in line with theories of relationships between myelination and sleep[96]. To our knowledge, such analyses have not been reported for sleep duration specifically.

### Genetic associations

Sleep duration is a complex trait modulated by more than the core circadian genes[51]. Previous studies have reported GWAS heritability to be modest[51–57], which limits the power of the MR approach. Still, an MR study from UKB found that both short and long sleep were related to poorer visual memory and longer reaction time[97]. In contrast, another study did not find causal relationships between sleep patterns and AD[98]. In the current study, the genetic association analyses yielded some interesting results. First, we found that the genetic correlation for below- versus above-average sleepers was negative. This means that the genes related to longer sleep in the shorter-than-average sleepers were related to shorter sleep in the longer-than-average sleepers. This could mean that there is a genetically influenced drive towards the average reported sleep duration, for both the above- and below-average sleepers. This is interesting considering the present findings of larger regional brain volumes and cortical thickness in participants sleeping 6.5 hours, as well as the above reviewed evidence that ~7 hours of sleep is associated with better health and cognitive performance[34]. Second, the GWAS results suggested that genes involved in metabolism (for example, *APOA1/4/5*, *APOC3* and *RFT1*) may contribute to inter-individual differences in sleep duration in the shorter-than-average sleepers. Genetic and cellular links between sleep and metabolism are a focus of current research[99,100], although a further investigation into the details of these relationships is beyond the scope of this paper. Third, we found that PGSs for ICV and TGV were

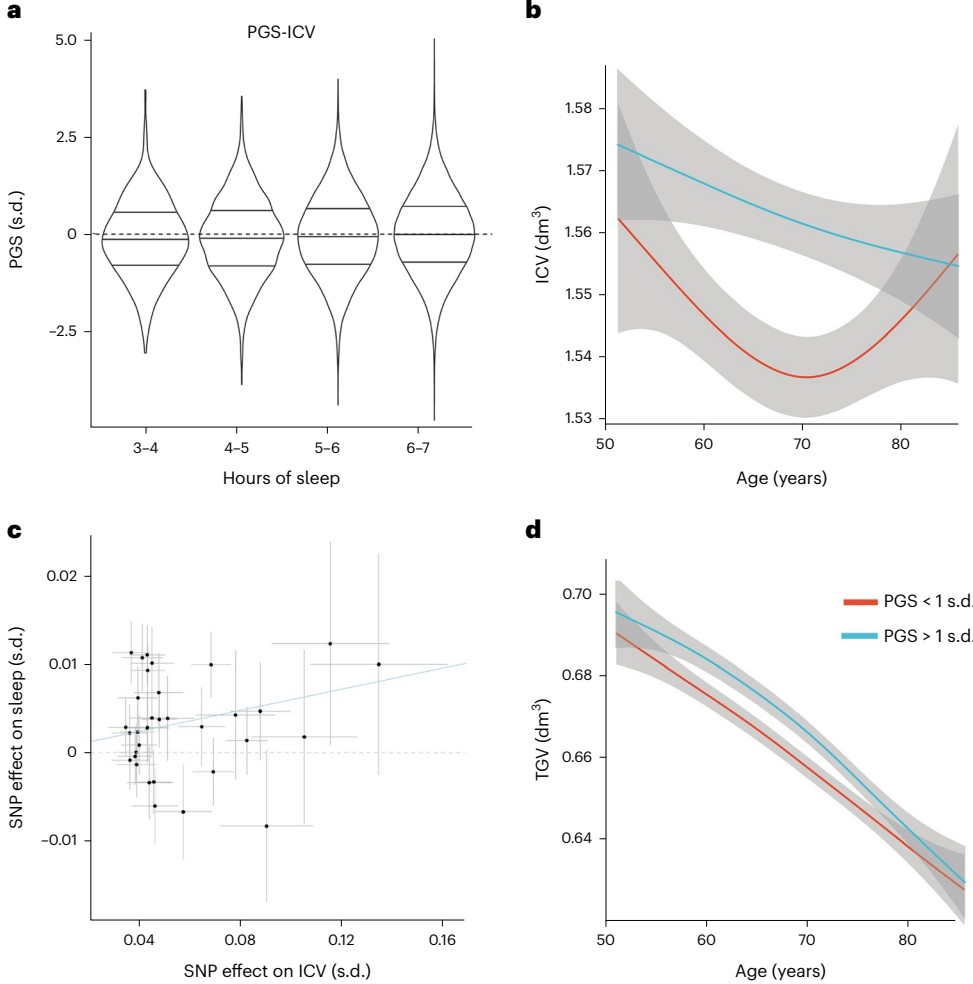

**Fig. 4 | Genetic relations between sleep duration and brain structure. a**, Distributions of PGSs for ICV in different sleep duration strata among shorter-than-average sleepers (3–4 h, N = 541; 4–5 h, N = 2,937; 5–6 h, N = 13,863; 6–7 h, N = 177,493), expressed in s.d. compared with the sample mean of 0. The horizontal lines in each violin represent the median group value and the interquartile range, the height represents the 95% CI and the width represents the probability density. **b**, ICV for shorter-than-average sleepers with one standard deviation above (blue) and below (red) the average PGS for sleep duration. The grey shaded areas represent the 95% CIs. **c**, SNP effects on ICV (x axis) and sleep duration (y axis) for the shorter-than-average sleepers. **d**, TGV for shorter-than-average sleepers with one standard deviation above (blue) and below (red) the average PGS for sleep duration. The grey shaded areas represent the 95% CIs.

significantly positively associated with sleep duration in the short sleepers, which means that genes related to larger brain volumes are also related to longer sleep in the short sleepers. This finding aligns relatively well with our estimate of 6.5 hours being associated with the largest relative brain volumes. This suggests partly shared genetic variation between regional brain volumes and sleep duration. Finally, the MR analyses showed an effect of ICV on sleep duration, an effect that was robust after accounting for confounding factors (BMI, smoking and drinking habits, and neuropsychiatric disorders; Supplementary Information). However, there was no evidence of causal effects of sleep duration on any MRI-derived brain measure. Hence, in the current samples, people with larger heads on average report that they sleep longer, and this relationship partly depends on genetics. The lack of evidence for an inverse influence of sleep duration on ICV was given, as ICV does not change in adults and hence cannot be affected by sleep. Still, the genetic results suggest that there may be a mechanistic relationship between ICV and sleep duration that could warrant further explorations. This effect was removed from the estimated sleep duration–brain volume relationships by covarying for ICV, which may contribute to explaining why the nominally significant relationship between TGV and sleep duration in the MR analysis did not survive

corrections. In sum, the genetic results were in coherence with a view of average and 'optimal' sleep duration as relatively well aligned and did not provide evidence for a causal relationship between sleep duration and brain structural features.

### Variation across persons, ages and regions

The meta-analytic estimate of 6.5 hours is a best approximation, not a magic number. First, there was substantial regional heterogeneity in the cross-sectional results. For instance, the hippocampus showed peak volume at 6.3 hours, the white matter compartments even lower, while people with the thickest cortex reported sleeping between 6.4 and 7.0 hours. This does not mean that less sleep is necessarily more optimal for hippocampal volume than for cortical thickness, and these differences should be interpreted while keeping in mind the lack of evidence for sleep–atrophy relationships in the longitudinal analyses. These numbers therefore probably represent stable relationships rather than reflecting the effects of sleep duration per se.

Second, an important qualification is that individual differences in sleep need exist, due to, for instance, genetic differences and previous sleep history[51–57,101]. If deviations from an individuals' sleep need led to poorer brain health for that individual, this may not be picked up

in our group analyses. We thus cannot from our results conclude that people should try to sleep 6.5 hours each night. Rather, people who report sleeping 6.5 hours tend to have the thickest cortex and largest regional brain volumes relative to ICV. Intra-individual causal effects of changing sleep duration on the brain can be assessed in sleep deprivation studies. Unfortunately, experimental sleep deprivation does not resemble habitual variations in sleep duration, and the long-term consequences on the brain from sleep deprivation, taking adaptations into account[101], are not known.

Finally, associations with sleep could vary with age[71], but the sleep–age interactions did not confirm that this was the case. The vertex-wise cortical thickness analyses (Supplementary Information) suggested that relationships with sleep duration were different in younger and older adults, but this was not confirmed in the GAMM analyses. We therefore believe that the meta-analytic results represent a good approximation of a general sleep–brain volume relationship on a group level, while ignoring that there naturally are variations across brain regions, ages and participants.

## Caveats and limitations

The first limitation of our study is that self-reported sleep duration is not accurate and may reflect several other aspects of sleep than duration only. There is no perfect way to measure sleep duration without disrupting routine[102]. Self-reports are only moderately correlated with actigraph measures[102,103]. However, although actigraph results often correlate highly with polysomnography[104], they tend to overestimate sleep duration[104–108], and it is not known how well actigraphs perform outside a sleep lab setting. One study reported that the same genetic loci were related to sleep duration whether it was measured by actigraphs or self-reports[51]. The international recommendations for sleep duration were mostly based on studies involving self-report[7,66], and self-reported sleep is the most relevant variable for clinical, public health and policy recommendations[7]. While acknowledging the limitations of self-reports, we also believe them to be the most relevant measure in the present context. Second, we studied morphometric brain measures only. Although other measures could show different sensitivity to sleep duration, such as white matter microstructure[109] or Aβ accumulation[110], brain morphometry is sensitive to normal and pathological brain changes[19], and atrophy has consistently been identified as a factor governing age-related sleep changes[111]. Third, we have not considered cognitive function, for which different ranges of sleep duration may be associated with the highest scores. It is still likely that associations between sleep duration and cognitive performance or mental health are transient and reversible after restorative sleep, whereas associations with brain structure may be more permanent. Fourth, the samples were not thoroughly screened for sleep disorders such as sleep apnea[112]. If individuals with sleep problems were included, this would probably not attenuate the relationships and is therefore unlikely to explain the weak sleep–brain associations observed in the study. Fifth, to study differential genetic influences on sleep duration in participants with different habitual sleep patterns, we stratified the sample by seven hours, a strategy that made our GWAS underpowered. While the findings are promising, large-scale independent validation is needed. Furthermore, given the limited power of the GWAS, the relations suggested by the MR analysis will also need future replication. Sixth, we had no quantitative measure of head motion, so to the extent that head motion is correlated with sleep duration, this could be a confounder. Seventh, a number of covariates that could influence the sleep–brain relationships were not controlled for, including cardiovascular risk factors other than BMI. Finally, although some of the samples are population based, no MRI study is fully representative of the population from which it is sampled. Despite including studies from multiple European countries and the United States, we cannot exclude the possibility that different sleep–brain patterns might exist in other populations.

## Conclusion

We did not find evidence suggesting that sleep duration was related to the rate of atrophy or that sleep shorter than the recommended duration[1,66] was associated with smaller regional brain volumes, thinner cortex or smaller ventricles. Rather, sleeping less than the recommended amount was associated with thicker cortex and greater regional brain volumes relative to ICV, and moderately long sleep showed a stronger association with smaller volumes than even very short sleep (for example, less than five hours). As the average sleep duration was almost perfectly aligned with the duration associated with the largest volumes, this may suggest that normal brains promote adequate sleeping patterns, which are shorter than the current recommendations.

## Methods

### Transparency

The current work contains many analyses and analytic choices, which may affect the results. These include, for instance, which covariates are included in the different analyses, the exclusion of outliers and restriction of data ranges (for example, for sleep duration), model specifications and model selection. This information is too extensive to fit in the main text. To optimize transparency, we have included these details in the Supplementary Information (an overview is provided in Supplementary Table 3).

### Sample

Community-dwelling participants were recruited from multiple countries in Europe and the United States. Some were convenience samples, whereas others were contacted on the basis of population registries. All participants at the age of majority gave written informed consent. All procedures were approved by a relevant ethics review board. For Lifebrain, approval was given by the Regional Ethical Committee for South Norway, and all sub-studies were approved by the relevant national review boards. For UKB, ethics approval was obtained from the National Health Service National Research Ethics Service (ref. no. 11/NW/0382).

In total, data from 47,039 participants (20.0–89.4 years) with information about sleep duration and MRI of the brain were included. For 3,910 participants, two or more MRI examinations were available, yielding a total of 51,320 MRIs (mean follow-up interval, 2.5 years; range, 0.005–11.2 years; 26,811 female and 24,509 male observations). The demographics of the samples are given in Table 1, and a brief description of each is given below (for the details, see the Supplementary Information, 'Sample characteristics').

### Sleep measures

For the Human Connectome Project and the Lifebrain samples except Betula, sleep duration and other characteristics were measured by the Pittsburgh Sleep Quality Index (PSQI)[113]. For Betula, sleep characteristics were measured by the Karolinska Sleep Questionnaire[114,115], which can be used to extract the same information covered by the PSQI[116]. For UKB, sleep was measured through multiple questions. For all samples except UKB, we calculated the PSQI global score following normal procedures but excluded the sleep duration component. For UKB, we calculated a sum score of different sleep-related measures (sleeplessness (field 1200), problems getting up in the morning (field 1170), daytime dozing (field 1220), snoring (field 1210) and chronotype (field 1180)). This global sleep quality score was used as a covariate in follow-up analyses of brain–sleep duration relationships.

### MRI

The Lifebrain MRI data originated from seven different scanners (for the details, see ref. 26 and the Supplementary Information, 'MRI methods'), processed with FreeSurfer version 6.0 (https://surfer.nmr.mgh.harvard.edu/)[117–120]. Because FreeSurfer is almost fully automated, to avoid introducing possible site-specific biases, we imposed gross quality control measures and did no manual editing. To assess the

influence of the scanner on volumetric estimates, seven participants were scanned on seven scanners across the consortium sites (see ref. 26 for the details). Using the hippocampus as the test region, there was a significant main effect of the scanner on volume ($F = 4.13$, $P = 0.046$), but the between-participant rank order was close to perfectly retained between scanners, with a mean between-scanner Pearson correlation of $r = 0.98$ (range, 0.94–1.00). Analyses of five additional volumetric cortical and subcortical ROIs (medial temporal lobe (entorhinal and parahippocampal cortex), precuneus, superior temporal, caudate nucleus and caudal middle frontal) showed correlations close to 1.0 for all regions except medial temporal lobe, where correlations were somewhat lower but still more than $r = 0.75$ (ref. 121). Thus, including site as a random effect covariate in the analyses is probably sufficient to remove the influence of scanner differences.

UKB participants were scanned using three identical Siemens 3T Prisma scanners (UKB Brain Imaging–Acquisition Protocol (https://www.fmrib.ox.ac.uk/ukbiobank/protocol/)). FreeSurfer outputs[122] and the volumetric scaling from T1 head image to standard space as a proxy for ICV were used in the analyses, generated using publicly available tools, primarily based on FSL (FMRIB Software library, https://fsl.fmrib.ox.ac.uk/fsl/fslwiki). The details of the imaging protocol (http://biobank.ctsu.ox.ac.uk/crystal/refer.cgi?id=2367) and structural image processing are provided on the UKB website (http://biobank.ctsu.ox.ac.uk/crystal/refer.cgi?id=1977).

## Statistical analyses

ROI analyses were run in R version 4.0.0 (ref. 123), by GAMMs using the packages gamm4 version 0.2–26 (ref. 124) and mgcv version 1.8–28 (ref. 62). GAMMs offer an attractive alternative to linear mixed models in that a priori specifications of polynomial functional forms are not necessary, and GAMMs are able to accurately fit trajectories of different forms and complexities[63]. The Desikan–Killiany parcellation included in FreeSurfer yields 34 regions, but the temporal and the frontal poles were excluded from analysis due to substantial noise in these regions. This atlas was selected because it is well validated and commonly used for cortical ROI-based analyses, which is a benefit when comparing results across studies. Volumetric outliers were defined by having a residual more than four times the magnitude of the residual standard error in an analysis of age effects and removed from the analyses. The FDR was used to adjust the $P$ values for multiple comparisons, because family-wise error correction methods such as Bonferroni are very strict and would lead to serious loss of power. With FDR methods, we know that the expected proportion of false discoveries is 0.05, which we consider acceptable. Hence, we used the Benjamini–Hochberg procedure to adjust the $P$ values. The set of all subcortical ROIs constrained one family of tests, and the set of all cortical ROIs (for the measure of thickness) constrained another family of tests. The computer code can be found in the Supplementary Information in the relevant sections.

## Genetic analyses

**GWAS.** Five GWAS analyses were performed: sleep duration for participants sleeping ≤7 hours ($N = 197,137$) and >7 hours ($N = 112,839$), separately, using a sample with no MRI data available; total hippocampal volume ($N = 29,155$); TGV ($N = 29,155$); and estimated ICV ($N = 29,155$). For each GWAS, sex, baseline age and the top ten genetic principal components were included as covariates in a linear regression model for identifying associate SNPs for each trait. In addition, each of the five traits was first normalized to have unit variance and zero mean. For total hippocampal volume and TGV, ICV was additionally included as a covariate. PLINK version 2.0 (ref. 125) was used for these analyses with the function glm. The FUMA server[126] was used to annotate the GWAS results to genomic regions and nearby genes with the default parameters. Additional details, Manhattan plots and QQ plots showing the GWAS results are presented in the Supplementary Information, 'Genetic analyses'.

GWAS were run instead of using summary statistics from previous genetic studies of sleep in UKB (for example, ref. 51) for three reasons: (1) we needed to ensure that we were using completely non-overlapping samples for the sleep and the sleep-MRI analyses; (2) we were interested in contrasting participants with below- versus above-average sleep duration and studying the variation within each group, which has not previously been done; and (3) an important aim is to assess whether there are plausible causal relations between sleep duration and brain structure, using PGS and MR methods. The widely used models for these methods assume monotonic relationships, where effects do not change direction across the range of phenotypic values. This does not fit the inverse U-shaped relationship between sleep duration and brain features. Thus, we could not use previously published summary statistics, particularly for the MR analysis.

**SNP-$h^2$ and genetic correlation.** The LD score regression model (ldsc)[77] was used to estimate SNP-$h^2$ for each trait and genetic correlation between sleep duration and hippocampal volume, TGV and ICV. LD structure provided by ldsc from the HapMap 3 data was used in this analysis. The other parameters of ldsc were set to its default values.

**PGS.** To accurately estimate the PGSs for a trait, we first computed the posterior effect size per SNP using the Bayesian mixture model implemented in PRS-CS4 (ref. 127). The polygenic risk score via continuous shrinkage priors (PRS-CS) model is a widely used method for computing PGSs for highly polygenic traits[127]. PRS-CS shrinks effect sizes estimated from GWAS using LD correlations in a Bayesian framework, assuming a two-component mixture prior distribution. The LD correlations provided by PRS-CS were based on the 1000 Genomes phase 3 European population. In total, about 1.3 million high-quality SNPs were used. In addition, PRS-CS does not need information from the target sample where the estimated posterior effect will be used for computing PGSs. The GWAS sample and the target sample were thus treated fully independently in the PGS computation. Furthermore, in light of previously published GWAS results as well as ours, we assumed a highly polygenic genetic architecture for both MRI-derived traits and sleep duration, by setting the parameter $\varphi$ to 0.01, instead of a grid-search strategy proposed by the model. We believe that our choice, though conservative, further reduces the overfitting risk. For the other parameters in PRS-CS, we used the default values.

The posterior effect sizes obtained by running PRS-CS on each GWAS were then used separately to compute PGSs. We did not use $P$ values or LD thresholds to select SNPs. Rather, genome-wide SNPs were used for computing the PGSs. After removing rare variants (MAF < 0.01) in UKB, variants not in the HapMap 3 data and variants that are not in Hardy–Weinberg equilibrium ($P < 10^{-6}$), we used the remaining 615,297 SNPs for computing the PGSs for each trait. Recent methodology studies all point to the advantage of using shrinkage-based methods over $P$-value-based thresholding methods (for example, LDpred2 (ref. 128), PRS-CS and the lasso-based models[129]), particularly for highly polygenic traits. The computed posterior effects were used as weights in the computation of PGSs for a trait by using the score function from PLINK version 2.0. To examine the associations between PGSs for a trait with a second trait, linear regression models were used. The same covariates included in the GWAS analysis were included as covariates in addition to PGSs in these models.

The PRS-CS[127] software, which implements Bayesian mixture models to incorporate LD structures in estimating allele effect sizes, was used for PGS analysis. High-quality SNPs provided by PRS-CS derived from the HapMap 3 dataset were used for constructing LD structures. The polygenicity parameter ($\varphi$) was set to 0.01, assuming a highly polygenic trait. Estimated effect sizes were used to compute PGSs using the score function from PLINK version 2.0 without further selection through association $P$ values or LD $r^2$ values.

**Two-sample MR.** The TwoSampleMR R package[130] was used to investigate the relations between sleep duration and the brain variables. Independent instrumental SNPs were selected using the following parameters: association $P \leq 10^{-6}$, MAF $\geq 0.05$, LD $r^2 \leq 0.1$ and LD distance = 10 kb. The LD structure was derived from 10,000 independent European participants randomly selected from UKB. The powerful inverse variance weighted model from TwoSampleMR was used as the main model. Other models implemented in the software were also run as sensitivity analysis. To further support the results, the analysis was reperformed with $P \leq 10^{-5}$, which would increase the strength of instrumenting for the less powerful sleep duration traits. For the only significant relation—that is, ICV to sleep duration for shorter-than-average sleepers—a third analysis with $P \leq 5 \times 10^{-8}$ used for selecting instrumental SNP was performed. The standard output from TwoSampleMR is shown in the Supplementary Information, 'Genetics notes'.

### Reporting summary

Further information on research design is available in the Nature Portfolio Reporting Summary linked to this article.

### Data availability

The data supporting the results of the current study are available through requests to the principal investigators of each sub-study, given appropriate ethics and data protection approvals. Specific limitations on data access apply to some samples. Contact information can be obtained from the corresponding authors. UKB data requests can be submitted to http://www.ukbiobank.ac.uk.

### Code availability

The R code for the statistical analyses is provided in the Supplementary Information.

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

## Acknowledgements

The Lifebrain consortium is funded by the EU Horizon 2020 grant agreement no. 732592 (Lifebrain). The different sub-studies are supported by different sources. LCBC is supported by the European Research Council under grant agreements no. 283634 and no. 725025 (to A.M.F.) and no. 313440 (to K.B.W.), as well as the Norwegian Research Council (to A.M.F. and K.B.W.) and the National Association for Public Health's dementia research programme, Norway (to A.M.F.). Betula is supported by a scholar grant from the Knut and Alice Wallenberg foundation to L.N. Barcelona is partially supported by a Spanish Ministry of Economy and Competitiveness grant to D.B.-F. (grant no. PSI2015-64227-R (AEI/FEDER, UE)); by the Walnuts and Healthy Aging study (http://www.clinicaltrials.gov; grant no. NCT01634841) funded by the California Walnut Commission, Sacramento, California; and by an ICREA Academia 2019 award. BASE-II has been supported by the German Federal Ministry of Education and Research under grant nos 16SV5537, 16SV5837, 16SV5538, 16SV5536K, 01UW0808, 01UW0706, 01GL1716A and 01GL1716B and by the European Research Council under grant agreement no. 677804 (to S.K.). Work on the Whitehall II Imaging Substudy was mainly funded by Lifelong Health and Well-being Programme grant no. G1001354 from the UK Medical Research Council ('Predicting MRI Abnormalities with Longitudinal Data of the Whitehall II Substudy') to K.P.E. The Wellcome Centre for Integrative Neuroimaging is supported by core funding from award no. 203139/Z/16/Z from the Wellcome Trust. The data were provided (in part) by the Human Connectome Project, WU-Minn Consortium (principal investigators: D. Van Essen and K. Ugurbil; 1U54MH091657), funded by the 16 NIH Institutes and Centers that support the NIH Blueprint for Neuroscience Research; and by the McDonnell Center for Systems Neuroscience at Washington University. Part of the research was conducted using the UKB resource under application no. 32048. The funders had no role in study design, data collection and analysis, decision to publish or preparation of the manuscript.

## Author contributions

A.M.F. wrote the first draft. Y.W. performed the genetic analyses, and Ø.S. conducted the other statistical analyses. I.K.A. and A.M.M. organized the data. D.B.-F., K.P.E., L.N., C.S.-P., L.O.W., L.B., A.M.F. and K.B.W. contributed data. Ø.S., Y.W., I.K.A., W.F.C.B., D.B.-F., L.B., C.-J.B., A.M.B., I.D., C.A.D., K.P.E., P.G., R.K., S.K., K.S.M., A.M.M., L.N., C.E.S., C.S.-P., D.V.-P., G.W., L.O.W. and K.B.W. critically reviewed earlier versions of the manuscript and approved the final version.

## Competing interests

C.E.S. reports consulting fees from Jazz Pharmaceuticals and is now a full-time employee of the US Alzheimer's Association. C.A.D. is a cofounder, stock owner, board member and consultant in the contract laboratory Vitas AS, performing personalized analyses of blood biomarkers. The rest of the authors report no competing interests.

## Additional information

**Correspondence and requests for materials** should be addressed to Anders M. Fjell.

[1]Center for Lifespan Changes in Brain and Cognition, University of Oslo, Oslo, Norway. [2]Department of Radiology and Nuclear Medicine, Oslo University Hospital, Oslo, Norway. [3]Danish Research Centre for Magnetic Resonance, Centre for Functional and Diagnostic Imaging and Research, Copenhagen University Hospital—Amager and Hvidovre, Copenhagen, Denmark. [4]Departament de Medicina, Facultat de Medicina i Ciències de la Salut, Universitat de Barcelona, Barcelona, Spain. [5]Institut de Neurociències, Universitat de Barcelona, Barcelona, Spain. [6]Institut d'Investigacions Biomèdiques August Pii Sunyer, Barcelona, Spain. [7]Lübeck Interdisciplinary Platform for Genome Analytics, University of Lübeck, Lübeck, Germany. [8]Umeå Center for Functional Brain Imaging, Umeå University, Umeå, Sweden. [9]Department of Radiation Sciences, Diagnostic Radiology, Umeå University, Umeå, Sweden. [10]Institute of Sports Medicine Copenhagen, Copenhagen University Hospital Bispebjerg, Copenhagen, Denmark. [11]Center for Lifespan Psychology, Max Planck Institute for Human Development, Berlin, Germany. [12]Department of Psychology, MSB Medical School Berlin, Berlin, Germany. [13]Department of Endocrinology and Metabolic Diseases (including Division of Lipid Metabolism), Biology of Aging Working Group, Charité—Universitätsmedizin Berlin, corporate member of Freie Universität Berlin and Humboldt-Universität zu Berlin, Berlin, Germany. [14]Berlin Institute of Health Center for Regenerative Therapies, Berlin Institute of Health at Charité—Universitätsmedizin Berlin, Berlin, Germany. [15]Vitas AS, Oslo, Norway. [16]Department of Nutrition, Institute of Basic Medical Sciences, Faculty of Medicine, University of Oslo, Oslo, Norway. [17]Department of Psychiatry, University of Oxford, Oxford, UK. [18]Faculty of Psychology and Educational Sciences, University of Geneva, Geneva, Switzerland. [19]UniDistance Suisse, Brig, Switzerland. [20]Swiss National Centre of Competence in Research LIVES, University of Geneva, Geneva, Switzerland. [21]Cognitive Neuroscience Department, Donders Institute for Brain, Cognition and Behavior, Radboud University Medical Center, Nijmegen, the Netherlands. [22]Department of Psychiatry and Psychotherapy, University Medical Center Hamburg-Eppendorf, Hamburg, Germany. [23]Radiography, Department of Technology, University College Copenhagen, Copenhagen, Denmark. [24]Global Brain Health Institute, Department of Neurology, University of California, San Francisco, San Francisco, CA, USA. [25]Wellcome Centre for Integrative Neuroimaging, University of Oxford, Oxford, UK. [26]Alzheimer's Association, Chicago, IL, USA. [27]Department of Psychiatry and Psychotherapy, Jena University Hospital, Jena, Germany. [28]Oslo Delirium Research Group, Department of Geriatric Medicine, University of Oslo, Oslo, Norway. [29]Department of Geriatric Medicine, Akershus University Hospital, Lørenskog, Norway. ✉e-mail: andersmf@psykologi.uio.no

# Reporting Summary

## Statistics

For all statistical analyses, confirm that the following items are present in the figure legend, table legend, main text, or Methods section.

| n/a | Confirmed | |
|---|---|---|
| ☐ | ☒ | The exact sample size ($n$) for each experimental group/condition, given as a discrete number and unit of measurement |
| ☐ | ☒ | A statement on whether measurements were taken from distinct samples or whether the same sample was measured repeatedly |
| ☐ | ☒ | The statistical test(s) used AND whether they are one- or two-sided<br>*Only common tests should be described solely by name; describe more complex techniques in the Methods section.* |
| ☐ | ☒ | A description of all covariates tested |
| ☐ | ☒ | A description of any assumptions or corrections, such as tests of normality and adjustment for multiple comparisons |
| ☐ | ☒ | A full description of the statistical parameters including central tendency (e.g. means) or other basic estimates (e.g. regression coefficient) AND variation (e.g. standard deviation) or associated estimates of uncertainty (e.g. confidence intervals) |
| ☐ | ☒ | For null hypothesis testing, the test statistic (e.g. $F$, $t$, $r$) with confidence intervals, effect sizes, degrees of freedom and $P$ value noted<br>*Give P values as exact values whenever suitable.* |
| ☒ | ☐ | For Bayesian analysis, information on the choice of priors and Markov chain Monte Carlo settings |
| ☐ | ☒ | For hierarchical and complex designs, identification of the appropriate level for tests and full reporting of outcomes |
| ☐ | ☒ | Estimates of effect sizes (e.g. Cohen's $d$, Pearson's $r$), indicating how they were calculated |

*Our web collection on statistics for biologists contains articles on many of the points above.*

## Software and code

Policy information about availability of computer code

| Data collection | No software was used for data collection for the current study. |
|---|---|
| Data analysis | FreeSurfer v6.0; R version 4.0.0 ("gamm4" version 0.2-26, "mgcv" version 1.8-28, "ggseg"), PLINK2 |

For manuscripts utilizing custom algorithms or software that are central to the research but not yet described in published literature, software must be made available to editors and reviewers. We strongly encourage code deposition in a community repository (e.g. GitHub). See the Nature Portfolio guidelines for submitting code & software for further information.

## Data

Policy information about availability of data

All manuscripts must include a data availability statement. This statement should provide the following information, where applicable:
- Accession codes, unique identifiers, or web links for publicly available datasets
- A description of any restrictions on data availability
- For clinical datasets or third party data, please ensure that the statement adheres to our policy

Data supporting the results of the current study are available by requests to the PIs of each sub-study, given appropriate ethics and data protection approvals. Specific limitations on data access applies to some samples. Contact information can be obtained from the corresponding authors. UK Biobank data requests can be submitted to http://www.ukbiobank.ac.uk.

# Research involving human participants, their data, or biological material

Policy information about studies with [underline]human participants or human data[/underline]. See also policy information about [underline]sex, gender (identity/presentation), and sexual orientation[/underline] and [underline]race, ethnicity and racism[/underline].

| | |
|---|---|
| Reporting on sex and gender | Sex was included as covariate in the analyses. In addition, post hoc analyses were run split by sex to assess whether specific relationships were seen for males or females, and a formal sex-interaction analysis was further run directly to test effects of sex. |
| Reporting on race, ethnicity, or other socially relevant groupings | For the genetic analyses, participants who were not self-reported white-British were excluded. In addition, the top 10 genetic principal components were included as covariates to control for population structure. The main statistical analyses were repeated controlling for socioeconomic status, based on self-reported income and education. No other social, race or ethnicity variables were used. |
| Population characteristics | Community-dwelling participants from multiple countries in Europe and the US. Some were convenience samples, whereas others were contacted based on population registries. In total, data from 47,039 participants (20.0-89.4 years) with information about sleep duration and MRI of the brain were included. For 3,910, two or more MRI examinations were available, yielding a total of 51,320 MRIs (mean follow-up interval 2.5 years, range 0.005-11.2, 26,811 female/ 24,509 male observations). |
| Recruitment | Sample<br>Community-dwelling participants were recruited from multiple countries in Europe and the US. Some were convenience samples, whereas others were contacted based on population registries. No MRI sample is fully representative of the populations from which they are drawn. Which effects this may have on the results are unknown. For all reported results, relevant population characteristics such as age, sex, education, income, BMI and depression symptoms were covaried. |
| Ethics oversight | All procedures were approved by a relevant ethics review board. For Lifebrain, approval was given by the Regional Ethical Committee for South Norway, and all sub-studies were approved by the relevant national review boards. For UKB, ethics approval was obtained from the National Health Service National Research Ethics Service (Ref 11/NW/0382). |

Note that full information on the approval of the study protocol must also be provided in the manuscript.

# Field-specific reporting

Please select the one below that is the best fit for your research. If you are not sure, read the appropriate sections before making your selection.

☒ Life sciences          ☐ Behavioural & social sciences          ☐ Ecological, evolutionary & environmental sciences

For a reference copy of the document with all sections, see [underline]nature.com/documents/nr-reporting-summary-flat.pdf[/underline]

# Life sciences study design

All studies must disclose on these points even when the disclosure is negative.

| | |
|---|---|
| Sample size | We used all available data points to maximize sample size, so this was not defined before the study. Sample size much larger than any existing study, yielding excellent statistical power. |
| Data exclusions | Exclusion criteria were predefined, and a detailed description is provided in the manuscript/ SI. Volumetric outliers were defined by having a residual more than four times the magnitude of the residuals standard error in an analysis of age effects and removed from the analyses. Each individual study feeding data into this work used different exclusion criteria before data were entered into the present study, detailed in the manuscript. For genetic analyses, we excluded participants who are not self-reported white-British (n=92.900), have relatives in the biobank (n=148.689), had been labeled as outliers in missingness or heterozygosity (n=968) or had conflicting self-reported vs genetic sex (n=378) by the UK Biobank team. |
| Replication | We did not have an independent replication sample. Running analyses on the full ample yielded maximal statistical power to detect miniute effects. As we did not find evidence for a relationship between sleep duration and brain change in our very big sample, replication was deemed unnecessary. Instead, permutation tests were used to assess stability. Results were evaluated based on effect sizes, p-values and confidence intervals. Proper statistical corrections for multiple comparisons were applied. |
| Randomization | This is an observational study, hence randomization is not relevant. |
| Blinding | This is an observational study, hence blinding is not relevant. |

# Reporting for specific materials, systems and methods

We require information from authors about some types of materials, experimental systems and methods used in many studies. Here, indicate whether each material, system or method listed is relevant to your study. If you are not sure if a list item applies to your research, read the appropriate section before selecting a response.

## Materials & experimental systems

| n/a | Involved in the study |
|---|---|
| ☒ | ☐ Antibodies |
| ☒ | ☐ Eukaryotic cell lines |
| ☒ | ☐ Palaeontology and archaeology |
| ☒ | ☐ Animals and other organisms |
| ☒ | ☐ Clinical data |
| ☒ | ☐ Dual use research of concern |
| ☒ | ☐ Plants |

## Methods

| n/a | Involved in the study |
|---|---|
| ☒ | ☐ ChIP-seq |
| ☒ | ☐ Flow cytometry |
| ☐ | ☒ MRI-based neuroimaging |

# Magnetic resonance imaging

## Experimental design

| | |
|---|---|
| Design type | Structural (T1w) |
| Design specifications | Structural (T1w) |
| Behavioral performance measures | No task during scanning |

## Acquisition

| | |
|---|---|
| Imaging type(s) | Structural (T1w) |
| Field strength | 1.5T and 3.0T |
| Sequence & imaging parameters | BASE-II Tim Trio Siemens 3.0 TR: 2500 ms, TE: 4.77 ms, TI: 1100 ms, flip angle: 7°, slice thickness: 1.0 mm, FoV 256×256 mm, 176 slices<br>Betula Discovery GE 3.0 TR: 8.19 ms, TE: 3.2 ms, TI: 450 ms, flip angle: 12°, slice thickness: 1 mm, FOV 250×250 mm, 180 slices<br>Cam-CAN Tim Trio<br>Siemens 3.0 TR: 2250 ms, TE: 2.98 ms, TI: 900 ms, flip angle: 9°, slice thickness 1 mm, FOV 256×240 mm, 192 slices<br>LCBC Avanto Siemens 1.5 TR: 2400 ms, TE: 3.61 ms, TI: 1000 ms, flip angle: 8°, slice thickness: 1.2 mm, FoV: 240×240 m, 160 slices, iPat = 2<br> Avanto Siemens 1.5 TR: 2400 ms, TE = 3.79 ms, TI = 1000 ms, flip angle = 8, slice thickness: 1.2 mm, FoV: 240 x 240 mm, 160 slices<br> Skyra Siemens 3.0 TR: 2300 ms, TE: 2.98 ms, TI: 850 ms, flip angle: 8°, slice thickness: 1 mm, FoV: 256×256 mm, 176 slices<br> Prisma Siemens 3.0 TR: 2400 ms, TE: 2.22 ms, TI: 1000 ms, flip angle: 8°, slice thickness: 0.8 mm, FoV: 240×256 mm, 208 slices, iPat = 2<br>UB Tim Trio Siemens 3.0 TR: 2300 ms, TE: 2.98, TI: 900 ms, slice thickness 1 mm, flip angle: 9°, FoV 256×256 mm, 240 slices<br>WH-II Verio Siemens 3.0 TR: 2530 ms, TE: 1.79/3.65/5.51/7.37 ms, TI: 1380 ms, flip angle: 7°, slice thickness: 1.0 mm, FOV: 256×256 mm<br>HCP Connectome<br>Skyra<br>Siemens* 3.0 TR: 2400 ms, TE: 2.14 ms, TI: 1000 ms, flip angle: 8°, slice thickness: 0.7 mm, FOV: 224 mm, 256 slices, GRAPPA = 2<br>UKB Skyra<br>Siemens 3.0 TR: 2000 ms, TI: 880 ms, slice thickness: 1 mm, FoV: 208×256 mm, 256 slices, iPAT=2 |
| Area of acquisition | Whole brain coverage |
| Diffusion MRI | ☐ Used    ☒ Not used |

## Preprocessing

| | |
|---|---|
| Preprocessing software | FreeSurfer 6.0 |
| Normalization | linear, T1 |
| Normalization template | Talairach |
| Noise and artifact removal | To avoid introducing site-specific biases, quality control measures were imposed and no manual editing was done. |
| Volume censoring | Not used (astructural scans only) |

## Statistical modeling & inference

| | |
|---|---|
| Model type and settings | Generalized additive mixed models; Spatio-temporal linear mixed models |
| Effect(s) tested | No tasks or stimulus conditions were used (structural only) |

Specify type of analysis: ☐ Whole brain ☐ ROI-based ☒ Both

| | |
|---|---|
| Anatomical location(s) | Vertex-wise, subcortical ROIs from FreeSurfer, Desikan-Killiany cortical parcellations |

| | |
|---|---|
| Statistic type for inference | Vertex-wise spatio-temporal linear mixed-effects models. |

(See Eklund et al. 2016)

| | |
|---|---|
| Correction | Z Monte Carlo simulations with a cluster forming threshold of p < .01 and a cluster threshold of .05 for vertex-wise analyses; FDR for ROI analyses |

## Models & analysis

| n/a | Involved in the study |
|---|---|
| ☒ | ☐ Functional and/or effective connectivity |
| ☒ | ☐ Graph analysis |
| ☒ | ☐ Multivariate modeling or predictive analysis |

