## [Peer Review File · Nature Human Behaviour]

Peer Review Information

Journal: Nature Human Behaviour

Manuscript Title: No phenotypic or genotypic evidence for a link between sleep duration and brain atrophy

Corresponding author name(s): Anders M Fjell

Reviewer Comments & Decisions:

Decision Letter, initial version:

11th April 2022

Dear Professor Fjell,

Thank you once again for your manuscript, entitled "Sleep duration and brain structure – phenotypic associations and genotypic covariance," and for your patience during the peer review process.

Your manuscript has now been evaluated by 3 reviewers, whose comments are included at the end of this letter. In the light of their advice, I regret that we cannot offer to publish your manuscript in Nature Human Behaviour.

While the reviewers find your work of some interest, they raise concerns about the strength of the novel conclusions that can be drawn at this stage and the appropriateness of the data analysis approaches. We feel that these reservations are sufficiently important as to preclude publication of this work in Nature Human Behaviour.

I am sorry that we cannot be more positive on this occasion but hope that you will find our reviewers' comments helpful when preparing your paper for submission elsewhere.

Sincerely,
Jamie

Dr Jamie Horder
Senior Editor
Nature Human Behaviour

Reviewers' Comments:

Reviewer #1:

Remarks to the Author:

In the current work, Fjell et al., evaluated the phenotypic association and genetic correlation between sleep duration and brain structure. To do so, they analyzed 51295 MRI images from 47039 individuals, and then computed the association between self-reported sleep duration associated with the most relevant local metrics of volume, thickness and ventricle size. Following they performed genetic analysis ?? to evaluate which genes are different between people that sleep long vs short.

Though the topic of sleep and its link with the brain is overall interesting, I have various concerns on the current manuscript that I hope can be of help.

1. In the abstract, the notion 'most favorable brain outcome' is used. This is vague and suggestive language, what would be a perfect brain outcome and what would it mean?
2. Overall the tone of the abstract is a bit odd, for example 'hippocampus showing largest volume at 6.3 hours' or 'not correcting for ICV yielded longer durations associated with maximal volume'. It seems to imply some sort of causality, possibly it is helpful to rephrase, also considering potential non-experts reading this work.
3. I find the terms optimal brain health, as posed in the introduction unnecessary vague and lacking theoretical foundation. Is there any work claiming there is optimal brain health? What if there are various local optima, or not such a thing at all?
4. Regarding the review on sleep duration and brain structure, it is unclear whether this is a complete overview or just a selection? If so, what is the selection based on?
5. The assumption is made that large volumes imply brain health, but is this association so clear cut? What about pruning/thinning of PFC?
6. Why did the authors calculate sleep duration associated with the most favorable (=thick and large?) brain properties and not 'just' focus on the association between both in a more neutral way, given there is no clear evidence of what favorable brain configurations are?
7. The genetic argument comes a bit out of the blue.
8. Figure 1; it would be helpful to see the individual datapoints?
9. Why did they authors opt for the desikan killiany atlas?
10. The cluster analysis is not clearly motivated, why were subcortical regions not incorporated as well?
11. Why were associations between vertex-wise brain and sleep links computed differently for long and short sleepers? Why was not also the interaction between age and sleep for example assessed but rather a young vs old model?
12. Minor point: in figure three below the statistical bar some letter seems to be only half removed
13. Doesn't sleep vs maximal volume analysis imply similar trajectories for different regions across ageing, yet this isn't the case?
14. The genetic analysis at the end give a somewhat additive impression, and the link with the previous steps are not that clear. Ideally this could be clarified a bit better? For example, is there a different association with the 3 clusters from the beginning of result section?
15. 'One interpretation is that there is a propensity towards optimal sleep duration for the brain' this is mentioned in the discussion but it seems the primary interpretation/driver of analysis of the authors

16. Overall, I believe the writing style is suggestive of causality whereas the data is correlational, possibly this may be something to edit?

17. In the end a comparison to recommendations of sleep length is made, but I believe the current study set-up is not fit for such a statement due to the study set-up?

Reviewer #2:

Remarks to the Author:

Key Results

Over close to 50,000 people, sleep durations around 6.8 hours were linked with the largest overall brain volumes. There were regional variations, with the hippocampus showing highest volume related to 6.3 hours of sleep, and cortical regions showing highest volumes with over 7 hours of sleep.

Originality and significance

This is a very large study in the field of brain measurements, giving a strong confidence in the reliability of the results. Specifically, the fact that sleep of less than 7 hours was associated with the highest brain volumes can be trusted. This finding raises the question of whether 7 hours as a recommended minimum is appropriate. The large study also gives reassurance that the substantial variations across brain regions of hours of sleep related to maximum volume are indeed present, and allows us to consider what might be the source of such variations. The findings challenge us to consider the relationship between "brain health" as a physiology and brain volume, on both a whole-brain and regional level.

I am less sure of the meaning of the genetic analyses; these seem completely unconnected in the analysis from brain volume.

I have little confidence in the significance of the findings from categorizing people who sleep less or more than 7 hours as shorter or longer sleepers; it seems these authors' own results suggest there is a range of sleep times that are healthy, so no variation related to poor health would be expected in that range – this idea is supported by much literature in the sleep field. Indeed, taking a slightly controversial view, one might argue that the present findings are a more reliable reflection of healthy sleep range...at least for the middle quartile, since one would assume that 25% of people would not have pathological sleep durations. In any case, most of the evidence only shows clear pathology (or poor health) at much shorter sleep times than 7 hours.

Conceptual & methodological novelty

Looking at a simple question in a way that confidently provides an answer is arguably a conceptual novelty. Certainly in the field of neuroimaging there are few large N studies, which also limits the types of appropriate analyses that can be performed, for example such as allowing for non-linear relationships between sleep and brain volume.

I do think conceptually the manuscript could do with a little explanation of how brain health can be reflected in brain volume, meaning explaining possible biological states and processes that lead to higher or lower brain volume.

Relatedly, it would be nice to read a brief explanation of how short sleep is associated with poor health in conditions that are known to affect the brain (e.g., insomnia, depression) and similarly for long sleep (e.g., atypical depression subtype, sleep apnea). There is a brief mention in the discussion, but it would help to explain in the Introduction why brain health can be reflected in brain volume. Inflammation, atrophy, learning/neuroplasticity are relevant concepts, to name a few.

I am not familiar with genetic analyses; I would have liked to see them related to brain volume, but as it stands I did not grasp the conceptual link between the genetics and brain volume or even brain health.

Data and methodology

Unique, outstanding data reflecting a chance to truly answer the research question with confidence. The basic methods as described appear to be appropriate, well documented, and replicable. I would like to have seen more reference in the manuscript text to quality control, since this is a key component of analyzing large datasets. Additionally, the manuscript text did not give much detail on the way the segmentations were performed.

Manuscript - General

Assumption that thicker cortex reflects better brain health is true on average, but there are unhealthy states (inflammation) that result in larger brain tissue volume. For example, is it possible that the caudate estimate of 11.8 hours for maximum volume reflects inflammation?

The premise that there is an optimal sleep time for brain health is challenged by the fact that different brain areas show different hours of sleep for maximum volume. Are the authors saying 6.3 hours leads to optimum hippocampal health but 6.8 hours leads to optimal cortical health? If so what sleep mechanism could differentially impact those two brain regions?

I do not see how the genetic analysis fits in to the premise of the article. It seems to be an afterthought, or an independent study.

I think it would improve readability to replace "below average sleep duration group" with "short sleepers", and "above average sleep duration group" with "long sleepers"

"It is a worry" – one example that demonstrates the need for an English language review for appropriate style. Most of the writing is good.

Abstract

Needs clarifying, since this is the most read part of any paper.

Need link between sufficient sleep and brain – just being explicit that "breath health as reflected by larger brain volume"

Change "favorable" and define

Genetics not introduced, and genetic results presented without context in relation to the brain. If there was no independent analysis of genetic vs brain associations, does this belong here?

The short/long sleep sentence is confusing. First off, it is unclear what a short or long sleeper is – presumably outside the 95% CI, but that is not clear. Secondly, I think the point is that one gene both

lengthened sleep in short sleepers, and shortened sleep in long sleepers – but how was that determined in a cross-sectional study? How can a person be defined being 1) a short sleeper and 2) having their sleep longer than in others with short sleep? It's just not clear. In the Results, there is "below average", but in the sleep field "short sleepers" are people with short sleep (for example threshold of <5, < 6 and < 7 hours have been used by various groups). Further in the Results, there is a section defining shorter and longer sleepers as below of above 7 hours, but we would really expect differences only in people outside a range, say perhaps the 95% CI.

Mendelian randomization unclear to most readers

Unclear how "habitual" short or long sleep was measured, so conclusion does not follow (related to above point about genetics).

Introduction

Seems to be suggesting AD is distinct from brain health, whereas AD could be considered an example of poor brain health

We can assume that sleep is essential

Results

This may be a journal style issue, but there are several sections where the methodology is described; should this be moved to the Methods section, which is very short at the moment?

Table 2 and related text – the abstract mentions ventricle to ICV ratio; is this what is being referred to here in the Results? Or are these results based on absolute volumes?

Figure 3 – the caption could better explain the relationships, since they are a bit confusing as a negative correlation in the below-average sleep duration means something different to the negative correlation in the above-average sleep duration analyses (at least assuming the bigger volume is better). It would also help to have the N and the definition of below-average sleep duration

Figure 7 – font sizes and labels need tidying; for example, legend does not explain "SleepLe7"

Methods

I'm not sure what this journal convention is, but it seems like there should be something about Mendelian randomization analyses in the manuscript; however, there is much methodology in the Results at present.

Discussion

Several comments above relate to the Discussion.

For example, in Conclusions, the term "Short sleep" has many interpretations and needs to be clarified

Reviewer #3:

Remarks to the Author:

Thank you for the opportunity to review this paper by Fjell and colleagues. It has some interesting analyses and the authors have clearly done a lot of work. I have mainly focused on the genetic aspects of the work and have some concerns that I detail below.

Minor comments:

1. Introduction: I'd take out the 'It is a worry' from the first sentence, as it's a bit of an odd opener for a scientific paper.
2. Are the authors defining brain health as only morphology (i.e., measures from MRI)? Or do they also consider for example, cognitive function and decline? If so, there is one large-scale Mendelian randomisation study with both linear and non-linear analyses showing a causal relationship between sleep duration and worse cognitive function (Henry et al, 2019, IJE).
3. I think the authors should not talk about their study until the end of the Introduction, as this is not conventional.
4. It is fine to cite studies for heritability and polygenic influences on sleep duration, but I think the authors need to clarify that the SNP heritability of sleep duration in GWA studies is very modest.
5. Why the specific choice of covariates in the post-hoc analyses? Why BMI, SES and depression?
6. Please avoid phrases like 'nominally significant effect' and all round I'd avoid 'significant' when talking about associations, even if causal.
7. Relatedly, can the authors present coefficients/effects with 95% CIs, instead of the beta/SE/p-value? Then we can focus on the size of the coefficient, direction and precision by seeing the 95% CIs and not focus on p-values.
8. I know it might seem obvious to some readers (including me), but I think that perhaps a solid justification for the MR analyses might be worth including - i.e., they wanted to try and understand causality and directionality in this case.

1. Major comments:

2. I have not often seen individuals grouped into 'short' and 'long' sleepers and classified as ≤ 7 h and > 7 h. Can the authors provide some justification for these categories? If the authors are following the US NSF guidelines, then they suggest that adults sleep between 7-9h and older adults sleep between 7-8h. This implies that a sleep duration of > 7 h is not long sleep?
3. Can the authors comment on/acknowledge potential issues with analysing a sample with such a wide age range? 20-89y is a very wide age range, especially for these sorts of phenotypes, such as sleep duration which changes throughout the life course. Heritability, for example, changes with age for certain complex traits (e.g., BMI) so would we not expect this perhaps for sleep duration and some of the brain morphology traits?
4. GWAS analyses: I am guessing, although this is not wholly clear from the description of the methods, that the authors decided to run their own GWAS for each trait because if they had only used summary statistics from published GWA studies they would not be able to discern which individuals had/had not undergone MRI scans? I'm not sure what else would be the justification for performing these GWASs, especially for sleep duration, given that large-scale UKB published GWASs exist (i.e., Dashti 2019, etc).
5. Polygenic risk score analyses: what is PRS-CS? The reference doesn't seem to match this. Can the authors define and explain what this is? Am I right that the authors derived their own internal weights and used these to create their PRSs? If so, I'm not sure this is the best approach, as it is usually

recommended that the base/training sample is different from the target/analytical sample to avoid overfitting. It is also not clear what p-value threshold was used for the PRSs, or what LD thresholds were used.

6. Mendelian randomisation analyses/methods:

- a. Instrument selection approach: this does not seem very conventional - usually instruments are chosen that meet genome-wide significance and if this is not the case then there needs to be a strong rationale for it. There are numerous papers that have used sleep duration SNPs as instruments in MR and I don't recall any of them using this approach. For the structural brain MRI instruments there are also published GWAS but these SNPs are perhaps less robust than the sleep ones (although they also have caveats, of course).
- b. Did the authors clump by LD using their own sample as the reference? I don't think this is common either, unless that is not what they did, but this is not clear from the text.
- c. What is the purpose of the sensitivity analyses with an even larger p-value threshold? This is likely to just introduce noise into the instrument.
- d. To be clear, the authors' main MR analyses are one-sample, is this correct? They then perform 2-sample MR analyses but as a sort of side thing...
- e. While I appreciate that this is a long paper with quite a lot of different types of analyses in it, I think that there needs to be a proper description of the MR results, particularly the fact that there is no mention of the 2-sample MR results in the main text, but this is the most popular way to do MR nowadays. There also needs to be some text telling the readers why the authors did what they did regarding for example, where the MR-Egger intercept p-value was <0.05 they then did some leave-one-out analyses (I'm assuming this is what this was at least) and perhaps then comment on what happened with the Egger intercept p-value after these analyses. They also do not comment anywhere on the inclusion of the heterogeneity statistics
- f. Can the authors also (instead of making the reader go to the rather large table in supplementary files) simply tell us what the key F-statistics and R^2 values were for their main instruments? These metrics for genetic instruments in MR are important and should be reported as standard in the text. The total instrument strength (R^2) provides information on how well powered the analyses are likely to be.
- g. When describing the 2-sample MR methods, the authors say that they used some popular methods, which is fine, but the main estimate is the IVW and the other methods such as Egger and WME are sensitivity tests for unbalanced horizontal pleiotropy.

Following suitable revisions, you may want to consider transferring your manuscript. To transfer your manuscript to Scientific Reports, or another Nature Portfolio journal, please use our manuscript transfer portal **[LINK REDACTED]. If you transfer to Nature journals or to the Communications journals, you will not have to re-supply manuscript metadata and files, unless you wish to make modifications. This link can only be used once and remains active until used.

All Nature Portfolio journals are editorially independent, and the decision on your manuscript will be taken by their editors. For more information, please see our manuscript transfer FAQ page.

Note that any decision to opt in to In Review at the original journal is not sent to the receiving journal on transfer. You can opt in to In Review at receiving journals that support this service by choosing to modify your manuscript on transfer. In Review is available for primary research manuscript types only.

Author Rebuttal to Initial comments

Responses to the reviewer's comments

We thank all three reviewers for highly constructive and useful comments to the previous version of our manuscript. The revised version is thoroughly revised accordingly. For instance, the genetic analyses are much better integrated in the manuscript. The largest change, however, is that we have included analyses of sleep duration vs brain change, using more than 8000 longitudinal MRIs. These new results now represent the major focus of the manuscript. The longitudinal data underscored the main message that shorter sleep than the current recommendations does not seem to have negative impact on brain health, as measured by brain morphometry. Further, we believe the combined longitudinal, cross-sectional and genetic results show that the inverted U-shaped relationships we and others find between sleep duration and brain structure represent stable relationships emerging earlier in life, not increased levels of brain atrophy in short (or long) sleepers.

We hope the reviewers agree that the new version of the manuscript is much improved. Specific comments to each of the points brought up by the reviewers are provided below.

Reviewer #1:

Remarks to the Author:

In the current work, Fjell et al., evaluated the phenotypic association and genetic correlation between sleep duration and brain structure. To do so, they analyzed 51295 MRI images from 47039 individuals, and then computed the association between self-reported sleep duration associated with the most relevant local metrics of volume, thickness and ventricle size. Following they performed genetic analysis ?? to evaluate which genes are different between people that sleep long vs short.

Though the topic of sleep and its link with the brain is overall interesting, I have various concerns on the current manuscript that I hope can be of help.

1. In the abstract, the notion 'most favorable brain outcome' is used. This is vague and suggestive language, what would be a perfect brain outcome and what would it mean?

Response: We agree that this term is unspecific and have removed it from the abstract. Towards the end of the Introduction we define our outcome properly:

“Here we tested the relationship between sleep duration and rates of brain atrophy. Higher rate of atrophy was regarded as a marker of declining brain health¹⁷⁻²⁰. (...) Additional analyses were conducted ... to estimate the amount of sleep associated with the overall thickest cortex and largest regional brain volumes.”

2. Overall the tone of the abstract is a bit odd, for example ‘hippocampus showing largest volume at 6.3 hours’ or ‘not correcting for ICV yielded longer durations associated with maximal volume’. It seems to imply some sort of causality, possibly it is helpful to rephrase, also considering potential non-experts reading this work.

Response: We have re-written the abstract, and hope the new version is clearer, also to non-experts.

3. I find the terms optimal brain health, as posed in the introduction unnecessary vague and lacking theoretical foundation. Is there any work claiming there is optimal brain health? What if there are various local optima, or not such a thing at all?

Response: We agree and have removed the term “optimal brain health” from the manuscript.

4. Regarding the review on sleep duration and brain structure, it is unclear whether this is a complete overview or just a selection? If so, what is the selection based on?

Response: It was intended to be a complete overview. However, with the new longitudinal results, we needed to cut back on some of the information in the introduction. We have therefore moved the table summarizing all the studies to SI, and rather describe the main conclusions that can be drawn from these studies in the manuscript.

5. The assumption is made that large volumes imply brain health, but is this association so clear cut? What about pruning/thinning of PFC?

Response: This is an important question, and we agree that in development, this relationship is not straightforward. There are also certain conditions which have been associated with larger brain volumes, especially in children/ adolescence. In healthy adults, however, we believe it is

solid evidence for a positive relationship between brain volumetric measures, brain health and cognitive function. But we agree with the reviewer that this should be better explained in the manuscript. In the revised version, the following is included in the Introduction:

“Brain health encompasses multiple features¹⁶. Important aspects can be indexed by rate of atrophy, which increases in normal aging¹⁷, cognitive decline¹⁸, AD¹⁹, and with e.g. cardiovascular risk factors²⁰. Lower rates of atrophy are related to healthy lifestyle²¹ and better maintained cognitive function²². Hence, if insufficient habitual sleep has detrimental effects on the brain, it is likely that short sleep will be associated with higher rates of atrophy.”

6. Why did the authors calculate sleep duration associated with the most favorable (=thick and large?) brain properties and not ‘just’ focus on the association between both in a more neutral way, given there is no clear evidence of what favorable brain configurations are?

Response: We apologize if the description of the analyses was not sufficiently clear. All analyses were done without any assumptions about the shape of the relationships. We used GAMM with a spline term for sleep, which will depict sleep-brain relationships of any form. Since most of them are characterized by an inverse-U shaped curve, we could use this to estimate the number of hours of sleep associated with the apex. However, this approach did not impose any restrictions on the possible form the relationships could take. The following is included in the manuscript which hopefully makes this clear:

“Associations were tested by use of Generalized Additive Mixed Models (GAMM) run in R58, a non-linear statistical approach which does not require a priori specification of a polynomial functional form⁵⁹. As the relationship between sleep duration and a range of health-related measures typically form an inverted U-shape, this approach allows us to accurately estimate the number of hours of sleep associated with the largest regional brain volumes and thickest cortex⁶⁰.”

7. The genetic argument comes a bit out of the blue.

Response: We agree that the genetic part was not sufficiently integrated. We have now re-written the genetic parts, including expanding the description of the rationale for the genetic analyses in the Introduction, expanded the genetic results, included a more integrated discussion of the genetic findings, re-designed the figures and also added new sections about the genetic analyses to the Methods section. We believe these changes have improved the integration of genetics with the rest of the manuscript. See also our responses to multiple comments about genetics by Reviewer 3.

8. Figure 1; it would be helpful to see the individual datapoints?

Response: We understand the reviewer's point but given the large number of observations (> 50.000), we have found no really good way of displaying the single data point. We have previously used density scatterplots with relatively large samples (≈ 20.000) but are not sure they really illustrate the data better than using estimated curves with standard errors when the number of datapoints is so great. We have added several scatterplots showing individual datapoints in the Supplemental Information, however.

9. Why did they authors opt for the desikan killiany atlas?

Response: The DK-atlas is arguably the most used in neuroimaging and is therefore be useful when comparing studies. In addition, we present vertex-wise analyses without use of an atlas in the Supplemental Information. In the revised version, we have added a short justification for this choice: "The Desikan-Killiany parcellation included in FreeSurfer yields 34 regions (...) This atlas was selected because it is well validated and commonly used for cortical region-of-interest based analyses, which is a benefit when comparing results across studies."

10. The cluster analysis is not clearly motivated, why were subcortical regions not incorporated as well?

Response: We performed a cluster analyses of the cortical results to reduce the dimensionality. As can be seen from the vertex-plots, the sleep-cortical effects are anatomically quite widespread, and in our view justifies some sort of data reduction. This is less of an issue for the volumetric structures since they are fewer and tend to form less uniform sleep-relationships. In any case, the results for all cortical as well as volumetric regions are presented in the manuscript. The following short justification is added to the manuscript (Results section): "The results were entered into a to a K-means cluster analysis to reduce the dimensionality of the cortical data."

11. Why were associations between vertex-wise brain and sleep links computed differently for long and short sleepers? Why was not also the interaction between age and sleep for example assessed but rather a young vs old model?

Response: The reason for this choice was that since we from previous research expected non-monotonous sleep-cortex relationships which possibly could interacting with age, the surface plots would be difficult to interpret without plotting the data. Hence, we decided that the easiest solution would be to use linear models separately for older vs younger and shorter vs longer sleep. However, for the remaining, region-based analyses, we modelled age as an interaction and used sleep as a continuous variable. Since we have added a substantial amount of new longitudinal analyses, the vertex-wise analyses are moved to the Supplemental Information. A justification for the described group analyses is provided in the supplement.

12. Minor point: in figure three below the statistical bar some letter seems to be only half removed

Response: Thanks for spotting this, it is now fixed (this figure is also moved to the SI).

13. Doesn't sleep vs maximal volume analysis imply similar trajectories for different regions across ageing, yet this isn't the case?

Response: For most structures, the sleep – brain volume effects did not vary with age. We explicitly tested this in the GAMM analyses, and the only region where sleep – volume interacted with age was the ventricles (the nature of the interaction is shown in the Supplemental Information). For cortical thickness, age seems to have a certain influence on the sleep relationships when inspecting the vertex-wise results. Still, formally testing age-interactions in the GAMM analyses did not suggest that this effect was significant. Nevertheless, we included age and the interaction between age and sleep in all models to account for this potential source of influence. Thus, the resulting meta-analytic estimate should give a good approximation for the sleep-brain relationships across the adult lifespan. In addition, the vertex-wise plots are presented separately for older and younger adults in the Supplemental Information. We believe the point raised by the reviewer is important, so we address this and related points about variability in the Discussion in a new section on “Variation across persons, age and regions”. The full section reads:

“The meta-analytic estimate of 6.8 hours is a best approximation, not a magic number. First, there were substantial regional heterogeneity in the cross-sectional results. For instance, hippocampus showed peak volume at 6.3 hours, and the white matter compartments even lower. Interpretation of these differences should be done keeping in mind the lack of sleep – atrophy relationships in the longitudinal analyses, and that these numbers therefore likely represent stable relationships rather than reflecting effects of sleep duration per se. People with the largest hippocampal volumes relative to ICV reported to sleep 6.3 hours, while people with the thickest cortex reported to sleep between 6.4 and 7.0 hours, but this cannot be interpreted to mean that less sleep is optimal for hippocampal volume than cortical thickness.

Second, an important qualification is that individual differences in sleep need exist, due to for instance genetic differences and previous sleep history^{50–56,93}. If deviations from an individual's sleep need led to poorer brain health for that individual, this may not be picked up in our group analyses. Thus, we cannot from our results conclude that people should try to sleep 6.8 hours each night. What the results show is that people who report to sleep 6.8 hours tend to have the thickest cortex and largest regional brain volumes relative to ICV. Intra-individual causal effects of changing sleep duration on the brain can be assessed in sleep deprivation studies. Unfortunately, experimental sleep deprivation does not resemble habitual variations in sleep duration, and the long-term consequences on the brain from of sleep deprivation, taking into account adaptations⁹³, are not known.

Finally, associations with sleep could potentially vary with age68, but the sleep-age interactions did not confirm that this was the case. The vertex-wise cortical thickness analyses (Supplemental Information) suggested that relationships with sleep duration were different in younger and older adults, but this was not confirmed in the GAMM analyses. Thus, we believe the meta-analytic results represent a good approximation of a general sleep – brain volume relationship on a group level, while ignoring that there naturally is variations across brain regions, age and participants.”

14. The genetic analysis at the end give a somewhat additive impression, and the link with the previous steps are not that clear. Ideally this could be clarified a bit better? For example, is there a different association with the 3 clusters from the beginning of result section?

Response: We agree with the reviewer that the genetic analyses and results were not sufficiently integrated with the rest of the manuscript. In the revised version, we have re-written the parts about genetic, and worked to show the relevance of the genetic results for the main research questions and conclusions. We believe this has improved the manuscript substantially.

15. ‘One interpretation is that there is a propensity towards optimal sleep duration for the brain’ this is mentioned in the discussion, but it seems the primary interpretation/driver of analysis of the authors

Response: It is true that we expected an inverted U-shaped sleep-brain relationship based on previous research. However, as mentioned in our response above, this did not constrain the analyses, which were not dependent on any assumptions about the shape of the relationships. In the revised manuscript, we hope this point is clear.

16. Overall, I believe the writing style is suggestive of causality whereas the data is correlational, possibly this may be something to edit?

Response: We do not want to claim that sleep duration has causal impact on the brain. We hope this is clear in the revised version. For instance, in the Abstract, we now conclude:

“The combined results challenge the notion that habitual short sleep causes brain atrophy, (...)”

In the discussion, the following is included:

“Thus, we cannot from our results conclude that people should try to sleep 6.8 hours each night. What the results show is that people who report to sleep 6.8 hours tend to have the thickest cortex and largest regional brain volumes relative to ICV. Intra-individual causal effects of

changing sleep duration on the brain can be assessed in sleep deprivation studies. Unfortunately, experimental sleep deprivation does not resemble habitual variations in sleep duration, and the long-term consequences on the brain from of sleep deprivation, taking into account adaptations⁹³, are not known.”

Despite this, we believe our genetic analyses and the new longitudinal results have implications for possible sleep-brain causality, which is discussed in some depth in the revised manuscript.

17. In the end a comparison to recommendations of sleep length is made, but I believe the current study set-up is not fit for such a statement due to the study set-up?

Response: We understand the reviewer’s point, and we are not trying to suggest new guidelines for sleep duration. However, we believe it is relevant to discuss that the current cross-sectional results are not in line with the recommendations, and that this may have implications for whether the guidelines should be applied to brain health. Even though we cannot isolate causality, we think it is interesting that the sleep duration associated with the largest regional brain volumes tend to be shorter than the recommendations. Further, considering the new longitudinal analyses, we believe the results have implications relevant for the questions of whether short sleep has negative (causal) effects on brain structure. Still, we agree that the current results have limitations, which we explicitly acknowledge in the revised manuscript. See also response to the comment above.

Reviewer #2:

Remarks to the Author:

Key Results

Over close to 50,000 people, sleep durations around 6.8 hours were linked with the largest overall brain volumes. There were regional variations, with the hippocampus showing highest volume related to 6.3 hours of sleep, and cortical regions showing highest volumes with over 7 hours of sleep.

Originality and significance

This is a very large study in the field of brain measurements, giving a strong confidence in the reliability of the results. Specifically, the fact that sleep of less than 7 hours was associated with the highest brain volumes can be trusted. This finding raises the question of whether 7 hours as a recommended minimum is appropriate. The large study also gives reassurance that the substantial variations across brain regions of hours of sleep related to maximum volume are indeed present, and allows us to consider what might be the source of such variations. The findings challenge us to consider the relationship between “brain health” as a physiology and brain volume, on both a whole-brain and regional level.

Response: We thank the reviewer for the positive comments.

I am less sure of the meaning of the genetic analyses; these seem completely unconnected in the analysis from brain volume.

Response: We agree with the reviewer that the genetic analyses and results were not sufficiently integrated with the rest of the manuscript. We have now re-written the genetic parts, including expanding the description of the rationale for the genetic analyses in the Introduction, expanded the genetic results, included a more integrated discussion of the genetic findings, re-designed the figures, and added new sections about the genetic analyses to the Methods section. We believe these changes have improved the integration of genetics with the rest of the manuscript. See also our responses to multiple comments about genetics by Reviewer 3.

I have little confidence in the significance of the findings from categorizing people who sleep less or more than 7 hours as shorter or longer sleepers; it seems the authors' own results suggest there is a range of sleep times that are healthy, so no variation related to poor health would be expected in that range – this idea is supported by much literature in the sleep field. Indeed, taking a slightly controversial view, one might argue that the present findings are a more reliable reflection of healthy sleep range...at least for the middle quartile, since one would assume that 25% of people would not have pathological sleep durations. In any case, most of the evidence only shows clear pathology (or poor health) at much shorter sleep times than 7 hours.

Response: We agree with the reviewer that below/ above 7 hours is not sufficient to categorize people as long/ short sleepers. This number is based on the average reported sleep in this study, as well as in recent large meta-analyses (cited in the manuscript). Interestingly, converging lines of evidence from multiple fields seem to point towards sleep duration of about 7 hours as being associated with the best health outcomes. Importantly, however, we are not primarily interested in contrasting these two groups specifically, rather we perform the genetic analyses of continuous sleep duration within each of the groups, which yielded quite interesting results. Except for the genetic analyses (and the vertex-wise plots which are now moved to SI), all analyses are GAMMs based on continuous sleep duration, where it would be evident if effects exist only at the more extreme ends of the sleep duration range: if, e.g. sleep < 5 hours is associated with particularly deviant results, this would be detected in these analyses. However, we agree with the reviewer that below 7 hours should not be termed "short sleepers". Therefore, we have replaced this term with "shorter than average sleepers" and "longer than average sleepers" throughout the manuscript.

Conceptual & methodological novelty

Looking at a simple question in a way that confidently provides an answer is arguably a conceptual

novelty. Certainly in the field of neuroimaging there are few large N studies, which also limits the types of appropriate analyses that can be performed, for example such as allowing for non-linear relationships between sleep and brain volume.

I do think conceptually the manuscript could do with a little explanation of how brain health can be reflected in brain volume, meaning explaining possible biological states and processes that lead to higher or lower brain volume.

Response: We agree that this could be useful, and have added the following descriptions to the manuscript:

Introduction: "Brain health encompasses multiple features¹⁶. Important aspects can be indexed by rate of atrophy, which increases in normal aging¹⁷, cognitive decline¹⁸, AD¹⁹, and with e.g. cardiovascular risk factors²⁰. Lower rates of atrophy are related to healthy lifestyle²¹ and better maintained cognitive function²². Hence, if insufficient habitual sleep has detrimental effects on the brain, it is likely that short sleep will be associated with higher rates of atrophy."

Introduction: "Importantly, such results cannot be used to make inferences about atrophy and brain change, as inter-individual brain volumetric differences even in adults mainly reflect early developmental processes^{17,36}. Accordingly, larger brain volumes are positively and stably related to life-long higher cognitive function and demographic variables such as education^{17,37,38}. Therefore, cross-sectional sleep – volume relationships^{26,39–49} represent mostly stable factors, not brain changes²⁶"

We have also added a section on possible biological states that may be related to the observed volumetric changes and differences.

Discussion: "Underlying neurobiology cannot be directly inferred from MRIs, but number of neurons have been shown to correlate with regional⁷⁴ and global⁷⁵ brain volumes cross sectionally. Longitudinally, volumetric reductions and cortical thinning occurring during adulthood may be associated with shrinkage of neurons, dendrites and axonal arborizations^{76,77}, reduced spine numbers and density⁷⁸, loss of synapses and dendritic branches⁷⁹, and in degenerative conditions such as AD, also neuronal loss⁷⁶, although neuronal^{77,80,81} or glial⁸² loss likely plays limited roles in the volumetric reductions seen in adulthood. Sleep duration – brain correlations were seen in the cross-sectional analyses only, so it is unlikely that these are caused by neurobiological events underlying morphometric changes observable with MRI during adulthood. Thus, events ongoing during earlier life stages, in development, may be more relevant. Processes such as synaptogenesis and synapse elimination/pruning⁸³, dendritic and axonal growth^{84,85}, and intracortical myelination⁸⁶ can be involved in morphometric changes in childhood development. Some of these, however, such as synaptic density, will have minute effects on volumetric measures because their total volume is very small^{85,87}. In any case, it must be stressed that the volumetric analyses in the present study are corrected for ICV, which means that relative and not absolute volumes are used in the calculations. Thus, it is not clear which or any of the processes above can contribute to explain the observed cross-sectional relationship. Research into the neurobiology of sleep has focused more on electrophysiological processes and neurotransmitter systems than on macrostructural differences. In addition, the association

between neurodegeneration and sleep problems may be due to any disturbance of normal brain function and structure likely affecting how we sleep, and the neurobiological foundation will then vary depending on the underlying condition. Hence, a neurobiological interpretation of the present findings will be speculative and must be based on general knowledge about the relationship between brain features and different human traits. For example, we have previously shown that sleep disturbances are associated with spatial expression patterns of oligodendrocytes and S1 Pyramidal Cell Genes⁶⁸, in line with theories of relationships between myelination and sleep⁸⁸. To our knowledge, such analyses have not been reported for sleep duration specifically."

Relatedly, it would be nice to read a brief explanation of how short sleep is associated with poor health in conditions that are known to affect the brain (e.g., insomnia, depression) and similarly for long sleep (e.g., atypical depression subtype, sleep apnea). There is a brief mention in the discussion, but it would help to explain in the Introduction why brain health can be reflected in brain volume. Inflammation, atrophy, learning/neuroplasticity are relevant concepts, to name a few.

Response: We agree that this is interesting and relevant. We have added a new section to the discussion about possible neurobiological substrates of morphometric differences and changes (see response to point made above). In the discussion, we have clarified and extended the sections about the relationship between brain health and brain morphometry. In combination, we hope these changes address the reviewer's point.

I am not familiar with genetic analyses; I would have liked to see them related to brain volume, but as it stands I did not grasp the conceptual link between the genetics and brain volume or even brain health.

Response: Yes, this was not clear in the previous version, but we believe the current manuscript integrates the genetic analyses and results in a much better way (see also responses above).

Data and methodology

Unique, outstanding data reflecting a chance to truly answer the research question with confidence. The basic methods as described appear to be appropriate, well documented, and replicable. I would like to have seen more reference in the manuscript text to quality control, since this is a key component of analyzing large datasets. Additionally, the manuscript text did not give much detail on the way the segmentations were performed.

Response: We agree that more detail should be added. In the revised version, we have added a separate section in the Methods on "Magnetic resonance imaging". This section reads:

“Lifebrain MRI data originated from seven different scanners (see 25 for details), processed with FreeSurfer 6.0 (<https://surfer.nmr.mgh.harvard.edu/>) 100–103. Because FreeSurfer is almost fully automated, to avoid introducing possible site-specific biases, gross quality control measures were imposed and no manual editing was done. To assess the influence of scanner on volumetric estimates, seven participants were scanned on seven scanners across the consortium sites (see 25 for details). Using hippocampus as test-region, there was a significant main effect of scanner on volume ($F = 4.13$, $p = .046$), but the between-participant rank order was close perfectly retained between scanners, with a mean between-scanner Pearson correlation of $r = .98$ (range .94–1.00). Thus, including site as a random effect covariate in the analyses of hippocampal volume is likely sufficient to remove the influence of scanner differences.

UKB participants were scanned using three identical Siemens 3T Prisma scanners (UK Biobank Brain Imaging – Acquisition Protocol (ox.ac.uk)). FreeSurfer outputs 104 and the volumetric scaling from T1 head image to standard space as proxy for ICV were used in the analyses, generated using publicly available tools, primarily based on FSL (FMRIB Software library, <https://fsl.fmrib.ox.ac.uk/fsl/fslwiki>). Details of the imaging protocol (<http://biobank.ctsu.ox.ac.uk/crystal/refer.cgi?id=2367>) and structural image processing are provided on the UK biobank website (<http://biobank.ctsu.ox.ac.uk/crystal/refer.cgi?id=1977>).

Manuscript – General

Assumption that thicker cortex reflects better brain health is true on average, but there are unhealthy states (inflammation) that result in larger brain tissue volume. For example, is it possible that the caudate estimate of 11.8 hours for maximum volume reflects inflammation?

Response: We agree with the reviewer that some conditions may be associated with thicker cortex, especially in development. However, these are relatively rare, and unlikely to affect the results in the present study, comprised of healthy participants. We have previously studied various markers of (neuro)inflammation in relation to atrophy, and find that sTREM2 (Halaas et al., Cereb Cortex 2020), YKL-40 (Sala-Llonch et al., J Alzheimers Disease 2017; Fjell et al Cereb Cortex 2018)) and CRP (Wang et al. Brain Behavior and Immunity, 2022) are related to smaller volumes/higher rates of atrophy. Thus, we believe it is unlikely that inflammation is a factor causing larger volumes in the present study. The caudate estimate may be due to the sleep-volume relationship for this structure being relatively flat, and hence the peak estimation is associated with more uncertainty. As mentioned above, we have added more information about possible underlying neurobiology, and try to be clearer about why we believe thicker cortex, larger volume and less atrophy can be considered as indices of good brain health, e.g. in these sections of the Introduction:

“Brain health encompasses multiple features¹⁶. Important aspects can be indexed by rate of atrophy, which increases in normal aging¹⁷, cognitive decline¹⁸, AD¹⁹, and with e.g. cardiovascular risk factors²⁰. Lower rates of atrophy are related to healthy lifestyle²¹ and better maintained cognitive function²². Hence, if insufficient habitual sleep has detrimental effects on the brain, it is likely that short sleep will be associated with higher rates of atrophy”

“(...) larger brain volumes are positively and stably related to life-long higher cognitive function and demographic variables such as education^{17,37,38}.”

The premise that there is an optimal sleep time for brain health is challenged by the fact that different brain areas show different hours of sleep for maximum volume. Are the authors saying 6.3 hours leads to optimum hippocampal health but 6.8 hours leads to optimal cortical health? If so what sleep mechanism could differentially impact those two brain regions?

Response: The reviewer is right that the peak of the inverted U-shaped sleep – brain volume relationships vary across brain structures. Whether this has any mechanistic implications is unknown. To directly address the reviewer’s point, we have added a new section to the Discussion, where we discuss variation across regions, age and individual differences. This section reads as follows:

“Variation across persons, age and regions

The meta-analytic estimate of 6.8 hours is a best approximation, not a magic number. First, there were substantial regional heterogeneity in the cross-sectional results. For instance, hippocampus showed peak volume at 6.3 hours, and the white matter compartments even lower. Interpretation of these differences should be done keeping in mind the lack of sleep – atrophy relationships in the longitudinal analyses, and that these numbers therefore likely represent stable relationships rather than reflecting effects of sleep duration per se. People with the largest hippocampal volumes relative to ICV reported to sleep 6.3 hours, while people with the thickest cortex reported to sleep between 6.4 and 7.0 hours, but this cannot be interpreted to mean that less sleep is optimal for hippocampal volume than cortical thickness.

Second, an important qualification is that individual differences in sleep need exist, due to for instance genetic differences and previous sleep history^{50–56,93}. If deviations from an individual’s sleep need led to poorer brain health for that individual, this may not be picked up in our group analyses. Thus, we cannot from our results conclude that people should try to sleep 6.8 hours each night. What the results show is that people who report to sleep 6.8 hours tend to have the thickest cortex and largest regional brain volumes relative to ICV. Intra-individual causal effects of changing sleep duration on the brain can be assessed in sleep deprivation studies. Unfortunately, experimental sleep deprivation does not resemble habitual variations in sleep duration, and the long-term consequences on the brain from of sleep deprivation, taking into account adaptations⁹³, are not known.

Finally, associations with sleep could potentially vary with age⁶⁸, but the sleep-age interactions did not confirm that this was the case. The vertex-wise cortical thickness analyses (Supplemental Information) suggested that relationships with sleep duration were different in younger and older adults, but this was not confirmed in the GAMM analyses. Thus, we believe the meta-analytic results represent a good approximation of a general sleep – brain volume relationship on a group level, while ignoring that there naturally is variations across brain regions, age and participants.”

I do not see how the genetic analysis fits in to the premise of the article. It seems to be an afterthought, or an independent study.

I think it would improve readability to replace “below average sleep duration group” with “short sleepers”, and “above average sleep duration group” with “long sleepers”

“It is a worry” – one example that demonstrates the need for an English language review for appropriate style. Most of the writing is good.

Response: Fully agree. We have re-written the genetics in the introduction, results, discussion, and methods, integrating the genetic analyses much better. We hope the genetic results now fit better into the overall presentation of the study. See also responses to comments above, as well as answers to reviewers 1 and 3 about this. As suggested by the reviewer, we have replaced the group names with “below average sleepers” and “above average sleepers”, respectively. Finally, we have removed the sentence starting with “It is a worry”.

Abstract

Needs clarifying, since this is the most read part of any paper.

Need link between sufficient sleep and brain – just being explicit that “breath health as reflected by larger brain volume”

Change “favorable” and definetics not introduced, and genetic results presented without context in relation to the brain. If there was no independent analysis of genetic vs brain associations, does this belong here?

The short/long sleep sentence is confusing. First off, it is unclear what a short or long sleeper is – presumably outside the 95% CI, but that is not clear. Secondly, I think the point is that one gene both lengthened sleep in short sleepers, and shortened sleep in long sleepers – but how was that determined in a cross-sectional study? How can a person be defined being 1) a short sleeper and 2) having their sleep longer than in others with short sleep? It’s just not clear. In the Results, there is “below average”, but in the sleep field “short sleepers” are people with short sleep (for example threshold of <5, < 6 and < 7 hours have been used by various groups). Further in the Results, there is a section defining shorter and longer sleepers as below of above 7 hours, but we would really expect differences only in people outside a range, say perhaps the 95% CI.

Mendelian randomization unclear to most readers

Unclear how “habitual” short or long sleep was measured, so conclusion does not follow (related to above point about genetics).

Response: We have re-written the abstract, and believe the present version is much clearer than the previous. We believe the points raised by the reviewer are now resolved.

Introduction

Seems to be suggesting AD is distinct from brain health, whereas AD could be considered an

example of poor brain health
We can assume that sleep is essential

Response: The first part of the Introduction is re-written, and we believe it is clearer and more accurate. The first paragraph now reads: "Adults are advised to sleep at least 7-8 hours each night¹⁻⁴, and it is widely perceived that shorter sleep could be a pervasive negative factor for physical, mental and cognitive health⁵⁻⁸, yielding increased risk of Alzheimer's Disease (AD) and other dementias⁹⁻¹⁵. However, we still do not know what amount of sleep is associated with good brain health, and whether a causal relationship between variations in habitual sleep duration and brain health exists. Here we address these questions, analyzing MRIs of the brain and genetic data in a combined longitudinal and cross-sectional design."

Results

This may be a journal style issue, but there are several sections where the methodology is described; should this be moved to the Methods section, which is very short at the moment?

Response: According to journal style, Methods are to follow the Discussion. Hence, we believe for readability it is necessary to explain some methodological aspects also in the preceding sections. However, in the revised version of the manuscript, the methods section is substantially expanded. We hope the revised version is easy to read and comprehend.

Table 2 and related text – the abstract mentions ventricle to ICV ratio; is this what is being referred to here in the Results? Or are these results based on absolute volumes?

Response: In the main volumetric analyses, ICV is used as covariate. However, we also report results without controlling for ICV. In the revised version, we make clear in each case whether ICV is controlled for or not. In the abstract, the relevant text reads: "... 6.8 (CI: 5.7, 7.9) hours were associated with thickest cortex and largest volumes relative to intracranial volume". In the results, we make clear that the volumetric analyses are done by covarying for ICV: "ICV was included as covariate of no interest in the volumetric analyses." This is also repeated in the discussion.

Figure 3 – the caption could better explain the relationships, since they are a bit confusing as a negative correlation in the below-average sleep duration means something different to the negative correlation in the above-average sleep duration analyses (at least assuming the bigger volume is better). It would also help to have the N and the definition of below-average sleep duration

Response: We regret that the caption was unclear. The direction tested is the same across, so that blue/ cyan means longer sleep goes with thinner cortex, and red/ yellow that shorter sleep goes with thinner cortex. This figure is moved to the Supplemental Information to make room for the new longitudinal analyses, but we have still revised the caption: “Across all panels, warm colors indicate that longer sleep is associated with thicker cortex and cold colors indicate that shorter sleep is associated with thicker cortex.”

Figure 7 – font sizes and labels need tidying; for example, legend does not explain “SleepLe7”

Response: We agree that this figure was did not have an optimal layout. We have redesigned the figure, and the genetic results are shown in Figure 5 and 6 in the revised manuscript. We believe these are substantially improved:

Figure 5. Genetic relations between sleep duration and brain structure. a. Distribution of PGS-ICV in different sleep duration strata among shorter than average sleepers (≤ 7 hours). b. ICV for shorter than average sleepers with one standard deviation above (blue) and below (red) average PGS for sleep duration c. Distribution of PGS for total gray matter volume in different sleep duration strata among shorter than average sleepers. d. Total gray matter volume for shorter than average sleepers with one standard deviation above (blue) and below (red) average PGS for sleep duration.

Figure 6 SNP effects on MRI-derived measures. a. SNP effects on intracranial volume (ICV) (x-axis) and sleep duration (y-axis) for the shorter than average sleepers. b. SNP effects on total gray matter volume (TGV) (x-axis) and sleep duration (y-axis) for the shorter than average sleepers.

Methods

I'm not sure what this journal convention is, but it seems like there should be something about Mendelian randomization analyses in the manuscript; however, there is much methodology in the Results at present.

Response: We agree that the Mendelian randomization analyses were too briefly described. We have now added more details about the genetic analyses to the main manuscript. The most relevant sections now read:

Introduction: "We took advantage of measured variation in genes for each trait of interest and used Mendelian randomization⁵⁹ to test the direction of causality between sleep duration and brain structure."

Results, in the section "GWAS, polygenic scores, genetic correlations and Mendelian randomization": "To determine the plausible direction of causality between brain structure and sleep duration, we performed a series of genetic analyses using cross-sectional data from UK Biobank. Hippocampus, total gray matter volume and ICV were chosen as regions of interest as they showed the typical inverted U-shaped relationship to sleep duration. For details about selection of participant, quality control procedures and genetic analyses, see SI Genetic analyses, SI Genetics notes and SI Genetics tables."

and

"We performed bidirectional Mendelian randomization analysis for each brain volumetric trait to sleep duration. Among the 12 pairs, ICV showed a significant causal effect (inverse-variance weighted beta = 0.060, se = 0.017, $p = 5.36 \times 10^{-4}$) on sleep duration for the shorter than average

sleepers (Figure 6a), and total gray matter volume showed an effect for the longer than average sleepers that did not survive correction for multiple comparisons (beta = -0.35, se = 0.14, p = 0.012, uncorrected, Figure 6b). Due to low heritability for sleep in the study, resulting in a weaker genetic instrument, we performed a robust MR analysis using robust adjusted profile score⁷¹ for the direction from sleep to brain traits, but did not detect significant causal relationships for the directionality of effects by this even more liberal threshold. The only significant causal relation was robust when we performed the analysis using both a stringent and a weaker instrument selection protocol. This suggests that there are no strong causal effects from sleep duration to brain morphometry.”

Methods: “Genetic analyses: Two-sample Mendelian randomization

The TwoSampleMR R package¹¹⁸ was used to investigate the existence of causal relations between sleep duration and the brain variables. Independent instrumental SNPs were selected using the following parameters: association p value $\leq 10^{-6}$, MAF ≥ 0.05 , LD- $r^2 \leq 0.1$, and LD-distance=10kb. The LD structure was derived from 10,000 independent European participants randomly selected from UKBB. The powerful inverse variance weighted (IVW) model from TwoSampleMR was used as the main model. Other models implemented in the software were also run as sensitivity analysis. To further support the results, the analysis was reperformed with p value $\leq 10^{-5}$, which would increase the strength of instrumenting for the less powerful sleep duration traits. For the only significant causal relation, i.e., ICV to sleep duration for shorter than average sleepers, a third analysis with p $\leq 5 \times 10^{-8}$ used for selecting instrumental SNP was performed. The standard output from TwoSampleMR is shown in Supplementary Note2”

Discussion

Several comments above relate to the Discussion.

For example, in Conclusions, the term “Short sleep” has many interpretations and needs to be clarified

Response: We agree, and believe this is resolved in the revision, see also responses to the comments above. In the Conclusion, the relevant part now reads: “Sleep duration was not related to rate of atrophy, and shorter than recommended^{1,63} sleep was not associated with smaller regional brain volumes, thinner cortex or smaller ventricles. Rather, sleeping less than the recommended amount was associated with thicker cortex and greater regional brain volumes relative to intracranial volume, and moderately long sleep showed a stronger association with smaller volumes than even very short sleep (e.g. < 5 hours).”

Reviewer #3:

Remarks to the Author:

Thank you for the opportunity to review this paper by Fjell and colleagues. It has some interesting analyses and the authors have clearly done a lot of work. I have mainly focused on the genetic

aspects of the work and have some concerns that I detail below.

Response: Thank you for the positive evaluation of our manuscript.

Minor comments:

1. Introduction: I'd take out the 'It is a worry' from the first sentence, as it's a bit of an odd opener for a scientific paper.

Response: Agree, this sentence is now removed.

2. Are the authors defining brain health as only morphology (i.e., measures from MRI)? Or do they also consider for example, cognitive function and decline? If so, there is one large-scale Mendelian randomisation study with both linear and non-linear analyses showing a causal relationship between sleep duration and worse cognitive function (Henry et al, 2019, IJE).

Response: In this paper, we analyze data on brain structure only. We have now made this clearer in the introduction: "Here we tested the relationship between sleep duration and rates of brain atrophy. Higher rate of atrophy was regarded as a marker of declining brain health¹⁷⁻²⁰."

3. I think the authors should not talk about their study until the end of the Introduction, as this is not conventional.

Response: There is probably different opinions on this issue, but we would prefer to tell the reader in the first paragraph what the main purpose of the study is. The current version of the manuscript is quite comprehensive, including longitudinal, cross-sectional and genetic analyses. Hence, to help the reader grasp the focus of the study, we have included a sentence detailing the overall strategy at the end of the first paragraph: "However, we still do not know what amount of sleep is associated with good brain health, and whether a causal relationship between variations in habitual sleep duration and brain health exists. Here we address these questions, analyzing MRIs of the brain and genetic data in a combined longitudinal and cross-sectional design."

4. It is fine to cite studies for heritability and polygenic influences on sleep duration, but I think the authors need to clarify that the SNP heritability of sleep duration in GWA studies is very modest.

Response: We agree. The following sentences are now included in the Introduction:

“Twin and genome wide association studies (GWAS) have demonstrated heritability and polygenic influences on sleep duration, although GWAS heritability is modest^{48–54} (...).”

5. Why the specific choice of covariates in the post-hoc analyses? Why BMI, SES and depression?

Response: These were chosen as covariates because they may influence sleep duration, brain structure and possibly the relationship between them. A short justification is added to the manuscript (Results section): “Post hoc analyses were run controlling for socioeconomic status (SES: income and education), BMI and depression symptoms in turn as covariates, as these variables may affect sleep duration, brain structure and possibly the relationship between them.”

6. Please avoid phrases like ‘nominally significant effect’ and all round I’d avoid ‘significant’ when talking about associations, even if causal.

Response. We have replaced the “nominal significant effect”. However, although we are aware that the use of significance testing has its weaknesses, it is still common to p-values to guide interpretations. Both in brain imaging and genetics are significance thresholds the basis for interpreting results. Thus, we still report the significance levels of the different statistical tests we perform (adjusted for multiple comparisons).

7. Relatedly, can the authors present coefficients/effects with 95% CIs, instead of the beta/SE/p-value? Then we can focus on the size of the coefficient, direction and precision by seeing the 95%CIs and not focus on p-values.

Response: Most of the brain analyses are based on GAMMs with spline functions. A coefficient cannot be derived from a spline function, but we can calculate a significance level for the smooth term. Still, we report multiple statistical parameters for each of the tests we perform, for the genetic analyses including betas (coefficient) with sign (direction) and standard errors (to evaluate precision). As most of the sleep-brain relationships are non-monotonous and or involving interactions, the betas and their signs are not always easy interpretable for the average reader. Hence, we believe that we present quite comprehensible results, and that using p-values to guide interpretations is warranted and a reasonable choice for communication with most readers.

8. I know it might seem obvious to some readers (including me), but I think that perhaps a solid justification for the MR analyses might be worth including – i.e., they wanted to try and understand causality and directionality in this case.

Response: We agree that the rationale for the genetic analyses, including MR, was not sufficiently clear. We have therefore revised and expanded all the sections about genetics in the manuscript. For instance, the following is included:

Introduction: "We took advantage of measured variation in genes for each trait of interest and used Mendelian randomization⁵⁹ to test the direction of causality between sleep duration and brain structure."

Results: To determine the plausible direction of causality between brain structure and sleep duration, we performed a series of genetic analyses using cross-sectional data from UK Biobank. (...) GWAS results for these brain features were used for the polygenic score and Mendelian randomization analysis below. (...) We performed bidirectional Mendelian randomization analysis for each brain volumetric trait to sleep duration. (...) "

Discussion: "Aligning with the longitudinal results, the Mendelian randomization analyses did not reveal evidence for a causal impact of short sleep on brain structure."

1. Major comments:

2. I have not often seen individuals grouped into 'short' and 'long' sleepers and classified as ≤ 7 h and > 7 h. Can the authors provide some justification for these categories? If the authors are following the US NSF guidelines, then they suggest that adults sleep between 7-9h and older adults sleep between 7-8h. This implies that a sleep duration of > 7 h is not long sleep?

Response: We agree with the reviewer that below/ above 7 hours is not sufficient to categorize people as long/ short sleepers. This number is based on the average reported sleep in this study, as well as in recent large meta-analyses (cited in the manuscript). In the revised manuscript we have replaced "short" and "long" sleep with "shorter than average sleepers" and "longer than average sleepers", respectively. Importantly, however, we do not run direct contrasts only between these two groups, rather we do continuous analyses of sleep duration within each of the groups. The distinction between above vs below average sleepers is only done for selected analyses, especially the genetic analyses. For the sleep-brain analyses, we use GAMMs and model the relationship continuously across the range of reported sleep durations.

3. Can the authors comment on/acknowledge potential issues with analysing a sample with such a wide age range? 20-89y is a very wide age range, especially for these sorts of phenotypes, such as sleep duration which changes throughout the life course. Heritability, for example, changes with age for certain complex traits (e.g., BMI) so would we not expect this perhaps for sleep duration and some of the brain morphology traits?

Response: The reviewer is right that some of the traits analyzed in the current paper change throughout life, especially the brain morphometric measures, although not so much sleep duration in the current sample (age explained a trivial amount of the variance in sleep duration).

We cannot exclude the possibility that heritability varies with age, sex, socioeconomic status etc. In the current analyses, we use these variables as covariates. Further, we have previously seen that sleep-brain relationships do not strongly interact with age. If we had restricted the sample to a narrower age range, this would negatively impact statistical power and generalizability. Thus, we believe our current approach is optimal, controlling for relevant covariates statistically, while maximizing sample size and statistical power. To make this clear, the following is included in the manuscript:

Methods, Genetic analyses: GWAS: "For each GWAS, sex, baseline age and the top 10 genetic principal components were included as covariates in a linear regression model for identifying associate SNPs for each trait."

4. GWAS analyses: I am guessing, although this is not wholly clear from the description of the methods, that the authors decided to run their own GWAS for each trait because if they had only used summary statistics from published GWA studies they would not be able to discern which individuals had/had not undergone MRI scans? I'm not sure what else would be the justification for performing these GWASs, especially for sleep duration, given that large-scale UKB published GWASs exist (i.e., Dashti 2019, etc).

Response: We apologize for the unclear description of the analyses. Indeed, we performed a series of GWAS using UK Biobank dataset. There were three reasons for this:

First, as the reviewer points out, we needed to run our own GWAS to ensure that we are using completely non-overlapping samples for the sleep and the sleep-MRI analyses.

Second, we were interested in contrasting participants with below vs. above average sleep duration. As we see converging evidence across different fields for sleep duration about 7 hours being associated with lowest mortality and best health, this contrast is especially interesting. Importantly, we study the variation within the below average sleeping group and within the above average sleeping group. This is different from Dashti et al., who contrasted short-sleepers (≤ 6 hours) and long-sleepers (≥ 9 hours) with normal sleeper (7-9 hours). This is interesting but deviates from the current approach. Combined with our MRI analyses, we believe these set of new GWASs can provide novel insight into the genetics of sleep duration and its relationship with brain health. For example, we found a significant negative genetic correlation between sleep duration in below average sleepers and sleep duration in above average sleepers ($r = -0.4$, $p = 9.65 \times 10^{-5}$), suggesting that duration-increasing polygenes for short-sleep may have duration-decreasing effect in long-sleeper. Considering the multiple lines of evidence showing inverse U-shaped relationships between sleep duration in various health outcomes, we think this is a quite interesting result.

Third, while our GWAS results themselves are interesting and novel, an important aim is to assess whether there are plausible causal relations between sleep duration and brain health, using polygenic score and Mendelian randomization methods. The widely used models for these two

methods assume of monotonic relationships, where effects do not change direction across the range of phenotypic values. This does not fit the inverse U-shaped relationship between sleep duration and brain features. Thus, using the sleep duration published by Dashti et al 2019 is not a sensible option in our case, particularly for the Mendelian randomization analysis.

To make our strategy clear, we have included a better description of the methods and rationale for our genetic analyses. The following is added to the manuscript:

Results section: "Samples were stratified into shorter (< 7 hours) and longer (>7 hours) than average sleepers, since different relationships were expected in these two groups. Two independent samples were used for GWAS: (1) participants sleeping < 7 hours without MRI (n=197,137), and (2) participants sleeping > 7 hours without MRI (n=112,839). GWAS were performed independently for each trait in each the corresponding sample. We further performed GWAS for hippocampal volume, total gray matter volume and ICV using the 29,155 UK Biobank participants who were not included for the sleep duration GWAS. GWAS results for these brain features were used for the polygenic score and Mendelian randomization analysis bellow. Further details, Manhattan plots and QQ plots showing GWAS results are presented in SI Genetic analyses. We did not observe noticeable inflation in association statistics ($\lambda = 1.03, 1.02$ for shorter and longer than average sleepers, respectively)."

Methods section: "GWAS were run instead of using summary statistics from previous genetic studies of sleep in UKB (e.g. 50) for three reasons: (1) we needed to ensure that we were using completely non-overlapping samples for the sleep and the sleep-MRI analyses, (2) we were interested in contrasting participants with below vs. above average sleep duration and study the variation within each group, which has not previously been done, and (3) an important aim is to assess whether there are plausible causal relations between sleep duration and brain structure, using polygenic score and Mendelian randomization methods. The widely used models for these methods assume of monotonic relationships, where effects do not change direction across the range of phenotypic values. This does not fit the inverse U-shaped relationship between sleep duration and brain features. Thus, we could not use previously published summary statistics, particularly for the Mendelian randomization analysis."

5. Polygenic risk score analyses: what is PRS-CS? The reference doesn't seem to match this. Can the authors define and explain what this is? Am I right that the authors derived their own internal weights and used these to create their PRSs? If so, I'm not sure this is the best approach, as it is usually recommended that the base/training sample is different from the target/analytical sample to avoid overfitting. It is also not clear what p-value threshold was used for the PRSs, or what LD thresholds were used.

Response: We regret that our descriptions were not sufficiently clear. The Polygenic risk score via continuous shrinkage priors (PRS-CS) model is a widely used method for computing polygenic score for highly polygenic traits¹. PRS-CS shrinks effect sizes estimated from GWAS using linkage-disequilibrium (LD) correlations in a Bayesian framework, assuming a two-component

mixture prior distribution. The LD correlations provided by PRS-CS were based on the 1000 Genomes phase 3 European population. In total, about 1.3 million high quality SNPs were used. In addition, PRS-CS does not need information from the target sample where the estimated posterior effect will be used for computing polygenic score. Thus, the GWAS sample and the target sample were treated fully independently in the polygenic score computation. Furthermore, considering ours and previously published GWAS results, we assumed a highly polygenic genetic architecture for both MRI derived traits and sleep duration, by setting the parameter ϕ to 0.01, instead of a grid-search strategy proposed by the model. We believe our choice, though conservative, further reduce the risk of overfitting. For other parameters in PRS-CS, we used the default values.

The posterior effect sizes obtained by running PRS-CS on each GWAS were then used separately to compute polygenic scores. We did not use p -values or LD threshold to select SNPs. Rather, genome wide SNPs were used for computing the polygenic scores. After removing rare variants (minor allele frequency < 0.01) in UK biobank, variants not in the HapMap 3 data, and variants that are not on the Hardy-Weinberg equilibrium ($p < 10^{-6}$), we used the remaining 615297 SNPs for computing the polygenic scores for each trait.

Recent methodology studies all point to the advantage of using shrinkage based methods over p -value based thresholding methods, for example LDpred², PRS-CS, and the lasso-based models³, particularly for highly polygenic traits.

In conclusion, we believe the choice of models are reasonable in the present study.

To make the methodology clear, and to justify our choices, the following is added to the manuscript (Methods, Genetic Analyses: Polygenic scores):

“To accurately estimate the polygenic scores for a trait, we first computed the posterior effect size per SNP using the Bayesian mixture model implemented in PRS-CS⁴. The Polygenic risk score via continuous shrinkage priors (PRS-CS) model is a widely used method for computing polygenic score for highly polygenic traits¹¹⁵. PRS-CS shrinks effect sizes estimated from GWAS using linkage-disequilibrium (LD) correlations in a Bayesian framework, assuming a two-component mixture prior distribution. The LD correlations provided by PRS-CS were based on the 1000 Genomes phase 3 European population. In total, about 1.3 million high quality SNPs were used. In addition, PRS-CS does not need information from the target sample where the estimated posterior effect will be used for computing polygenic score. Thus, the GWAS sample and the target sample were treated fully independently in the polygenic score computation. Furthermore, in light of ours and previously published GWAS results, we assumed a highly polygenic genetic architecture for both MRI derived traits and sleep duration, by setting the parameter ϕ to 0.01, instead of a grid-search strategy proposed by the model. We believe our choice, though conservative, further reduce the overfitting risk. For other parameters in PRS-CS, we used the default values.

The posterior effect sizes obtained by running PRS-CS on each GWAS were then used separately to compute polygenic scores. We did not use p -values or LD threshold to select SNPs. Rather,

genome wide SNPs were used for computing the polygenic scores. After removing rare variants (minor allele frequency < 0.01) in UK biobank, variants not in the HapMap 3 data, and variants that are not on the Hardy-Weinberg equilibrium ($p < 10^{-6}$), we used the remaining 615297 SNPs for computing the polygenic scores for each trait. Recent methodology studies all point to the advantage of using shrinkage-based methods over p-value based thresholding methods, for example LDpred2, PRS-CS, and the lasso-based models³, particularly for highly polygenic traits. The computed posterior effects were used as weights in the computation of PGSs for a trait by using the score function from PLINK2. To examine the associations between PGS for a trait with a second trait, linear regression models were used. The same covariates included in the GWAS analysis were included as covariates in addition to PGSs in these models.”

6. Mendelian randomisation analyses/methods:

a. Instrument selection approach: this does not seem very conventional – usually instruments are chosen that meet genome-wide significance and if this is not the case then there needs to be a strong rationale for it. There are numerous papers that have used sleep duration SNPs as instruments in MR and I don't recall any of them using this approach. For the structural brain MRI instruments there are also published GWAS but these SNPs are perhaps less robust than the sleep ones (although they also have caveats, of course).

Response: We thank the reviewer for raising this issue. We agree with the reviewer that our strategies for selecting instruments are a bit liberal, i.e., selected SNPs at the suggestive level of association with exposures ($p < 10^{-6}$) instead of the GWAS threshold ($p < 5 \times 10^{-8}$). This is mainly because of the relative low power of our GWAS for sleep duration, which used at most one third of the sample size used in previous sleep-duration studies. The comparatively low-powered GWAS combined with high polygenicity and low heritability estimates for sleep duration yields very few genome-wide significant loci/SNPs (4 for short-sleeper and 3 for long-sleeper). These small numbers of instruments explain only a minor portion of the variance, and we are therefore unable to sufficiently 'randomize' exposure values. Hence, we had to use a trade-off between strength of the instrument and risk of horizontal pleiotropy, and therefore employed the suggestive p value thresholds for selecting instrument SNPs.

As we did not find significant causal effects of sleep duration on MRI-derived traits using $p < 10^{-6}$. We considered the instrument strengths to be still too low, exposure variance explained by instruments was 0.0014 for short-sleep and 0.0007 for long-sleeper, estimated by the Steiger test, and re-performed MR using SNPs with an association $p < 10^{-5}$, where the explained variances increased to 0.003 for short-sleeper and 0.005 for long-sleeper. We found no significant causal effects by this even more liberal threshold neither. Therefore, we concluded that with the current sample size, there was no detectable causal effect of sleep duration on brain structural variation. We cautioned the reader that future large studies may provide more convincing conclusion than ours.

For brain structural traits, a much larger portion of the variance was explained by the instruments at $p < 10^{-6}$ level (hippocampal volume: 0.058; ICV: 0.047; and total gray matter volume: 0.022). We found that larger ICV may be causally related to sleep duration for below-average sleepers

at this level of instrument selection. No other relations found. To make our analysis protocol consistent in both directions, we also performed the analysis with $p < 10^{-5}$. Here, the explained variance increased further, to about 0.097 for ICV. But again, neither hippocampal volume nor total gray matter volume showed any causal effect on either sleep duration. In our revised manuscript, we performed ICV vs sleep duration for shorter than average sleepers using the widely accepted instrument selection strategy to further corroborate our findings. We confirmed that our original findings are robust when using $p < 5 \times 10^{-8}$ as instrument selection threshold in terms of direction. However, the MR p -values become borderline significant ($p = 0.066$). This result is expected given that the explained variances drop to 0.022, yielding much weaker instrumental strength.

In summary, we confirmed that with the current sample sizes, no detectable causal effects from sleep duration to variation in MRI-derived measures were found, regardless of how stringent p -thresholds were used for selecting instruments. In contrast, the causal relationship from ICV to sleep duration in below-average sleepers are convincing using both instrument selection strategies.

Again, to make the methodology and our choices clear, the following is added to the manuscript:

Results section: "We performed bidirectional Mendelian randomization analysis for each brain volumetric trait to sleep duration. Among the 12 pairs, ICV showed a significant causal effect (inverse-variance weighted beta = 0.060, se = 0.017, $p = 5.36 \times 10^{-4}$) on sleep duration for the shorter than average sleepers (Figure 6a), and total gray matter volume showed an effect for the longer than average sleepers that did not survive correction for multiple comparisons (beta = -0.35, se = 0.14, $p = 0.012$, uncorrected, Figure 6b). Due to low heritability for sleep in the study, resulting in a weaker genetic instrument, we performed a robust MR analysis using robust adjusted profile score⁷¹ for the direction from sleep to brain traits, but did not detect significant causal relationships for the directionality of effects by this even more liberal threshold. The only significant causal relation was robust when we performed the analysis using both a stringent and a weaker instrument selection protocol. This suggests that there are no strong causal effects from sleep duration to brain morphometry."

SI Genetic analyses: "As we did not find significant causal effects of sleep duration on MRI-derived traits using $p < 10^{-6}$, we reasoned that the instrument strengths were still too low, and the exposure variance explained by instruments was 0.0014 for below-average and 0.0007 for above-average sleepers, estimated by the Steiger test, and reperformed MR using SNPs with an association $p < 10^{-5}$, where the explained variances increased to 0.003 for shorter than average sleepers and 0.005 for longer than average sleepers. We found no significant causal effects by this even more liberal threshold neither."

b. Did the authors clump by LD using their own sample as the reference? I don't think this is common either, unless that is not what they did, but this is not clear from the text.

Response: We apologize for this misunderstanding. We randomly selected 10 000 European participants from UK Biobank for LD clumping. We could have used an independent data set for this purpose, but the publicly accessible 1000 Genomes database has only 505 participants, which is widely considered as suboptimal for estimating LD structures for European populations. We have now clarified our strategy in our revised Method section, "Genetic analyses: Two-sample Mendelian randomization: The TwoSampleMR R package¹¹⁸ was used to investigate the existence of causal relations between sleep duration and the brain variables. Independent instrumental SNPs were selected using the following parameters: association p value $\leq 10^{-6}$, MAF ≥ 0.05 , LD- $r^2 \leq 0.1$, and LD-distance=10kb. The LD structure was derived from 10,000 independent European participants randomly selected from UKBB. The powerful inverse variance weighted (IVW) model from TwoSampleMR was used as the main model. Other models implemented in the software were also run as sensitivity analysis. To further support the results, the analysis was reperformed with p value $\leq 10^{-5}$, which would increase the strength of instrumenting for the less powerful sleep duration traits. For the only significant causal relation, i.e., ICV to sleep duration for shorter than average sleepers, a third analysis with $p \leq 5 \times 10^{-8}$ used for selecting instrumental SNP was performed. The standard output from TwoSampleMR is shown in Supplementary Note2."

c. What is the purpose of the sensitivity analyses with an even larger p-value threshold? This is likely to just introduce noise into the instrument.

Response: We agree that lowering instrument selection p-values could increase the chance of including noise than signals. However, for polygenic traits, it is widely agreed that most association signals fall short of GWAS significant levels. Properly including these sub-significant variants in MR can improve statistical power. In this case, lowering the threshold further strengthened the conclusion there were no causal effects from sleep duration to brain structures. We explain our logic further in our response to comment a. above.

d. To be clear, the authors' main MR analyses are one-sample, is this correct? They then perform 2-sample MR analyses but as a sort of side thing...

Response: We apologize for our unclear description. We used two-sample MR throughout this work. We have clarified this strategy in the revised manuscript and adding a section in Methods. See response to comment b. above.

e. While I appreciate that this is a long paper with quite a lot of different types of analyses in it, I think that there needs to be a proper description of the MR results, particularly the fact that there is

no mention of the 2-sample MR results in the main text, but this is the most popular way to do MR nowadays. There also needs to be some text telling the readers why the authors did what they did regarding for example, where the MR-Egger intercept p-value was <0.05 they then did some leave-one-out analyses (I'm assuming this is what this was at least) and perhaps then comment on what happened with the Egger intercept p-value after these analyses. They also do not comment anywhere on the inclusion of the heterogeneity statistics

Response: We thank the reviewer for these suggestions to make our work easier to comprehend. We have now updated the revised manuscript, including the information requested from the reviewer. See also responses to the comments above.

f. Can the authors also (instead of making the reader go to the rather large table in supplementary files) simply tell us what the key F-statistics and R² values were for their main instruments? These metrics for genetic instruments in MR are important and should be reported as standard in the text. The total instrument strength (R²) provides information on how well powered the analyses are likely to be.

Response: We again appreciate these suggestions. Since the genetic results are only one set of many different methods and results, we tried to keep the manuscript as short and focused as possible, but we realize that more information about the genetic analyses should have been included also in the main text. Thus, we have greatly expanded the description of the genetic analyses, results and the background for these, redesigned the genetic figures as well as included a longer discussion of the genetic effects. We hope the reviewer agrees with us that this has clearly strengthened the genetics in the manuscript.

g. When describing the 2-sample MR methods, the authors say that they used some popular methods, which is fine, but the main estimate is the IVW and the other methods such as Egger and WME are sensitivity tests for unbalanced horizontal pleiotropy.

Response: We thank the reviewer for pointing out the unclarity in our manuscript. We presented the IVW results in the main text and showed the results by other methods in our supplementary information. IVW is considered as the statistically most powerful MR methods assuming that the four IV assumptions (InSIDE assumption) are met. Other methods that to a certain extent can correct for conditions that failed the InSIDE assumption typically have lower power. It becomes difficult to tell whether non-significant results from these alternative models are consequences of low power or whether the significant relations from IVW are false positives due to failure of the InSIDE conditions.

Since we used MR as our explorative strategy for the sleep duration vs brain health relations after our main MRI vs sleep analysis, we leave the details in supplementary notes for the interested readers. See also responses to the reviewers' comments above.

Decision Letter, first revision:

9th January 2023

Dear Professor Fjell,

Thank you once again for your manuscript, entitled "Sleep duration and brain structure – phenotypic associations and genotypic covariance," and for your patience during the peer review process. I apologize for the delay in this round of review.

Your manuscript has now been evaluated by 3 reviewers, whose comments are included at the end of this letter. Reviewer #1 is one of the original reviewers, and they are now happy to recommend publication. Reviewers #2 and #3 were unable to re-review the manuscript. In light of the fact that this version of the manuscript differs significantly from the original version and includes additional data, we recruited two additional reviewers. Reviewer #4 has expertise in longitudinal analysis of MRI datasets, while Reviewer #5 has expertise in Mendelian randomization (to replace Reviewer #3).

You will see that Reviewer #4 and #5 find the study to be of interest, but raise a number of queries which will need to be addressed. We would like to consider your response to these concerns in the form of a revised manuscript before we make a decision on publication.

Finally, your revised manuscript must comply fully with our editorial policies and formatting requirements. Failure to do so will result in your manuscript being returned to you, which will delay its consideration. If you have any questions about any of our policies or formatting, please don't hesitate to contact me.

In sum, we invite you to revise your manuscript taking into account all reviewer and editor comments. We are committed to providing a fair and constructive peer-review process. Do not hesitate to contact us if there are specific requests from the reviewers that you believe are technically impossible or unlikely to yield a meaningful outcome.

We hope to receive your revised manuscript within 4 months. I would be grateful if you could contact us as soon as possible if you foresee difficulties with meeting this target resubmission date.

- Include a "Response to the editors and reviewers" document detailing, point-by-point, how you addressed each editor and referee comment. If no action was taken to address a point, you must provide a compelling argument. When formatting this document, please respond to each reviewer

comment individually, including the full text of the reviewer comment verbatim followed by your response to the individual point. This response will be used by the editors to evaluate your revision and sent back to the reviewers along with the revised manuscript.

- Highlight all changes made to your manuscript or provide us with a version that tracks changes.

[REDACTED]

We look forward to seeing the revised manuscript and thank you for the opportunity to review your work. Please do not hesitate to contact me if you have any questions or would like to discuss these revisions further.

Sincerely,
Jamie

Dr Jamie Horder
Senior Editor
Nature Human Behaviour

REVIEWER COMMENTS:

Reviewer #1:
Remarks to the Author:

I have no further concerns.

Reviewer #4:
Remarks to the Author:

In this interesting manuscript, the authors report on associations between sleep duration and regional brain volumes measured using questionnaires and MRI in very large-sized cross-sectional and longitudinal samples of adults using state-of-the-art methodology. The LIFEbrain consortium is a unique international effort. Accordingly, results do not indicate that sleep duration correlated with longitudinal measures of brain volumes. Also, genetic mendelian randomisation results do not offer evidence of causal relations between sleep duration and regional brain volume change across the adult lifespan. While the topic is timely and of high interest, and the neuroimaging samples are huge providing commendable statistical power, I have some comments that might need consideration and

limit my enthusiasm for the manuscript in its current form:

- Study rationale: While habitual sleep duration may be one proxy of sleep-related health, other aspects of sleep such as sleep quality might be of similar, or higher, importance: If I understood correctly, in previous studies published 2014 and 2020 by the same group with partly overlapping MRI samples, the authors report that 'significant self-reported sleep relates to hippocampal atrophy across the adult lifespan' based on several outcomes of sleep questionnaires. What was the rationale to focus on sleep duration only, and what are the implications of this? Do measures of sleep quality relate to longitudinal measures of brain volumes?

- With regard to longitudinal analyses, I wondered whether sleep duration was assessed at baseline only, or at multiple time points? From a public health perspective, which I understood is part of the study rationale, within-person effects might be more interesting to look at than between-person effects: Within-person effects offer the interpretation that a modification of in this case sleep duration indeed changes (or not) the outcome, plus if habitual sleep duration is a modifiable trait. This could be looked at with longitudinal measures only.

- MRI analysis: What was the rationale to perform ROI-based thickness analysis covering the whole brain, instead of whole-brain vertex/voxel-wise analyses? Averaging across ROIs might result in lower power.

- Did you consider to control for head motion in the analyses?

<https://www.ncbi.nlm.nih.gov/pmc/articles/PMC5217095/#>,

<https://braininformatics.springeropen.com/articles/10.1186/s40708-021-00128-2>, etc.

It might also be that head motion shares variance with sleep duration as it is the case for certain personality traits or psychiatric diseases.

- It has been noted that different scanners (even upgrades etc.) can produce non-linear differences in regional estimates of grey matter volume/thickness up to 4%

<https://doi.org/10.1371/journal.pone.0239021>. These regional inhomogeneities might have resulted in measureable confounding when analysing such large sample sizes. Did you consider to run a two-stage analysis, i.e. first whole-brain VBM or cortical thickness within cohorts (scanners) and second a meta-analysis of the different cohorts (scanners), or if not, what were the reasons?

- GAMM analysis:

- One of the benefits of mixed effects are the possibility to include single timepoints in longitudinal analysis, thereby reducing selection bias. Therefore I wondered why MRIs of one timepoint only were excluded.

- The GAMM smooth function analysis seems well suited for the question at hand. However, I wondered why both age and sleep duration would not be simply modelled with exponential terms?

- l. 180: if I understood GAMM in R and the information given in SI correctly, the code does not align with the manuscript text with regard to the main question. Specifically, did you indeed address whether a function of sleep duration - in variation dependent on time passed between MRIs - links to brain volumes (code = $s(\text{sleep}) \times \text{time}$)? or as written in the text, assessed whether a function of the interaction between sleep duration and time between MRIs affects brain volumes (text = $s(\text{sleep} \times \text{time})$)? This needs to be clarified as it affects interpretation of results massively. Please correct me if I got the syntax meaning wrong.

- in general, the description and SI results lack sufficient detail, for example with regard to set-up of the terms and baseline assumptions, etc. In my point of view, all effects and design specifications need to be given in the main manuscript, for example it needs to be written in the main text that random intercepts have been modelled for within-participants, not for further intercepts/slopes (if I understood the SI correctly).
- I could not understand whether (and how) the GAMM models were tested against null models to assess significance?
- Unfortunately the hundreds of pages long pdf-Supplementary Information is unreasonably demanding in its current form and needs a better structure and full code, not only the results outputs with some comments.

- Differences in (brain) health related to sex/gender are well described and we also know of the importance to consider sex and gender in the neurosciences. I therefore strongly recommend to conduct and report sex/gender-stratified cross-sectional and longitudinal analyses. Also, please define more accurately sex/gender throughout the manuscript, e.g. consider whether chromosome-assigned sex based on e.g. genotyping, gender assigned at birth, or current self-reported gender was available in the cohorts, and how diverse/non-binarity gender was assessed. With such large partly population-based sample sizes we can hope to at least contribute to increase representation of marginalized groups.

- I could not follow the rationale of the applied multiplicity control. Please explain in the main manuscript's methods part which outcomes/results constrain a family of tests, and why, and the rationale to apply FDR or FWE correction thresholds. If FDR, it would be more appropriate to report q-values instead of p-values.

- What was the rationale to include the given confounders? Did you consider to add cerebrovascular risk factors that partly link with sleep problems, e.g. type 2 diabetes, hypertension, etc.?

- Interpretation: Looking at the results, the inverted U-shaped association of sleep duration with ICV as noted by the authors is remarkably.
 - Did you check for any physical, physiological or anatomical underpinning of this relation? For example related to gradient distortion, pulse- or head movement-related associations? If this would have to do with genetics, we would expect an overlap of SNPs linked with both head size (or height) and sleep duration, or other mutual factors such as maybe birth weight or so.
 - If larger heads (or larger brains?) would promote 6-8h sleep duration, this could be tested with the alternative path of the Mendelian randomisation, if I am correct (SNPs for brain health/head size  brain thickness  sleep duration (u shaped or separate short/long-sleeper groups)
 - In any case, if this signal is related to image artifacts (whatever those may be) and/or genetic predisposition (whatever those may be), it would be very difficult to disentangle from any grey matter atrophy - sleep duration association. This might be considered as a limitation. Could one way around be to use yearly absolute or relative atrophy rates of e.g. hippocampus volumes per participant as outcome?

- Please indicate whether analyses were pre-registered and if/how the data and codes would be shared.
- Table 1, please add β s and effect size measures for meta-analyses
- L. 242 what are 'modest effect sizes'?

- Please add CIs to Figs. 3+4

Reviewer #5:

Remarks to the Author:

Thank you for the opportunity to review this interesting manuscript investigating the relationships between brain structure and sleep. As requested, I am only commenting on the genetics and MR analyses, which are in general sound and appropriate.

Comments:

GWAS:

Please provide a little bit more background on why you split the sample. You would have more power to detect SNP for sleep duration when combining the samples and use sleep duration as continuous outcome. This might be useful also for later MR analyses and sensitivity analyses.

MR:

Consider including the MR-STROBE reporting guidelines, which is best practice for MR studies. Here's the link: <https://www.strobe-mr.org/>.

Instrument selection: Please consider adding the number of SNPs selected for the instrument to the main manuscript and also include F-statistic for instrument strength. Related, the authors note that the sleep MR instruments may not explain enough of the variance, which they and they end up also incorporating a relaxed P-value threshold to construct their instruments included as sensitivity analyses. I would suggest a power calculation using <https://shiny.cnsgenomics.com/mRnd/> to assess whether their sleep brain MR analyses were sufficiently powered to detect associations. If these analyses are sufficiently powered, then it improves the inference of a null association in this direction.

I wonder if the stratification of the UK Biobank participants by sleep duration (≤ 7 hours versus ≥ 7 hours) may be missing a global effect. Perhaps it would be worthwhile to include as a sensitivity analysis MR of the overall sleep duration GWAS, which is publicly available and would be easy to run: [https://gwas.mrcieu.ac.uk/datasets/?trait__icontains=sleep duration](https://gwas.mrcieu.ac.uk/datasets/?trait__icontains=sleep+duration) (see also above for GWAS).

Please address the issue that some relationships between mental health/brain structure/sleep might be impacting the brain structure sleep MR findings. You could consider running a multivariate MR to account for major comorbidities, including BMI, psychiatric disorders and alcohol/tobacco use. Or Perhaps as a sensitivity analysis, you could perform a look up of the brain structure SNP instruments for their associations with major psychiatric disorders. Are there corresponding associations, for example, with depression? If not then it may be interesting to note. Conversely, if there are, then it may be worthwhile, to perform a quick sensitivity analysis MR where you construct a brain structure instrument leaving those variants out. Similarly, you could perform a multivariable MR analysis to further test the robustness of the finding: <https://wspiller.github.io/MVMR/articles/MVMR.html>

Author Rebuttal, first revision:

Response to the editors and reviewers

Thank you very much for all the constructive and very useful comments and suggestions, which clearly have improved the quality of the manuscript. Detailed responses to each are provided below.

In addition to changes in response to the comments from the reviewers, we have also decided to do a minor change in the meta-analysis of the cross-sectional subcortical brain data. We found that for some structures, such as the brain stem, there were some artifactual results at the upper extreme end of the sleep duration range, i.e. 10-12 hours of sleep. We have therefore decided to restrict sleep duration to 4-10 hours in the meta-analysis, which makes more sense. This is in accordance also with the existing figures in the manuscript, which shows the 4-10 hour range. The new sleep range has moved the results of the meta-analysis slightly downwards, with a new optimum at 6.5 hours.

Reviewer #1:

Remarks to the Author:

I have no further concerns.

Reviewer #4:

Reviewer 4: In this interesting manuscript, the authors report on associations between sleep duration and regional brain volumes measured using questionnaires and MRI in very large-sized cross-sectional and longitudinal samples of adults using state-of-the-art methodology. The LIFEbrain consortium is a unique international effort. Accordingly, results do not indicate that sleep duration correlated with longitudinal measures of brain volumes. Also, genetic mendelian randomisation results do not offer evidence of causal relations between sleep duration and regional brain volume change across the adult lifespan. While the topic is timely and of high interest, and the neuroimaging samples are huge providing commendable statistical power, I have some comments that might need consideration and limit my enthusiasm for the manuscript in its current form:

Response: *We appreciate the reviewer's positive evaluation of our manuscript.*

Reviewer 4 – Study rationale: While habitual sleep duration may be one proxy of sleep-related health, other aspects of sleep such as sleep quality might be of similar, or higher, importance: If

I understood correctly, in previous studies published 2014 and 2020 by the same group with partly overlapping MRI samples, the authors report that 'significant self-reported sleep relates to hippocampal atrophy across the adult lifespan' based on several outcomes of sleep questionnaires. What was the rationale to focus on sleep duration only, and what are the implications of this? Do measures of sleep quality relate to longitudinal measures of brain volumes?

Response: *The rationale for focusing on sleep duration was that this is the most basic and widely used measure in sleep studies, for which expert consensus recommendations exist, and which is the possibly most realistic for people to affect by lifestyle choices. To make our rationale clearer, we have added the following to the revised manuscript (Introduction): "Sleep duration was chosen as the sleep metric of focus because it is the most widely used, represents an aspect of sleep that for many people is under voluntary control, and constitutes the basis for most recommendations about sleep."*

Besides this, we fully agree with the reviewer that other aspects of sleep, such as sleep quality, also are of interest. In response to the reviewer's comment, we have therefore performed additional analyses to test this. For all samples except UKB, we calculated the PSQI global score following normal procedures, but removed the sleep duration component. For UKB, we calculated a sum of the different available sleep-related measures. We re-ran the main analyses controlling for this global sleep quality score. The description of this procedure is added to the manuscript (Methods, the new section Sleep measures): "For the HCP and the Lifebrain samples except Betula, sleep duration and other characteristics were measured by the Pittsburgh Sleep Quality Index (PSQI) 110. For Betula, sleep characteristics were measured by The Karolinska Sleep Questionnaire (KSQ) 111,112, which can be used to extract the same information covered by PSQI113. For UKB, sleep was measured through multiple questions. For all samples except UKB, we calculated the PSQI global score following normal procedures, but excluded the sleep duration component. For UKB, we calculated a sum score of different sleep-related measures (sleeplessness [field 1200], problems getting up in the morning [field 1170], daytime dozing [field 1220], snoring [field 1210] and chronotype [field 1180]). This global sleep quality score was used as covariate in follow-up analyses of brain – sleep duration relationships."

The results did not change notably when we controlled for the global sleep quality measure. For example, controlling for the sleep quality score did not weaken the sleep duration – brain change relationships in the longitudinal analyses, but the relationships for brain stem and putamen became significant when controlling for the global score. Likewise, only minor

differences in estimated sleep duration at peak volume were seen when controlling for the global sleep quality score. The new analyses are described in the revised manuscript, and the results summarized in Table 1 and the new figure 5, as can be seen below.

For the longitudinal analyses: “Controlling for the global sleep quality score did not weaken the duration – brain change relationships, but the relationships for brain stem and putamen became significant when controlling for the global score.”

For the cross-sectional analyses: “The analyses were also run controlling for the global sleep quality score (Figure 5). Except for thalamus, where sleep duration at maximum volume was reduced to the lower limit (4 hours) when controlling for global sleep quality, most peak estimates were similar for the default model vs. the model including global sleep quality as covariate.”

In addition, the new results are incorporated in Table 1 (longitudinal results) and presented in the new Figure 5 (longitudinal results):

	Range of sleep duration		Direction of effect	Controlling for additional covariates			
	Full range	Restricted (5-9h)		SES	BMI	Depression	Sleep quality
Accumbens	0.88	0.69	Negative	0.59	0.62	0.45	0.16
Amygdala	0.13	0.15		0.59	0.03	0.24	0.44
Brain stem	0.051	0.03		0.63	0.52	0.45	0.02
Caudate	0.02	0.07		0.66	0.02	0.00	0.02
CC anterior	0.98	0.98		0.69	0.83	0.88	0.99
CC central	0.70	0.98		0.90	0.42	0.67	0.30
CC mid ant	0.43	0.36		0.90	0.52	0.60	0.07
CC mid posterior	0.94	0.80		0.65	0.83	0.74	0.10
CC posterior	0.22	0.15		0.66	0.42	0.46	0.28
Cerebellum cortex	0.70	0.58		0.99	0.91	0.93	0.67
Cerebellum WM	0.01	0.01	Positive	0.65	0.01	0.04	0.00
Cerebral WM	0.22	0.15		0.65	0.83	0.76	0.44
ICV	0.70	0.52	0.65	0.83	0.85	0.99	
Hippocampus	0.43	0.72	0.64	0.52	0.45	0.44	
Pallidum	0.07	0.15	0.59	0.08	0.21	0.10	
Putamen	0.58	0.70	0.65	0.91	0.46	0.04	
Thalamus	0.01	0.01	Positive	0.59	0.03	0.20	0.00
TGV	0.70	0.70		0.59	0.42	0.46	0.89
Ventricles	0.02	0.02	Negative	0.90	0.44	0.45	0.00

Table 1 Associations between sleep duration and brain volumetric change.

P-values are adjusted using the Benjamini-Hochberg procedure, and all models were controlled for baseline age, sex, site and follow up time. Direction of effect negative: Longer sleep associated with greater volume loss. Direction of effect positive: Shorter sleep associated with greater volume loss. TGV: Total Gray Matter Volume. ICV: Intracranial volume. CC: Corpus callosum.

Figure 5 Sleep at maximum subcortical volume

The sleep durations associated with maximum subcortical volume are indicated by the dots. Only regions significantly related to sleep duration shown. Error bars indicate 95% CI. The default model is shown in red and the model including global sleep quality as covariate is shown in turquoise.

Finally, these new results are also referred to in the discussion.

Reviewer 4 – With regard to longitudinal analyses, I wondered whether sleep duration was assessed at baseline only, or at multiple time points? From a public health perspective, which I understand is part of the study rationale, within-person effects might be more interesting to look at than between-person effects: Within-person effects offer the interpretation that a modification of in this case sleep duration indeed changes (or not) the outcome, plus if habitual sleep duration is a modifiable trait. This could be looked at with longitudinal measures only.

Response: This is a very relevant comment. Unfortunately, data on sleep was available from baseline only for most participants, so we did not have the opportunity to run well-powered change-change analyses. For the few participants from whom more than one observation of sleep was available, we used the average score in the statistical models. We have previously seen high correlations between reported sleep across timepoint, so we believe the baseline measure is a good proxy of the participants’ sleep during the study. To make this point clear, the following was added to the manuscript (Results, section on Longitudinal sleep – brain atrophy associations): “As sleep duration was available for one timepoint only for most of the participants, we used the average value across timepoints for the small number of participants for whom more than one observation was available.”

Reviewer 4 – MRI analysis: What was the rationale to perform ROI-based thickness analysis covering the whole brain, instead of whole-brain vertex/voxel-wise analyses? Averaging across ROIs might result in lower power.

Response: *We apologize that this was not clear in the manuscript. We did perform vertex-wise analyses in addition to the ROI-analyses, but these were removed from the main manuscript before submission due to a worry about information overload. The vertex-wise maps were instead placed in the Supplemental Information (SI Cortical Vertex analyses) and referred to in the results section (page 10 and 11 in the revised manuscript) and in the discussion (page 21 in the revised manuscript): “The vertex-wise cortical thickness analyses (Supplemental Information) suggested that relationships with sleep duration were different in younger and older adults, but this was not confirmed in the GAMM analyses. Thus, we believe the meta-analytic results represent a good approximation of a general sleep – brain volume relationship on a group level, while ignoring that there naturally are variations across brain regions, age and participants.”*

The reason we chose a region-based instead of a vertex-based approach for the main analyses was that we expected a non-linear and non-monotonous relationship between sleep and brain characteristics. With statistical surface maps, these relationships are difficult to visualize. Further, we did not expect highly localized effects, which means that the region-based analyses probably were as sensitive as the vertex-wise analyses. The ROI-based approach also allowed us to use the exact same statistical models for the cortical and the subcortical regions. To accommodate the reviewer’s comment, we have expanded the description of the vertex-results, by including the following in the revised manuscript (Results, section on Cross-sectional sleep – brain morphometry associations): “The vertex-wise analyses confirmed this general finding, showing only positive relationships between sleep duration and thickness in the below-average sleepers, and only negative relationships in the above-average sleepers (SI Cortical vertex Analyses).”

Reviewer 4 – Did you consider to control for head motion in the analyses? <https://www.ncbi.nlm.nih.gov/pmc/articles/PMC5217095/#>, <https://braininformatics.springeropen.com/articles/10.1186/s40708-021-00128-2>, etc.

It might also be that head motion shares variance with sleep duration as it is the case for certain personality traits or psychiatric diseases.

Response We agree with the reviewer that motion during scanning can affect the structural FreeSurfer outputs. Unfortunately, we are not able to derive any direct measure of motion from the T1-scans alone, except removing scans with visible motion artifacts or otherwise failed our quality check procedure. A proxy based on fMRI scans could have been calculated

as suggested in one of the references provided by the reviewer, but this was not available for all participants. Still, we believe it is unlikely that head motion affected the present results, but now acknowledge the reviewer's point in the revised manuscript (Discussion, section on Caveats and limitations): "...we had no quantitative measure of head motion, so to the extent head motion correlated with sleep duration, this could potentially be a confounder."

Reviewer 4 - It has been noted that different scanners (even upgrades etc.) can produce non-linear differences in regional estimates of grey matter volume/thickness up to 4% <https://doi.org/10.1371/journal.pone.0239021>. These regional inhomogeneities might have resulted in measureable confounding when analysing such large sample sizes. Did you consider to run a two-stage analysis, i.e. first whole-brain VBM or cortical thickness within cohorts (scanners) and second a meta-analysis of the different cohorts (scanners), or if not, what were the reasons?

Response We agree with the reviewer that these are important issues, and we have taken great care to ensure that scanner differences have not affected the results. Importantly, we scanned seven participants on seven of the scanners used in the Lifebrian cohorts, allowing us to compare results across scanners. In the previous version of the manuscript, between-scanner correlations for hippocampal volume only were provided. We have now added additional results for five additional cortical volumetric ROIs. In short, there was excellent correspondence across scanners, which means that absolute differences in volume between scanners very likely will be effectively removed by using scanner as a random effects term in the analyses, without biasing the results. The following is included in the revised manuscript (Methods, section on Magnetic resonance imaging): "To assess the influence of scanner on volumetric estimates, seven participants were scanned on seven scanners across the consortium sites (see 26 for details). Using hippocampus as test-region, there was a significant main effect of scanner on volume ($F = 4.13$, $p = .046$), but the between-participant rank order was close to perfectly retained between scanners, with a mean between-scanner Pearson correlation of $r = .98$ (range .94-1.00). Analyses of five additional volumetric cortical and subcortical ROIs (medial temporal lobe [entorhinal & parahippocampal cortex], precuneus, superior temporal, caudate nucleus, caudal middle frontal) showed correlations close to 1.0 for all regions except MTL, where correlations were somewhat lower but still $> r = .75$ (see 110). Thus, including site as a random effect covariate in the analyses is likely sufficient to remove the influence of scanner differences."

Still, we believe the reviewer's suggestion of a two-stage approach is good, and we therefore performed a meta-analysis of the different cohorts/ scanners for the three cortical ROIs reported in the manuscript. The results are shown below for thickness, volume and area. These clearly demonstrate that our mega-analysis yields close to identical results compared

to the two-stage analysis. The following is added to the manuscript (Results, section on Cross-sectional sleep – brain morphometry associations): “To exclude the possibility that the use of different scanners influenced the results, we also used a two-stage approach. First, we ran the thickness – sleep duration GAMMs separately in each sample, and then performed meta-analysis of the different cohort results. This yielded very similar estimates, demonstrating that the use of different scanners did not bias the results (SI Mega vs meta-analytic approach).”

Reviewer 4- GAMM analysis:

- One of the benefits of mixed effects are the possibility to include single timepoints in longitudinal analysis, thereby reducing selection bias. Therefore I wondered why MRIs of one timepoint only were excluded.

Response *The reviewer is indeed right that mixed models handle missing data well. Therefore, all available data were included in the cross-sectional analyses. For the longitudinal analyses, participants with only one time-point would not improve the change-estimations, and we therefore ran these on the longitudinal data only. We agree with the reviewer that selective dropout can bias the results, but in most cases, participants with only one time point were not dropouts but a result of study design. To be sure that our procedure did not affect the results, we re-ran the longitudinal analyses on the full dataset including all available observations, and the results were identical. We first included this in the Supplemental Information, but then removed these results again to keep the Supplemental Information more focused (see also response to comment below).*

Reviewer 4 - The GAMM smooth function analysis seems well suited for the question at hand. However, I wondered why both age and sleep duration would not be simply modelled with exponential terms?

Response *We agree that exponential terms could potentially also work. The great advantage using GAMM is that we do not need to make any specification of the functional shape of the relationships, which are allowed take any form of any complexity. In addition, GAMM is a local fit model, which means that the curve at one point is less influenced by sampling at distant points. For details, please see our previous papers on the use of GAMMs in lifespan brain research (e.g. Sørensen, Walhovd, Fjell, 2020, A recipe for accurate estimation of lifespan brain trajectories, distinguishing longitudinal and cohort effects, NeuroImage). In the revised ms, we have added a justification for the use of GAMM (section on Statistical analyses): "GAMM offers an attractive alternative to linear mixed models in that a priori specifications of polynomial functional forms are not necessary, and GAMMs are able to accurately fit trajectories of different forms and complexities62."*

Reviewer 4 - l. 180: if I understood GAMM in R and the information given in SI correctly, the code does not align with the manuscript text with regard to the main question. Specifically, did you indeed address whether a function of sleep duration - in variation dependent on time passed between MRIs - links to brain volumes (code= s(sleep)xtime)? or as written in the text, assessed whether a function of the interaction between sleep duration and time between

MRI affects brain volumes (text = s(sleep x time))? This needs to be clarified as it affects interpretation of results massively. Please correct me if I got the syntax meaning wrong.

Response We agree that this was not clear in the previous version of the manuscript, as the syntax is not completely intuitive. We assume this is the relevant part from the SI.

```
mod <- gamm4(
  value ~ sex + site + icv + s(bl_age, k = 10, bs = 'cr') +
    s(bl_age, by = sleep_z, bs = 'cr') +
    s(bl_age, by = time, k = 5, bs = 'cr') +
    s(sleep_z, by = time, k = 5, bs = 'cr', pc = 0),
  random = ~(1|id), data = long_dat, REML = FALSE
)
```

Time here denotes time since baseline, and hence represents individual aging, i.e. the longitudinal effects of aging on brain volumes. The reason we split it this way is that the baseline age term mostly will capture cross-sectional effects of age and the time term will capture longitudinal effects of aging (i.e., average effect of aging within a given birth cohort). We use linear terms for the effect of time, as it is reasonable to assume that the effect of aging is well approximated by a linear function in the relatively short follow-up intervals. The relevant term for the reviewer's comment is $s(\text{sleep_z}, \text{by} = \text{time}, k = 5, \text{bs} = \text{'cr'}, \text{pc} = 0)$. This represents the interaction between the effect of sleep duration and the effect of time. The point constraint $\text{pc} = 0$ ensures that this term is identically zero at $\text{sleep_z} = 0$. By definition it will also be identically zero at $\text{time} = 0$, since time is entered as a linear term. This again means that it is a pure interaction term. It is hence correct as is written in the manuscript, that this term represents "the effect of sleep on brain change". Perhaps more statistically framed, we can say that it is the effect of sleep on the longitudinal effect of aging on the brain measure. In a linear model, this would be $\beta \times \text{sleep} \times \text{time}$, but we here allow the value of this interaction to depend smoothly on sleep duration, so we have a varying-coefficient model which is typically written as $\beta(\text{sleep}) \times \text{time}$. We have now attempted to clarify this and other statistical unclarities by expanding the description of the statistical procedures in the manuscript. See response to the reviewer's comment below for a detailed account of the changes made.

Reviewer 4 - in general, the description and SI results lack sufficient detail, for example with regard to set-up of the terms and baseline assumptions, etc. In my point of view, all effects and design specifications need to be given in the main manuscript, for example it needs to be written in the main text that random intercepts have been modelled for within-participants, not for further intercepts/slopes (if I understood the SI correctly).

Response: We agree that the descriptions of the statistical procedures and design specifications, as well as the descriptions in SI, were not sufficiently clear. In addition to revising the SI, among other things by including the full model specification and syntax for the analyses, we have added substantially to the main text, which hopefully makes the procedures clearer. Specifically, the following is included in the revised manuscript: Results, section Longitudinal sleep – brain atrophy associations:

“For each measure y , we ran the following model for the i^{th} observation of the j^{th} participant:

$$y_{ij} = f(\text{age}_{bl,j}) + \beta_1(\text{age}_{bl,j}) \times \text{sleep}_j + \beta_2(\text{age}_{bl,j}) \times \text{time}_{ij} + \beta_3(\text{sleep}_j) \times \text{time}_{ij} + \text{covariates}_{ij} + b_j + \varepsilon_{ij}$$

Here, $f(\text{age}_{bl,j})$ is a smooth function of age at baseline $\text{age}_{bl,j}$. Next, $\beta_1(\text{age}_{bl,j})$, $\beta_2(\text{age}_{bl,j})$, and $\beta_3(\text{sleep}_j)$ are varying-coefficient terms which depend smoothly on their argument⁶⁶.

All smooth terms were constructed with cubic regression splines and penalized based on their squared second derivatives. The term time_{ij} denotes the time since baseline at the i^{th} timepoint of the j^{th} participant, and sleep_j denotes the sleep duration of the j^{th} participant.

The first three smooth terms serve to control for the effect of age on the brain measure, the cross-sectional (between-participant) effect of sleep on the brain measure, and how the effect of time depends on age, respectively. The fourth term, $\beta_3(\text{sleep}_j) \times \text{time}_{ij}$, is of main interest, since it describes how the effect of time depends on sleep duration. Baseline age, self-reported sex, site and follow up time were used as covariates. ICV was included as covariate of no interest in the volumetric analyses. Finally, b_j is a random intercept term for participant j and ε_{ij} is a residual, both assumed normally distributed. The model was estimated using maximum marginal likelihood.”

And

Results, section Cross-sectional sleep – brain morphometry associations:

“We ran the model

$$y_{ij} = f(\text{age}_{ij}, \text{sleep}_j) + \text{covariates}_{ij} + b_j + \varepsilon_{ij}$$

for each brain variable, where y_{ij} denotes the volume/thickness of participant j at timepoint i , $f(\text{age}_{ij}, \text{sleep}_j)$ is a tensor interaction term constructed with cubic regression splines according to⁷⁰, covariates are sex, site, and (for volumetric analyses) ICV. b_j are random intercepts and ε_{ij} are residuals. Note that although the estimated effects are cross-sectional, all available data were used, and hence random intercepts were included. The full model was compared to two reduced models: First a model in which the tensor interaction term was

replaced by two additive terms $f_1(\text{age}_{ij})$ and $f_2(\text{sleep}_j)$, and second a model in which sleep was completely removed. As these models are nested, comparison in terms of likelihood ratio tests is valid. We hence based model selection on a likelihood ratio test with 5% significance level.”

Reviewer 4 – I could not understand whether (and how) the GAMM models were tested against null models to assess significance?

Response We apologize that this was not clearly explained, and hope that it is now clear from the above. Specifically, the following addresses the reviewer’s comment: “As these models are nested, comparison in terms of likelihood ratio tests is valid. We hence based model selection on a likelihood ratio test with 5% significance level.”

Reviewer 4 – Unfortunately the hundreds of pages long pdf-Supplementary Information is unreasonably demanding in its current form and needs a better structure and full code, not only the results outputs with some comments.

Response We agree with the reviewer that the amount and organization of the information in SI was not optimal. Thus, we worked to organize it in a more efficient way, while keeping a lot of the details for the interested readers. So, each of the long SI sections start with a headline and a table of content. In addition to page numbers, we also use hypertext so one can click on the information of interest and get directly to the results. After a presentation of the data and the statistical model (with code), the results for each brain region is presented, with separate headlines for each type of analysis (e.g. for the different covariates). For instance, for SI Subcortical Measures Longitudinal, page 1 looks like this:

Subcortical volumes longitudinal

Contents

Introduction	7
Data	7
Models	7
Results per region	8
Accumbens-area	8
Main analysis	8
Full data	8
Restricted to sleep ≥ 5 and ≤ 9 hours	9
Controlling for socioeconomic status	10
Full data	10
Restricted to sleep ≥ 5 and ≤ 9 hours	12
Controlling for BMI	13
Full data	13
Restricted to sleep ≥ 5 and ≤ 9 hours	15
Controlling for depression	16
Full data	16
Restricted to sleep ≥ 5 and ≤ 9 hours	18
Controlling for sleep quality	19
Full data	19
Restricted to sleep ≥ 5 and ≤ 9 hours	21

Then if one is interested in inspecting the general model run, one can click on “Models”, which takes the reader directly to this:

Models

The following GAMM was fitted:

```
mod <- gamm4(
  value ~ sex + site + icv + s(bl_age, k = 10, bs = 'cr') +
    s(bl_age, by = sleep_z, bs = 'cr') +
    s(bl_age, by = time, k = 5, bs = 'cr') +
    s(sleep_z, by = time, k = 5, bs = 'cr', pc = 0),
  random = ~(1|id), data = long_dat, REML = FALSE
)
```

Here is an explanation of each of the smooth terms. The number of “knots” was chosen relatively low for the sake of computational speed, but increasing this would have minimal impact on the results, as the estimated degrees of freedom is well below the maximum provided by the `k`.

- `s(bl_age, k = 10, bs = 'cr')` models the main effect of baseline age. In a linear model, the equivalent would be `bl_age`.
- `s(bl_age, by = sleep_z, bs = 'cr')` models the effect of sleep and how this effect varies with baseline age. In a linear model, the equivalent to this term would be `sleep_z + bl_age:sleep_z`.
- `s(bl_age, by = time, k = 5, bs = 'cr')` models how the effect of time depends on baseline age. In a linear model, the equivalent to this term would be `time + bl_age:time`.
- `s(sleep_z, by = time, k = 5, bs = 'cr', pc = 0)` models the interaction between time and sleep. The argument `pc = 0` constrains this term to not include a main effect of `time`, as this is covered by the previous term. In a linear model, the equivalent to this term would be `sleep_z:time`. This is the term of main interest.

Then, if one is interested in e.g. Accumbens-area results from the main analysis, one can click on the variable name, which takes the reader to the code and results for each analysis involving Accumbens-area:

Accumbens-area

Main analysis

Full data Below is the model output. The term `s(sleep_z):time` is probably what we care most about.

```
##
## Family: gaussian
## Link function: identity
##
## Formula:
## value ~ sex + site + icv + s(bl_age, k = 10, bs = "cr") + s(bl_age,
##   by = sleep_z, bs = "cr") + +s(bl_age, by = time, k = 5, bs = "cr") +
##   s(sleep_z, by = time, k = 5, bs = "cr", pc = 0)
## <environment: 0x55779f17e8e8>
##
## Parametric coefficients:
##           Estimate Std. Error t value Pr(>|t|)
## (Intercept)   925.987    10.605  87.319 < 2e-16 ***
## sexmale        14.029     5.443   2.577  0.00998 **
## siteousAvanto 114.136    13.087   8.722 < 2e-16 ***
## siteousPrisma -66.977    42.893  -1.561  0.11845
## siteousSkyra  274.062    12.714  21.557 < 2e-16 ***
## siteUB        -24.983    24.241  -1.031  0.30274
## siteUCAM     -100.142    13.242  -7.562  4.39e-14 ***
## siteUKB       -10.489    10.089  -1.040  0.29852
## siteUmU       289.790    15.136  19.145 < 2e-16 ***
## icv           57.456     2.656  21.632 < 2e-16 ***
## ---
## Signif. codes:  0 '***' 0.001 '**' 0.01 '*' 0.05 '.' 0.1 ' ' 1
##
## Approximate significance of smooth terms:
##           edf Ref.df      F p-value
## s(bl_age)      4.34  4.34 490.591 <2e-16 ***
## s(bl_age):sleep_z 2.00  2.00  0.522  0.593
## s(bl_age):time  2.00  2.00 275.339 <2e-16 ***
## s(sleep_z):time 1.00  1.00  0.070  0.791
## ---
## Signif. codes:  0 '***' 0.001 '**' 0.01 '*' 0.05 '.' 0.1 ' ' 1
##
## R-sq.(adj) =  0.58
## lmer.REML = 98394 Scale est. = 2920.6    n = 8153
```

(followed by plots showing the shape of the different effects).

We realize that SI still contains much information, but we have decided to keep a lot of the details in to maximize transparency, and to allow the interested reader to inspect the detailed

procedures and results. Following the descriptions in the SI, all the results in the paper can be replicated, given the data.

We hope the reviewer find the new and improved SI acceptable, although we realize that it is still covers many pages. Therefore, we have added a more detailed description of how to navigate in the Supplementary Information in the main text. We now include a section named Transparency, placed at the beginning of Methods & Protocols. Here we explain the rationale and the structure of the Supplemental Information. The following text was added to the manuscript: "Transparency - The current work contains many analyses and analytic choices, which may affect the results. This regards for instance which covariates that are included in the different analyses, exclusion of outliers and restriction of data ranges (e.g. for sleep duration), model specifications and model selection. This information is too extensive to fit in the main text. To optimize transparency, we have included these details in SI. An overview is provided in Table 3."

SI filename	Page	Description
Extended background information		
Reviewed studies	5	Reference, sample description and main results of 19 studies testing sleep duration – brain morphometric relationships in adults
MRI methods	26	Overview of scanners and sequence parameters for each sample, description of MRI processing
Sample characteristics	5, 25	Description of recruitment, inclusion/exclusion criteria and cognitive testing in each sample
Statistical analyses and results		
Cortical Longitudinal	8, 9	R-code and results for each of 32 cortical regions -thickness, area, volume (longitudinal) -stratified by sex -full range of sleep duration vs. 5-9 hours
Subcortical Longitudinal	9	R-code and results for each of 19 regions -volume (longitudinal) -full range of sleep duration vs. 5-9 hours -effects of covariates: SES, BMI, depression, global sleep quality
Cortical cross-sectional	6, 9, 10	R-code and results for each of 32 cortical regions -thickness, area, volume (cross-sectional) -cortical clustering procedures & results for each metric
Subcortical cross-sectional	13	R-code and results for each of 19 regions -volume (cross-sectional) -raw volumes vs. controlling for ICV -effects of covariates: SES, BMI, depression, global sleep quality
Meta-analysis	13	R-code and results for the meta-analysis -subcortical volume and cortical thickness (cross-sectional) -raw volumes vs. controlling for ICV
Cortical vertex analyses	10, 11	Vertex wise results -cortical thickness

Mega vs meta-analytic approach	11	-compare results from mega- vs meta-analysis
Genetic analyses and results		
Genetic analyses	13, 14	-GWAS procedures & results -Calculation of PGSs -Mendelian randomization procedures & results
Genetics notes	13	-Detailed Mendelian randomization results -Comparisons of different Mendelian randomization methods
Genetics tables	13	-Detailed SNP-level results

25

STROBE-MR-checklist	15	- Checklist of recommended items to address in reports of Mendelian randomization studies
Instrumental variables	16	Tables of the instrumental variables for the MR analysis (ICV → sleep duration)

Table 3 Overview of Supplemental Information (SI) adding background and details to the main manuscript, including R-code.

Reviewer 4 – Differences in (brain) health related to sex/gender are well described and we also know of the importance to consider sex and gender in the neurosciences. I therefore strongly recommend to conduct and report sex/gender-stratified cross-sectional and longitudinal analyses.

Response We have followed the reviewer's advice, adding sex-stratified analyses to the manuscript. For the longitudinal cortical analyses, there were no relationship between sleep duration and brain change for either sex. This is now described in the Results, section on «Longitudinal sleep - brain atrophy associations»: "In the main analyses, sex was included as a regressor. We also ran separate analyses for males and females, still yielding no significant relationships with thickness change for any cortical region (see SI Cortical longitudinal))."

The same analyses were run for the cross-sectional data. For cortical thickness, the pattern of effects was very similar between males and females, although cuneus, lateral orbitofrontal, lateral occipital, fusiform and entorhinal cortices showed significant relationships in males but not females. Among these regions, a formal interaction analysis showed a significant effect of sex only for the lateral orbitofrontal cortex. The following is included in the manuscript (Results, section Cross-sectional sleep-brain morphometry associations: "Splitting the analyses by sex, five regions showed significant sleep-thickness relationships in males only (cuneus, lateral orbitofrontal, lateral occipital, fusiform, entorhinal). A formal sex-interaction analysis of these regions showed a significant effect of sex only for lateral orbitofrontal cortex, where very short and very long sleep was more associated with thinner cortex in males than females."

Reviewer 4 Also, please define more accurately sex/gender throughout the manuscript, e.g. consider whether chromosome-assigned sex based on e.g. genotyping, gender assigned at birth, or current self-reported gender was available in the cohorts, and how diverse/non-binarity gender was assessed. With such large partly population-based sample sizes we can hope to at least contribute to increase representation of marginalized groups.

Response Unfortunately, none of the samples included information on gender, only self-reported biological sex. For the UKB genetic analyses, discrepancy between biological and self-reported sex leads to exclusion of the participant as part of the quality control procedure done by the UKB genetics team before the data is released to researchers, so we do not have access to this information. We have replaced "sex" with "self-reported sex" at the first use in the manuscript.

Reviewer 4 - I could not follow the rationale of the applied multiplicity control. Please explain in the main manuscript's methods part which outcomes/results constrain a family of tests, and why, and the rationale to apply FDR or FWE correction thresholds. If FDR, it would be more appropriate to report q-values instead of p-values.

Response We have now clarified the use of multiple comparison corrections in the manuscript. The set of all subcortical ROIs constrained one family of tests, and the set of all cortical ROIs (for measure of thickness) constrained another family of tests. We chose to use FDR methods because FWE correction methods like Bonferroni are very strict and would lead to serious loss of power. With FDR methods, we know that the expected proportion of false discoveries is 0.05, which we consider acceptable. We used the Benjamini-Hochberg procedure, whereas Q-values as suggested by reviewer 4 are defined for the Storey-Tibshirani procedure. To make our choices clearer, we have added the following to the manuscript (Methods, section Statistical analyses): “FDR was used to adjust p-values for multiple comparisons, because family wise error correction methods like Bonferroni are very strict and would lead to serious loss of power. With FDR methods, we know that the expected proportion of false discoveries is 0.05, which we consider acceptable.”

Reviewer 4 – What was the rationale to include the given confounders? Did you consider to add cerebrovascular risk factors that partly link with sleep problems, e.g. type 2 diabetes, hypertension, etc.?

Response The main confounders included, in addition to age and sex, SES, BMI and depression. Both BMI and depression have been found to be related to sleep, in that higher BMI is reported to be related to more sleep problems and depression has been reported to be related to different form of sleep disturbances. We now justify this in the manuscript, and have also included citations of three additional studies supporting this (Results, section on Longitudinal sleep – brain atrophy associations): “Post hoc analyses were run controlling for socioeconomic status (SES: income and education), BMI, depression symptoms and a measure of global sleep quality in turn as covariates, as these variables may affect sleep duration, brain structure and possibly the relationship between them^{70–72}.”

When it comes to cardiovascular risk, we had access to other measures than BMI related to cardiovascular risk factors for part of the sample only. Since we agree that this is relevant, we have added this as a limitation (Discussion, section on Caveats and limitations): “...a number of covariates that could potentially influence the sleep-brain relationships were not controlled for, including cardiovascular risk factors other than BMI.”

Reviewer 4 – Interpretation: Looking at the results, the inverted U-shaped association of sleep duration with ICV as noted by the authors is remarkably.

Response Yes, we agree with the reviewer that this is interesting, as ICV does not change in adults. We believe this serves to strengthen our point that non-causal factors may be

involved in often-observed cross-sectional sleep-brain relationships. The discussion of the ICV results is expanded in the revised manuscript (see response to the next comment).

Reviewer 4 - Did you check for any physical, physiological or anatomical underpinning of this relation? For example related to gradient distortion, pulse- or head movement-related associations? If this would have to do with genetics, we would expect an overlap of SNPs linked with both head size (or height) and sleep duration, or other mutual factors such as maybe birth weight or so.

Response *Yes, we tested several possible explanations for this. Height and BMI was only weakly related to sleep duration, and the sleep-ICV relationship was not due to body size. PGS for ICV was significantly associated with sleep duration in the shorter than average sleepers, and PGS for sleep duration was associated with ICV in the same group, so there is an overlapping genetic foundation for the relationship. This is described in the Results (section "GWAS, polygenic scores, genetic correlations and Mendelian randomization") and in the Discussion (section "Genetic associations"), see also response below.*

Reviewer 4 - If larger heads (or larger brains?) would promote 6–8h sleep duration, this could be tested with the alternative path of the Mendelian randomisation, if I am correct (SNPs for brain health/head size  brain thickness  sleep duration (u shaped or separate short/long-sleeper groups))

Response *The Mendelian randomization analyses suggested a causal effect of ICV on sleep duration. PGS for total brain matter volume, controlling for ICV, was related to sleep duration in the shorter than average sleepers, while the inverse relationship was weaker. However, the Mendelian randomization analysis did not reveal causal significant relationships between brain volume and sleep duration. Thus, although we agree that the reviewer's suggestion could be a plausible path of influence, the results do not provide clear evidence for this. We have expanded the reporting and discussion on the ICV-sleep relationship in the manuscript in the revised manuscript, and hope this may contribute to clarify the potential meaning of these relationships. The most relevant parts of the revised manuscript read: Results, section "PGSs for ICV (PGS-ICV: $t = 8.47$, $p_{fdr} = 2.4 \times 10^{-15}$) and total gray matter volume (PGS-TGV: PGS-TGV, $t = 4.65$, $p_{fdr} = 3.28 \times 10^{-5}$) were significantly associated with sleep duration in the shorter than average sleepers (Figure 5a, c). PGS for sleep duration in the shorter than average sleepers was significantly related to ICV ($t = 6.99$, $p_{fdr} = 3.03 \times 10^{-11}$, Figure 5b) and to a lesser extent with total gray matter volume ($t = 2.69$, $p_{fdr} = 6.42 \times 10^{-2}$, Figure 5d). No significant associations were identified for other pairs of traits."*

And

“We performed bidirectional Mendelian randomization analysis for each brain volumetric trait to sleep duration (see SI STROBE-MR-checklist for details, reporting according to best practice for MR studies). Among the 12 pairs, ICV showed a significant causal effect (34 instrumental SNP, minimal F stats > 24; inverse-variance weighted beta = 0.060, se = 0.017, $p = 5.36 \times 10^{-4}$) on sleep duration for the shorter than average sleepers (Figure 6 and SI Instrumental variables), with no causal effects of sleep on ICV. Total gray matter volume showed a trend level effect for the shorter than average sleepers ($p = 0.12$).”

In the Discussion, section “Sleep duration and the brain”:

“In this regard, the association between ICV and sleep duration is interesting. ICV was the MRI-derived measure most positively associated with sleep duration, and the Mendelian randomization analysis suggested a causal effect of ICV on sleep duration in the shorter than average sleepers but not the inverse. As sleep has no causal effect on ICV in adults, this relationship must reflect other factors, and demonstrates that associations between sleep duration and MRI-derived volumes may reflect non-causal and stable relationships which do not emerge as a function of variations in sleep duration. The partly common genetic underpinning of ICV and sleep duration suggests that there may be a mechanistic association, but this is not caused by sleep. Controlling for ICV removes the effect of global scaling, i.e. that regional brain volumes scale with head size. Since ICV is sometimes regarded as a proxy for maximal brain size, controlling for ICV yields regional volumes representing deviations from the expected based on head size. Controlling for ICV also control to some extent for body size, as head size and body size are normally related, although height and BMI were weakly related to sleep duration in the present data. Not controlling for ICV naturally led to a higher sleep duration estimate of 7.4 hours associated with maximal brain volumes and thickness, as ICV and sleep duration were positively related.”

And section “Genetic associations”:

“Finally, the Mendelian randomization analyses showed causal effects of ICV on sleep duration, an effect that was robust after accounting for confounding factors (BMI, smoking and drinking habits, neuropsychiatric disorders, see SI). However, there were no causal effects of sleep duration on any MRI-derived brain measure. Hence, in the current samples, people with larger heads on average report to sleep longer, and this relationship partly depends on genetics. The lack of an inverse influence of sleep duration on ICV was given, as ICV does not change in adults and hence cannot be affected by sleep. Still, the genetic results suggested that there may be a mechanistic relationship between ICV and sleep duration which could warrant further explorations. This effect was removed from the estimated sleep duration – brain volume relationships by covarying for ICV, which may contribute to explain that the nominally significant relationship between total gray matter volume and sleep duration in the Mendelian randomization analysis did not survive corrections. In sum, the genetic results were in coherence with a view of average and “optimal” sleep duration as

relatively well aligned and did not provide evidence for a causal relationship of sleep duration on brain structural features.”

Reviewer 4 – In any case, if this signal is related to image artifacts (whatever those may be) and/or genetic predisposition (whatever those may be), it would be very difficult to disentangle from any grey matter atrophy – sleep duration association. This might be considered as a limitation. Could one way around be to use yearly absolute or relative atrophy rates of e.g. hippocampus volumes per participant as outcome?

Response *In the longitudinal analyses, volume change was captured by the sleep duration x time interaction, controlling for absolute volume or thickness of the region in question. This would then be similar to relative atrophy rate. Here we found weak relationships between sleep duration and brain change. Hence, we are convinced that the relationship between ICV and sleep duration represents stable factors. This also serves to suggest that the other sleep duration – brain volume relationships observed in the cross-sectional analyses at least to a certain degree may reflect stable factors. Although we agree with the reviewer that image artefacts may affect any observed brain-relationship, we cannot think of any specific type of artifact that could cause the present relationship between ICV and sleep duration. See responses to comment above for a more thorough discussion.*

Reviewer 4 – Please indicate whether analyses were pre-registered and if/how the data and codes would be shared.

Response *Analyses were not preregistered. The data are partly legacy data we are not allowed to freely share. Other data cannot be freely shared due to differences in ownership and the nature of the ethical and data protection approvals that are in place for each subsample. Requests for access can be directed to the data owners. A Data availability statement is included in the manuscript: “Data availability. Data supporting the results of the current study are available from the PI of each sub-study on request, given appropriate ethics and data protection approvals. Contact information can be obtained from the corresponding authors. UK Biobank data requests can be submitted to <http://www.ukbiobank.ac.uk>. Most of the r-code for the statistical analyses are provided in SI.”*

Reviewer 4 – Table 1, please add β s and effect size measures for meta-analyses

Response We used GAMMs to test the sleep – brain relationships, which unfortunately do not produce effect size measures like β s. Instead, effect sizes are presented in Figures 3 and 4. Here, we show the effect on the brain by variations in sleep duration. Although not similar to β s, we believe these give a good indicating of the strength of the sleep duration – brain relationship for each structure and region.

Reviewer 4 – L. 242 what are 'modest effect sizes'?

Response We agree that this is imprecise language. We have now deleted this part of the sentence.

Reviewer 4 – Please add CIs to Figs. 3+4

Response We understand the reviewer's request. However, regarding Figure 4, the red dots just show the average reported sleep duration, and it would be less relevant to present CIs for this. While we could add CIs around the fit lines themselves, this would represent CIs for percentage deviation from 100% volume, which we believe is less relevant. However, we do present standard errors for the estimated sleep duration associated with the peak thickness/volume in Table 2.

Reviewer #5:

Remarks to the Author:

Thank you for the opportunity to review this interesting manuscript investigating the relationships between brain structure and sleep. As requested, I am only commenting on the genetics and MR analyses, which are in general sound and appropriate.

Response Thank you for the generally positive evaluation of our manuscript.

Comments:

GWAS:

Please provide a little bit more background on why you split the sample. You would have more power to detect SNP for sleep duration when combining the samples and use sleep duration as continuous outcome. This might be useful also for later MR analyses and sensitivity analyses.

Response: We regret that the rationale for our choice was not clearly explained. Since it is established that the relationship between sleep duration and health, including brain health, is inverse U-shaped, both short and long sleep is associated with poorer health. Importantly, it is possible – and even likely – that the genetic contributions to sleep duration and brain health are different in short compared to long sleepers. This was seen in the primary GWAS paper on sleep duration (Dashti, H.S. et al. Genome-wide association study identifies genetic loci for self-reported habitual sleep duration supported by accelerometer-derived estimates. *Nature Communications* 10, 1100 (2019)). For this reason, we decided that to split the sample was a better choice than using sleep duration as a continuous outcome. As the mean reported sleep duration was 7 hours, we used this as a cut-off. To make our reasoning clear to the reader, the following was added to the revised manuscript (Results, section on the genetic analyses): “The sample was stratified into shorter (< 7 hours) and longer (>7 hours) than average sleepers. Since an inverse U-shaped relationship between sleep duration and health – including brain health – is established, both short and long sleep is associated with poorer health. Importantly, the genetic contributions to sleep duration and brain health may be different in short compared to long sleepers⁵⁰, and hence different relationships were expected in these two groups.”

MR:

Consider including the MR-STROBE reporting guidelines, which is best practice for MR studies. Here’s the link: <https://www.strobe-mr.org/>.

Response: We agree with the reviewer that this is a good idea. We have therefore completed the recommended checklist for MR-STROBE for the items we believe are appropriate. The checklist is now included in the Supplemental Information. As the present study is not a pure MR study, MR is only one element, and we therefore decided that adding “Mendelian Randomization study” to the title would tend to be misleading. We refer to the checklist at the beginning of the presentation of the MR results: “We performed bidirectional Mendelian randomization analysis for each brain volumetric trait to sleep duration (see also SI STROBE-MR-checklist, reporting according to best practice for MR studies).”

Instrument selection: Please consider adding the number of SNPs selected for the instrument to the main manuscript and also include F-statistic for instrument strength. Related, the authors note that the sleep MR instruments may not explain enough of the variance, which they and they end up also incorporating a relaxed P-value threshold to construct their instruments included as sensitivity analyses. I would suggest a power calculation using <https://shiny.cnsgenomics.com/mRnd/> to assess whether their sleep brain MR analyses

were sufficiently powered to detect associations. If these analyses are sufficiently powered, then it improves the inference of a null association in this direction.

Response: Thank you for this comment. We have now added the number of SNPs and F-statistics for instrument strength. The following is included in the manuscript (Results, section on GWAS, ...): "Among the 12 pairs, ICV showed a significant causal effect (34 instrumental SNP, minimal F stats > 24; inverse-variance weighted beta = 0.060, se = 0.017, p = 5.36x10⁻⁴) on sleep duration for the sho Instrumental variables), with no causal effect

Regarding statistical power: Our argument that sleep-duration traits are statistically underpowered for being exposures in any MR analysis was based on a power analysis we have previously performed, using the model proposed by Freeman et al. (Freeman, G., Cowling, B.J. & Schooling, C.M. Power and sample size calculations for Mendelian randomization studies using one genetic instrument. International Journal of Epidemiology 42, 1157-1163 (2013)). The Brion model suggested by the reviewer is well suited for situations where abundant correlational studies

have been performed and the unadjusted and adjusted effect sizes are available. The Brion model can then use these correlations to calculate the required sample size and provide power estimates. However, in the case of sleep duration vs brain structure, almost no such studies are available. Hence, we believe the advantage of the Brion model is not obvious in this case. The Freeman model is simpler but could underestimate power in certain situations. In both models, the case parameter is the strength of the instrument, parameterized as the proportion of exposure variance accounted for by selected instrumental SNPs. We used the Steiger model (Hemani, G., Tilling, K. & Davey Smith, G. Orienting the causal relationship between imprecisely measured traits using GWAS summary data. PLOS Genetics 13, e1007081 (2017)) to estimate these proportions for the trait we study in the present manuscript: HippV (0.06); ICV (0.05), TGV (0.02), short-sleep (0.001), long-sleep (0.0007). We then have used these estimates, assuming a range of true causal effects, to regenerate power curves (see figure). With the current sample sizes (MRI ~30000, dashed line on the figure), we do have a power >=0.8 for hippocampal volume and ICV given a true causal effect larger than 0.3. However, for sleep traits, the needed sample size may be on the order

of million to achieve the same power. It is possible that that other advanced MR models may be powerful enough to use smaller samples. We have added the following to the manuscript (section on genetic results): "The low heritability for sleep in the study resulted in a weaker genetic instrument. While we were powered (>80%) to detect a true causal effect for hippocampal volume and ICV of 0.3 or larger, the low heritability for sleep in the study required a much larger sample size, based on the Freeman model for power calculations in Mendelian randomization studies⁷⁶."

I wonder if the stratification of the UK Biobank participants by sleep duration (≤ 7 hours versus ≥ 7 hours) may be missing a global effect. Perhaps it would be worthwhile to include as a sensitivity analysis MR of the overall sleep duration GWAS, which is publicly available and would be easy to run: https://gwas.mrcieu.ac.uk/datasets/?trait__icontains=sleep duration (see also above for GWAS).

Response: *We understand the reviewer's point, but believe previous research strongly suggest analyses stratified by sleeping more or less than average sleep duration. We have performed MR analysis for overall sleep duration using the GWAS summary statistics from Dashti et al. mentioned above. The effect of ICV to overall sleep duration is not significant (33 SNP, IVW-beta=0.03, p=0.1). It is of note that this GWAS contains all samples we used for ICV GWAS, with 100% overlap. Thus, the estimated causal effect will, by theory, be biased towards the direction of the phenotype-level correlation. In response to the reviewer's comment, we have now included the tables of the instrumental variables in the Supplemental Information (SI Instrumental variables ICV SleepDur snps).*

Please address the issue that some relationships between mental health/brain structure/sleep might be impacting the brain structure sleep MR findings. You could consider running a multivariate MR to account for major comorbidities, including BMI, psychiatric disorders and alcohol/tobacco use. Or Perhaps as a sensitivity analysis, you could perform a look up of the brain structure SNP instruments for their associations with major psychiatric disorders. Are there corresponding associations, for example, with depression? If not then it may be interesting to note. Conversely, if there are, then it may be worthwhile, to perform a quick sensitivity analysis MR where you construct a brain structure instrument leaving those variants out. Similarly, you could perform a multivariable MR analysis to further test the robustness of the finding: <https://wspiller.github.io/MVMR/articles/MVMR.html>

Response: *We thank the reviewer for these excellent suggestions, and we have now performed both the strategies recommended by the reviewer. We have now performed MVMR using data from the largest GWAS for schizophrenia (SCZ) (Trubetsky, V. et al.*

Mapping genomic loci implicates genes and synaptic biology in schizophrenia. Nature 604, 502–508 (2022)), bipolar (BIP) (Mullins, N. et al. Genome-wide association study of more than 40,000 bipolar disorder cases provides new insights into the underlying biology. Nature Genetics 53, 817–829 (2021)), major depression (MDD, European sample without 23andMe and UKBB) (Wray, N.R. et al. Genome-wide association analyses identify 44 risk variants and refine the genetic architecture of major depression. Nat Genet 50, 668–681 (2018).), body mass index (BMI, European sample including UKBB) (Yengo, L. et al. Meta-analysis of genome-wide association studies for height and body mass index in ~700000 individuals of European ancestry. Hum Mol Genet 27, 3641–3649 (2018)), cigarette per day (CPD, European sample including UKBB) (Saunders, G.R.B. et al. Genetic diversity fuels gene discovery for tobacco and alcohol use. Nature 612, 720–724 (2022)) and drinks per week (DPW, European sample including UKBB) (Saunders et al. 2022). The psychiatric GWAS is from largely independent samples and is thus less likely to generate phenotype-level correlations. However, the rest of the data inevitably bring in phenotypic covariance, as we have not been able to locate dataset with sufficient power without including UKB data. We are aware that the GWAS for BMI published in 2015 included part of UKB, but only SNPs in the Hapmap2 are available for this data. Using the 2015 BMI GWAS made the total instrumental SNPs drop from 1100 to 500, and hence greatly reduced the power of the MVMR, and we therefore show results based on the latest BMI data. In the table we show the effect of ICV on short sleep after accounting for the other six covariates. Although the ICV effect is still significant, MVMR would need even larger data to avoid weak instrument bias. From the conditional F statistics, we see that weak instruments are the rule rather than the exception (<10).

Exposure	Beta	SE	P	Conditional F
ICV	0.044097438	0.010840220	5.111632e-05	2.547072
SCZ	-0.005964905	0.004813331	2.155415e-01	4.400903
BIP	0.007461740	0.006561274	2.557073e-01	2.923795
MDD	-0.049188770	0.010320705	2.153741e-06	1.989265
BMI	-0.042138188	0.014207058	3.087816e-03	18.83428
CPD	-0.062593181	0.015997048	9.732912e-05	8.103234
DPW	0.027281699	0.021747253	2.099538e-01	8.888054

As the MVMR results may be contaminated by weak instruments and/ or overlapping samples, we further performed ad hoc analyses, by excluding instrumental SNPs for ICV showing associations with any of the six potentially confounding traits or disorders. SNPs with association $p < 10^{-6}$ with any of these traits/disorders were excluded from the instrument list, which reduced the total number of instruments for ICV from 34 to 26. The effect of ICV on sleep duration in short-sleepers was still significant (IVW beta=0.057, se=0.019, p=0.0032). We believe these results are reassuring, but as discussed above, does not justify being included in the main manuscript due to among other things overlapping samples. The

following is included in the discussion: "Finally, the Mendelian randomization analyses showed causal effects of ICV on sleep duration but no causal effects of sleep duration on any MRI-derived brain measure, an effect that was robust after accounting for confounding factors (BMI, smoking and drinking habits, neuropsychiatric disorders, see SI). However, there were no causal effects of sleep duration on any MRI-derived brain measure."

Decision Letter, second revision:

5th June 2023

Dear Dr. Fjell,

Thank you for your patience as we've prepared the guidelines for final submission of your Nature Human Behaviour manuscript, "Sleep duration and brain structure – phenotypic associations and genotypic covariance" (NATHUMBEHAV-22030522B). Please carefully follow the step-by-step instructions provided in the attached file, and add a response in each row of the table to indicate the changes that you have made. Please also address the additional marked-up edits we have proposed within the reporting summary. Ensuring that each point is addressed will help to ensure that your revised manuscript can be swiftly handed over to our production team.

We would hope to receive your revised paper, with all of the requested files and forms within two-three weeks. Please get in contact with us if you anticipate delays.

Nature Human Behaviour offers a Transparent Peer Review option for new original research manuscripts submitted after December 1st, 2019. As part of this initiative, we encourage our authors to support increased transparency into the peer review process by agreeing to have the reviewer comments, author rebuttal letters, and editorial decision letters published as a Supplementary item. When you submit your final files please clearly state in your cover letter whether or not you would like to participate in this initiative. Please note that failure to state your preference will result in delays in accepting your manuscript for publication.

In recognition of the time and expertise our reviewers provide to Nature Human Behaviour's editorial process, we would like to formally acknowledge their contribution to the external peer review of your manuscript entitled "Sleep duration and brain structure – phenotypic associations and genotypic covariance". For those reviewers who give their assent, we will be publishing their names alongside

the published article.

Cover suggestions

As you prepare your final files we encourage you to consider whether you have any images or illustrations that may be appropriate for use on the cover of Nature Human Behaviour.

ORCID

Non-corresponding authors do not have to link their ORCIDs but are encouraged to do so. Please note that it will not be possible to add/modify ORCIDs at proof. Thus, please let your co-authors know that if they wish to have their ORCID added to the paper they must follow the procedure described in the following link prior to acceptance:

Nature Human Behaviour has now transitioned to a unified Rights Collection system which will allow our Author Services team to quickly and easily collect the rights and permissions required to publish your work. Approximately 10 days after your paper is formally accepted, you will receive an email in providing you with a link to complete the grant of rights. If your paper is eligible for Open Access, our Author Services team will also be in touch regarding any additional information that may be required to arrange payment for your article. Please note that you will not receive your proofs until the publishing agreement has been received through our system.

Please note that *Nature Human Behaviour* is a Transformative Journal (TJ). Authors may publish their research with us through the traditional subscription access route or make their paper immediately open access through payment of an article-processing charge (APC). Authors will not be required to make a final decision about access to their article until it has been accepted. Find out more about Transformative Journals

Authors may need to take specific actions to achieve compliance with funder and institutional open access mandates. If your research is supported by a funder that requires immediate open access (e.g. according to Plan S principles) then you should select the gold OA route, and we will direct you to the compliant route where possible. For authors selecting the subscription

publication route, the journal's standard licensing terms will need to be accepted, including self-archiving policies. Those licensing terms will supersede any other terms that the author or any third party may assert apply to any version of the manuscript.

[REDACTED]

Best regards,
Alex McKay
Editorial Assistant
Nature Human Behaviour

On behalf of

Giacomo Ariani
Editor
Nature Human Behaviour

Reviewer #4:
None

Reviewer #5:
Remarks to the Author:
This is a much improved revision. The genetic / MR concerns were adequately addressed.

Final Decision Letter:

Dear Prof Fjell,

We are pleased to inform you that your Article "No phenotypic or genotypic evidence for a link between sleep duration and brain atrophy", has now been accepted for publication in Nature Human Behaviour.

Please note that *Nature Human Behaviour* is a Transformative Journal (TJ). Authors may publish their research with us through the traditional subscription access route or make their paper immediately open access through payment of an article-processing charge (APC). Authors will not be required to make a

final decision about access to their article until it has been accepted. Find out more about Transformative Journals

With best regards,

Giacomo Ariani
Editor
Nature Human Behaviour